# Fair Ranking with Noisy Protected Attributes

**Anay Mehrotra**
Yale University

**Nisheeth K. Vishnoi**
Yale University

## Abstract

The fair-ranking problem, which asks to rank a given set of items to maximize utility subject to group fairness constraints, has received attention in the fairness, information retrieval, and machine learning literature. Recent works, however, observe that errors in socially-salient (including protected) attributes of items can significantly undermine fairness guarantees of existing fair-ranking algorithms and raise the problem of mitigating the effect of such errors. We study the fair-ranking problem under a model where socially-salient attributes of items are randomly and independently perturbed. We present a fair-ranking framework that incorporates group fairness requirements along with probabilistic information about perturbations in socially-salient attributes. We provide provable guarantees on the fairness and utility attainable by our framework and show that it is information-theoretically impossible to significantly beat these guarantees. Our framework works for multiple non-disjoint attributes and a general class of fairness constraints that includes proportional and equal representation. Empirically, we observe that, compared to baselines, our algorithm outputs rankings with higher fairness, and has a similar or better fairness-utility trade-off compared to baselines.

## 1 Introduction

Given a query and a set of $m$ items, ranking problems require one to output an ordering of a small subset of items in decreasing order of *relevance* to the query. Such ranking problems have been extensively studied in the information retrieval [46] and the machine learning [45] literature, and algorithms for them are used in applications such as search engines, personalized feed generators, and online recruiting platforms [44, 12, 8]. Several studies have observed that when the outputs of ranking algorithms are consumed by end-users, e.g., image results for occupation-related queries, articles with different political leanings, and job applicants in online recruiting, the outputs can mislead or alter their perceptions about socially-salient groups [38], polarize their opinions [24, 50], and affect economic opportunities available to individuals [32]. A reason is that relevance (or utilities) input to ranking algorithms may be influenced by human or societal biases, leading to output rankings that skew representations of socially-salient, and often legally-protected, groups such as women and Black people [55].

A growing number of works aim to make the output of ranking algorithms *fair* with respect to socially-salient attributes [74, 75, 58]. As for notions of fairness, in the case when each item belongs to one of two socially-salient groups ($G_1$ or $G_2$), equal representation requires that, for every $k$, (roughly) $\frac{k}{2}$ items from each of $G_1$ and $G_2$ appear in the first $k$ positions of the output ranking. Proportional representation requires that at most $k \cdot \frac{|G_\ell|}{m}$ items from each $G_\ell$ appear in the first $k$ positions. Fairness criteria that generalize proportional representation and involve $p \geq 2$ groups $G_1, \ldots, G_p$, where each item may belong to multiple groups, have also been considered: Given values $U_{k\ell}$, they require that at most $U_{k\ell}$ items from $G_\ell$ appear in the first $k$ positions of the output ranking [61, 18]. One set of works in the fair-ranking literature tries to improve fairness in utility-estimation [72, 62, 73, 51]. Such approaches have the benefit that no changes to the existing ranking algorithm are necessary but they may be unable to guarantee that the output ranking satisfies the required fairness criteria [27]. Another set of works use the given utilities as-it-is and change the ranking algorithm to output

36th Conference on Neural Information Processing Systems (NeurIPS 2022).

the ranking with the highest utility subject to satisfying the specified fairness criteria by including them as *fairness constraints* [61, 9, 18, 27, 30]. While these latter approaches can guarantee fairness, they require coming up with new algorithms to solve the arising constrained ranking problems. Both approaches, however, rely on knowledge of the socially-salient attributes of the items [56].

Assuming precise access to socially-salient attributes is reasonable in some contexts and has led to successful deployment of fair-ranking frameworks; see [27]. However, in several contexts, socially-salient attributes can be erroneous, missing, or known only probabilistically. For instance, errors can arise due to misreporting, which is a common concern with self-reported attributes [4]. Attributes can also be missing, as is the case with images in web-search or in settings where it is illegal to collect certain socially-salient attributes [20]. Often attributes are predicted using ML-classifiers, but such prediction has inaccuracies [10]. In such cases, one can calibrate the confidence scores of classifiers to derive (aggregate) probabilistic information about the true attributes [35]. Moreover, probabilistic information about socially-salient (protected) attributes can be sometimes computed from other attributes. For instance, name and location of an individual, combined with aggregate census data may be used to get a conditional distribution of their race [23, 36, 20]. Even accurate attributes may be randomly and independently flipped to preserve user privacy, and the distribution of flipped attributes is determined by public parameters of, e.g., the randomized response mechanism [37, 69].

Several models of inaccuracies in data have been proposed [47, 26]. We consider one such model (due to [5]) to capture inaccuracies in socially-salient attributes. Each item $i$ belongs to the $\ell$-th group with a known probability $P_{i\ell}$. For each item $i$, the distribution corresponding to $P_{i\ell}$s over groups is assumed to be independent of corresponding distributions of other items. This model can be used in cases where these probabilities are available or can be derived, as in some of the aforementioned examples (see Section 5 and Supplementary Material A). In other cases, e.g., when errors are strategic or adversarial, other models are needed. This model and its variants have also been used by works on designing fair algorithms in the presence of inaccuracies, for problems including classification [42, 67, 66, 14], subset selection [48], and clustering [25] (Supplementary Material B briefly discusses these works).

In this noise model, while socially-salient attributes are not explicitly specified, one could still use existing fair-ranking algorithms by first sampling groups for items from the given probabilities. Indeed, [29] evaluate existing fair-ranking algorithms on attributes obtained from the probabilities derived from ML classifiers. They find that "errors in [socially-salient attributes] can dramatically undermine fair-ranking algorithms" and can cause "[non-disadvantaged groups] to become disadvantaged after a 'fair' re-ranking." We confirm this observation on a synthetic dataset when the goal is to finding a ranking that satisfies equal representation (Section 5.1). We assigned each item the socially-salient group that is most likely and find that when existing fair-ranking algorithms (for equal representation) are run with this group information, they output rankings that significantly violate the equal representation criteria (Figure 1). Further, we mathematically analyze two natural methods to sample groups from probabilities and give examples where taking such information as input, existing fair-ranking algorithms output rankings which provably violate the equal representation criteria (Supplementary Material C). Thus, new ideas are needed to design fair-ranking frameworks that can guarantee given fairness criteria under this noise model.

**Our contributions.** We present a fair-ranking framework that guarantees given fairness criteria when the socially-salient attributes are assumed to follow the probabilistic noise model mentioned above. In particular, it finds a utility maximizing ranking subject to a class of constraints that only rely on given probability distributions (Program (7)). These constraints relax the given fairness criteria by a carefully chosen factor: for equal representation, the relaxation is by roughly a $1 + \frac{1}{\sqrt{k}}$ multiplicative factor for position $k$ for any $k$. Moreover, instead of sampling the attribute values and applying constraints on them, these constraints apply the relaxed-fairness criteria to the expected number of items from each group that appear in the first $k$ positions. We show that these constraints ensure that any ranking approximately satisfying the given fairness criteria is feasible for them and any ranking feasible for them approximately satisfies the given fairness criteria (Theorem 4.1). Our fair-ranking framework works for the general class of fairness criteria introduced earlier, which involve multiple overlapping groups $G_1, \ldots, G_p$ and upper bound $U_{k\ell}$ for the $\ell$-th group and $k$-th position (Theorem 4.1), and for their position-weighted versions (Theorem F.1).

We show that our fair-ranking framework, besides nearly satisfying the given fairness criteria, has a provably high utility (Theorem 4.1). Complementing Theorem 4.1, we prove near-tightness of the fairness guarantee (Theorem 4.2): for equal representation fairness criteria, this results shows that it

is information theoretically impossible to output a ranking that violates this criteria by less than a multiplicative factor of $1 + \widetilde{O}(\frac{1}{\sqrt{k}})$ at the $k$-th position for any $k$. Finally, we give a polynomial-time algorithm to approximately solve Program (7) (Theorem 4.3).

Empirically, we evaluate our framework on both synthetic and real-world data against standard metrics like weighted-risk difference (RD) that measure deviation from specific fairness criteria (Section 5). We compare its performance to key baselines [18, 61, 27, 48] on both single and multiple attributes. In all simulations, compared to baselines, our framework has a higher maximum fairness (2-10% for RD; Figures 1 to 3) and a similar/better fairness-utility trade-off (Figures 2, 8, 10 and 14 to 16).

## 2  Related work

Work on automated information retrieval dates back to 1940s [43, 21]. Since then the IR literature has devoted a significant effort in measuring relevance of items to specific queries across different tasks: including, web search [7], personalization [34], and product rating [22]; we also refer the reader to [46] and the references therein. In the last three decades, works in the ML literature have also made significant contributions to relevance-estimation [45], by proposing methods that: (1) supplement traditional IR approaches, e.g., by automatically tuning their–previously hard to tune–parameters [65] and by improving their efficiency through clustering-based techniques [64, 3], and (2) substitute traditional IR approaches by neural-network based models to predict item relevance [12, 11, 68, 8].

**Fair ranking.** Existing works on the fair-ranking problem take diverse approaches: Among works that de-bias utilities, different approaches include, post-processing the utilities so that the post-processed utilities satisfy some fairness requirement [71], introducing a "fairness penalty" in the objective function used to train learning-to-rank models [62, 73, 49], and modifying feature representations generated by up-stream algorithms so that the utilities learned from the modified representations satisfy some fairness requirements [72]. Works that alter the ranking algorithms can also be further categorized into those which satisfy the constraints for each ranking [18, 70, 27, 30] and those that satisfy the constraints in aggregate over multiple rankings [61, 9]. Among aforementioned works, [49] uses a version of adversarial training to make (fair) learning-to-rank models robust to outliers but, unlike this work, they require socially-salient attributes of items to be accurately known to specify the "fairness penalty." All of the other aforementioned works also need access to the socially-salient attributes of items. When protected attributes are inaccurate, these works can fail to satisfy their fairness and/or utility guarantees [29].

**Effect of inaccuracies on fair-ranking algorithms.** Some recent works have considered assessing fairness of rankings and ranking algorithms with missing or inaccurate protected attributes. [39] analyze the setting where all protected attributes are missing, but can be purchased at a fixed cost per item. They give statistical-techniques to estimate the fairness-value of a given ranking at a small cost. [29] use ML-classifiers to infer protected attributes from real-world data and study performance of the fair-ranking algorithm by [28] when given inferred attributes as input. While these works underscore the need for fair-ranking algorithms to be robust to inaccuracies in protected attributes, they only assess fairness in the presence of noisy protected attributes.

## 3  Model of fair ranking with noisy attributes

**Ranking problem.** In ranking problems, given $m$ items, one has to select a subset of $n$ items and output a permutation of the selected items. This permutation is said to be a *ranking*. There is a large body of work on estimating the relevance of items and personalizing these estimates to specific users/queries [46, 45]. We consider a ranking problem where the relevance of items are known. Abstracting relevance estimation, in this problem, one is given an $m \times n$ matrix $W$, such that placing the $i$-th item at the $j$-th position generates *utility* $W_{ij}$. The utility of a ranking is the sum of utilities generated by each item in its assigned position. The algorithmic task in the ranking problem is to output a ranking with the highest utility. We denote rankings by assignment matrices $R \in \{0, 1\}^{m \times n}$, where $R_{ij} = 1$ indicates that item $i$ appears in position $j$, and $R_{ij} = 0$ indicates otherwise. In this notation, the utility of a ranking is $\langle R, W \rangle := \sum_{i=1}^{m} \sum_{j=1}^{n} R_{ij} W_{ij}$. Then this ranking problem is to solve: $\max_{R \in \mathcal{R}} \langle R, W \rangle$. Where $\mathcal{R}$ is the set of all assignment matrices denoting a ranking:

$$\mathcal{R} := \left\{ X \in \{0, 1\}^{m \times n} : \forall i \in [m], \sum_{j=1}^{n} X_{ij} \leq 1, \ \forall j \in [n], \sum_{i=1}^{m} X_{ij} = 1 \right\}. \qquad (1)$$

Here, the constraint $\sum_{i=1}^{m} X_{ij} = 1$ ensures position $j$ has exactly one item and the constraint $\sum_{j=1}^{n} X_{ij} \leq 1$ ensures that item $i$ occupies at most one position. While this model captures a variety of applications, in some cases, the entries of $W$ may be skewed by an unknown amount [40, 16] or not known accurately [63] and the utility of the ranking may not be linear in the entries of $W$ [2]. These are interesting directions but are not studied in this work.

**Fair-ranking problem.** There are several versions of the fair-ranking problem. We consider a version with $p \geq 2$ *socially-salient groups* $G_1, G_2, \ldots, G_p \subseteq [m]$ (e.g., the group of all women or all Black people) which are often protected by law. Each of the $m$ items belongs to *one or more* of these socially-salient groups (henceforth referred to as just groups). This fair-ranking problem is to output the ranking with maximum utility subject to satisfying certain fairness criteria with respect to these groups. The appropriate notion of fairness is context dependent, and to capture different fairness criteria numerous *fairness constraints* have been proposed. We consider a class of general fairness constraints.

**Definition 3.1 (Fairness constraints).** Given a matrix $U \in \mathbb{Z}_+^{n \times p}$, a ranking $R$ satisfies the upper bound constraint if $\sum_{i \in G_\ell} \sum_{j=1}^{k} R_{ij} \leq U_{k\ell}$, for all $\ell \in [p]$ and $k \in [n]$.

Existing works consider similar constraints and show that they can encapsulate a variety of fairness criteria [61]. For instance, when groups are disjoint, to capture equal and proportional representation, one can choose $U_{k\ell} := \left\lceil \frac{k}{p} \right\rceil$ and $U_{k\ell} := \left\lceil k \cdot \frac{|G_\ell|}{m} \right\rceil$ for all $k$ and $\ell$ respectively. (That said, they do not capture qualitative differences among groups, such as, misrepresentation of demographics in image results [38, 55], which could arise even when rankings has sufficient individuals from each group.) As a running example, we consider the fair-ranking problem with equal representation with two disjoint groups, i.e.,

$$\max_{R \in \mathcal{R}} \langle R, W \rangle \quad \text{s.t.} \quad \forall k \in [n] \ \forall \ell \in [2], \quad \sum_{i \in G_\ell} \sum_{j=1}^{k} R_{ij} \leq \left\lceil \frac{k}{2} \right\rceil. \tag{2}$$

To ease readability, we omit ceilings-operators henceforth.

**Noise model.** If the socially-salient attributes of items are known accurately, then one can solve the fair-ranking problem. However, as discussed, in many contexts, attributes are inaccurate, missing, or only probabilistically known. Several models have been proposed to capture different errors in attributes. Here, we consider a model (due to [5]) which has also appeared in [25, 42, 48].

**Definition 3.2 (Noise model).** Let $P \in [0,1]^{m \times p}$ be a known matrix. The groups $G_1, \ldots, G_p \subseteq [m]$ are random variables, such that, for each $i \in [m]$ and $\ell \in [p]$, $\Pr[G_\ell \ni i] = P_{i\ell}$. Moreover, for different items $i \neq j$ the events $G_\ell \ni i$ and $G_k \ni j$ are *independent* for all $\ell, k \in [p]$.

Definition 3.2 makes two key assumptions: the matrix $P$ is known and for each item $i$, the events $G_\ell \ni i$ over groups $\ell$ are independent of the corresponding events for other items. Both of these assumptions hold when attributes are flipped to preserve local differential privacy (Remark A.1). In other settings, $P$'s estimate can be inaccurate and above events may be correlated. These can adversely affect the performance of our framework. We empirically study this in simulations where $P$ is estimated using confidence scores of off-the-shelf classifiers and is *miscalibrated* (Figures 2 and 3). Supplementary Material A shows how Definition 3.2 captures both disjoint and overlapping groups.

**Fairness constraint with noisy attributes.** Most existing fairness constraints assume that the groups are deterministic. Hence, it is not clear how to impose them when groups are random variables, as in Definition 3.2. One definition is to require the constraints to be approximately satisfied with high probability. Consider the instantiation of this definition for equal representation: A ranking $R$ satisfies $(\rho, \delta)$-equal representation, if with probability $1 - \delta$, at most $\frac{k}{2}(1 + \rho)$ items from $G_\ell$ appear in the first $k$ positions in $R$ places for all $k \in [n]$ and $\ell \in [p]$. Naturally, one would like to satisfy this definition for small $\delta, \rho$. However, it turns out to be too stringent and is infeasible for any small $\delta, \rho$.

**Proposition 3.3.** *No ranking satisfies $(\rho, \delta)$-equal representation for $\rho < 1$, $\delta \leq \frac{1}{2}$, and $P = \left[\frac{1}{2}\right]_{m \times p}$.*

The proof of Proposition 3.3 shows that any ranking $R$ violates the equal-representation constraint at the 2nd position by a multiplicative factor of 2 with probability $\frac{1}{2}$. The issue is that the same relaxation parameter $\rho$ is used for each position $k$ (whereas the information theoretically best-achievable relaxation parameter at $k$ improves as $k$ increases, this, e.g., follows by Theorem 4.1.) Motivated by this observation, we consider the following alternate version of upper bound constraints.

**Definition 3.4** (($\varepsilon, \delta$)-**constraint**). For any $\varepsilon \in \mathbb{R}^n_{\geq 0}$ and $\delta \in (0, 1]$, a ranking $R$ is said to satisfy $(\varepsilon, \delta)$-constraint if with probability at least $1 - \delta$ over the draw of $G_1, \ldots, G_p$

$$\forall k \in [n] \; \forall \ell \in [p], \quad \sum_{i \in G_\ell} \sum_{j=1}^{k} R_{ij} \leq U_{k\ell}(1 + \varepsilon_k). \tag{3}$$

We would like to output a ranking that satisfies Definition 3.4 for small $\delta$ and small $\varepsilon_1, \varepsilon_2, \ldots, \varepsilon_n$.

*Problem* 3.5 (**Ranking problem with noisy attributes**). Given matrices $P$, $U$, and $W$, find the ranking $R$ maximizing utility $\langle R, W \rangle$ subject to satisfying $(\varepsilon, \delta)$-constraint for some small $\varepsilon$ and $\delta$.

### 3.1 Challenges in solving Problem 3.5

In this section we discuss potential approaches for solving Problem 3.5. In other words, solving:

$$\max_{R \in \mathcal{R}} \langle R, W \rangle, \; \text{s.t.,} \; R \text{ satisfies } (\varepsilon, \delta)\text{-constraint.} \tag{4}$$

Even for two disjoint groups, given $V$, it is **NP**-hard to decide if the value of Program (4) is at least $V$ (Theorem E.12). To bypass this hardness, one can consider approximation algorithms. Program (4) is an integer program (IP) because the entries of the matrix $R$ are required to be integers (Equation (1)). A standard approach to (approximately) solve IPs is to: (1) consider their continuous relaxation that drops the integrality constraints, (2) compute the optimal solution $R_c$ of the relaxed problem, and then (3) "round" $R_c$ to satisfy integrality constraints while "retaining" its utility and fairness properties. To take this approach, we first need an efficient algorithm to find $R_c$. However, not just Program (4), but even its continuous relaxation is non-convex. Hence, it is unclear how to solve it to find $R_c$.

Due to the independence assumption in Definition 3.2, the number of items from $G_\ell$ appearing in the first $k$ positions of a ranking is concentrated around its expectation (for large $k$). This implies that if, in expectation, less that $U_{k\ell}$ items from $G_\ell$ appear in the top $k$ positions then, with high probability, the number of items from $G_\ell$ in the top $k$ positions is not much larger than $U_{k\ell}$. Using this one can show that a ranking satisfying the following constraints

$$\forall k \in [n] \; \forall \ell \in [p], \quad \mathbb{E}\left[\sum_{i \in G_\ell} \sum_{j=1}^{k} R_{ij}\right] \leq U_{k\ell} \tag{5}$$

also satisfies $(\varepsilon, \delta)$-constraint for small $\varepsilon$ and $\delta$. One idea is to find the ranking maximizing utility subject to satisfying Constraint (5). A feature of Constraint (5) is that it is linear in $R$ as $\mathbb{E}[\sum_{i \in G_\ell} \sum_{j=1}^{k} R_{ij}] = \sum_{i=1}^{m} \sum_{j=1}^{k} P_{i\ell} R_{ij}$ and, hence, one may hope to find the ranking with the maximum utility subject to satisfying Constraint (5). However, the issue is that there are examples where any ranking satisfying Constraint (5) has 0 utility and there are rankings that satisfy $(\varepsilon, \delta)$-constraint and have a large positive utility (Lemma E.10). Hence, this approach can output rankings whose utility is significantly smaller than the utility of the solution to Problem 3.5. To overcome this, we relax Constraint (5) by a carefully chosen position-dependent factor, such that, any ranking satisfying the $(\varepsilon, \delta)$-constraint (for appropriate $\varepsilon$ and $\delta$) is also feasible for our framework.

## 4 Theoretical results

In this section we present our optimization framework and its fairness and utility guarantees.

---

*Input:* Matrices $P \in [0, 1]^{m \times p}$, $W \in \mathbb{R}^{m \times n}_{\geq 0}$, $U \in \mathbb{R}^{n \times p}$

*Parameters:* Constant $c > 1$, failure probability $\delta \in (0, 1]$, and $k \in [n]$, relaxation parameter

$$\gamma_k := 12 \cdot \log\left(\frac{2np}{\delta}\right) \cdot \max_{\ell \in [p]} \sqrt{\frac{1}{U_{k\ell}}}. \tag{6}$$

*Our Fair-Ranking Program*

$$\max_{R \in \mathcal{R}} \langle R, W \rangle, \qquad \text{(Noise Resilient)} \tag{7}$$
$$\text{s.t. } \forall \ell \in [p] \; \forall k \in [n]$$
$$\sum_{\substack{i \in [m], \\ j \in [k]}} P_{i\ell} R_{ij} \leq U_{k\ell}\left(1 + \left(1 - \frac{1}{2\sqrt{c}}\right)\gamma_k\right). \tag{8}$$

---

The above program modifies the program for fair ranking with accurate groups: It has the same objective but different constraints. Instead of sampling the attribute values and applying constraints on the sampled values, Constraint (8) applies upper bounds on the expected number of items in the first $k$ positions from group $\ell$ (Section 3.1). Further, Constraint (8) relaxes upper bounds $U_{k\ell}$ by a small position-dependent factor. Like for Constraint (5), one can show that any ranking satisfying Constraint (8) also satisfies $(\varepsilon, \delta)$-constraint (for small $\varepsilon_1, \ldots, \varepsilon_n$ and $\delta$). But unlike Constraint (5),

and somewhat surprisingly, any ranking that satisfies $(\varepsilon, \delta)$-constraint (for appropriate $\varepsilon_1, \ldots, \varepsilon_n$ and $\delta$) must also satisfy Constraint (8). (In fact, $\gamma_k$ is chosen to be the smallest, up to logarithmic factors, value such that this is true.) We use this to prove Theorem 4.1's utility guarantee.

Our first result bounds the fairness and utility of the optimal solution of Program (7).

**Theorem 4.1.** *Let $\gamma \in \mathbb{R}^n$ be as defined in Equation* (6). *There is an optimization program (Program* (7)*), parameterized by a constant $c$ and failure probability $\delta$, such that for any $c > 1$ and $\delta \in (0, \frac{1}{2}]$ its optimal solution satisfies $(c\gamma, \delta)$-constraint and has a utility at least as large as the utility of any ranking satisfying $((c - \sqrt{c})\gamma, \delta)$-constraint.*

For equal representation, $\gamma_k$ is $\widetilde{O}\left(\frac{1}{\sqrt{k}}\right)$. Thus, Theorem 4.1 guarantees that, with high probability, the optimal solution of Program (7) multiplicatively violates equal representation at the $k$-th position by at most $1 + \widetilde{O}\left(\frac{1}{\sqrt{k}}\right)$. Further, this solution's utility is higher than the utility of any ranking satisfying a slight relaxation of this fairness guarantee. Theorem 4.1 can be extended to position-weighted versions of fairness constraints (Theorem F.1), where the fairness constraint is $\sum_{i \in G_\ell} \sum_{j \in [k]} v_j R_{ij} \leq U_{k\ell}$ (for all $k$ and $\ell$) for specified discount factors $v_1 \geq \cdots \geq v_n$ such as NDCG [33]. If we are also guaranteed $U_{k\ell} \geq \psi k$ for some constant $\psi > 0$ and all $k$ and $\ell$, then we can improve $\gamma_k$'s dependence on $\delta$ from $\log \frac{1}{\delta}$ to $\sqrt{\log \frac{1}{\delta}}$ (Supplementary Material E.3). The proof of Theorem 4.1 appears in Section 6.

**Lower bound on fairness guarantee.** Our next result complements Theorem 4.1's fairness guarantee.

**Theorem 4.2.** *There is a family of matrices $U \in \mathbb{Z}_+^{n \times p}$ such that for any $U$ in the family and any parameters $\delta \in [0, 1)$ and $\varepsilon_1, \ldots, \varepsilon_n \geq 0$, if for any position $k \in [n]$, $\varepsilon_k \leq 1$ and $\varepsilon_k < \max_{\ell \in [p]} \sqrt{\frac{1}{2 U_{k\ell}} \log \frac{1}{4\delta}}$ then there exists a matrix $P \in [0, 1]^{m \times p}$, such that it is information theoretically impossible to output a ranking that satisfies $(\varepsilon, \delta)$-constraint. This family, in particular, contains the matrices $U$ corresponding to equal representation and proportional representation constraints.*

Since $\gamma_k$ is $O\left(\log(\frac{np}{\delta}) \cdot \max_\ell \sqrt{\frac{1}{U_{k\ell}}}\right)$, Theorem 4.2 shows that Theorem 4.1's fairness guarantee is optimal up to log-factors. Supplementary Material D.1 proves Theorem 4.2.

**An efficient algorithm.** As for solving our optimization program, it is **NP**-hard to check its feasibility (Theorem E.7). However, because Constraint (8) is linear in $R$, the continuous relaxation of Program (7) is a standard linear program and can be solved efficiently. Our algorithm (Algorithm 1) solves the standard linear programming relaxation of Program (7) to find a solution $R_c$ and then uses a dependent-rounding algorithm by [19] to convert $R_c$ to a ranking. (See Supplementary Material D.2 for brief discussion of why straightforward rounding approaches are insufficient.)

**Theorem 4.3.** *There is a randomized algorithm (Algorithm 1) that given constants $d > 2$, a failure probability $0 < \delta \leq 1$, and matrices $P \in [0, 1]^{m \times p}$ and $W \in [0, 1]^{m \times n}$, outputs a ranking satisfying $(O(d\gamma), \delta)$-constraint and with probability at least $1 - \delta$, and has a utility at least $\left(1 - \frac{1}{d}\right) \cdot V - \widetilde{O}(\sqrt{dn})$, where $V$ is the utility of any ranking satisfying $((d - \sqrt{d})\gamma, \delta)$-constraint. The algorithm runs in polynomial time in $d$ and the bit complexity[*] of the input.*

The tension in setting $d$ is that decreasing $d$ improves the fairness guarantee and the utility guarantee's second term, but worsens the first term in the utility guarantee. Under the mild assumption that $V = \Omega(n)$, increasing $d$ improves the utility guarantee because the first term in the utility guarantee dominates the second term. In this case, the utility guarantee improves to $(1 - \frac{1}{d} - o(1)) \cdot V$. Finally, while Theorem 4.3 requires utilities to be between 0 and 1, it can be extended to any non-negative and bounded utilities by scaling. The proof of Theorem 4.3 appears in Supplementary Material D.2.

## 5 Empirical results

In this section we evaluate our framework's performance on synthetic and real-world data.[†]

**Baselines and metrics.** The correct choice of fairness metric is context-dependent and beyond the scope of this work [60]. To illustrate our results, we arbitrarily fix the fairness metric as weighted

---

[*]The bit complexity of the inputs is the number if bits required to encode the input using the standard binary encoding (which, e.g., maps integers to their binary representation, rational numbers as pair of integers, and vectors/matrices as a tuple of their entries) [31, Section 1.3].

[†]Code for our simulations is available at `https://github.com/AnayMehrotra/FairRankingWithNoisyAttributes`

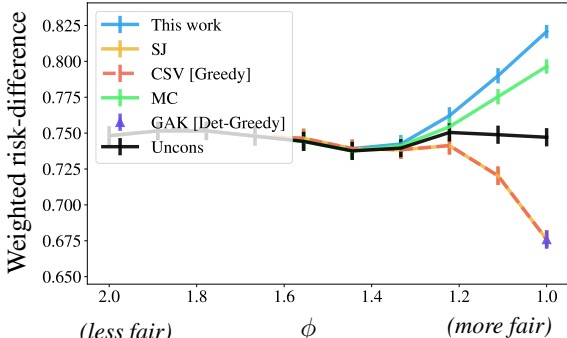

Figure 1: *Synthetic Data: Nonuniform Error Rate.* We consider synthetic data where imputed socially-salient attributes have a higher false-discovery rate on the minority group. We vary the fairness constraint ($\phi$) and observe the weighted risk-difference (RD) of algorithms. The $y$-axis plots RD and $x$-axis plots $\phi$. (*Note that the $x$-axis decreases toward the right*). We observe that **NResilient** achieves the most fair RD, while obtaining a similar utility for all $\phi$ (Figure 8). Error-bars denote the error of the mean.

risk-difference (RD). This is a position-weighted version of the standard risk-difference metric [13] and measures the extent to which a ranking violates equal representation. The RD of a ranking $R$ is:

$$1 - \frac{1}{Z} \sum_{k=5,10,\ldots} \frac{1}{\log k} \max_{\ell,q \in [p]} \left| \sum_{i \in G_\ell, j \in [k]} R_{ij} - \sum_{i \in G_q, j \in [k]} R_{ij} \right|,$$

Where $G$ denotes the ground-truth protected groups and $Z$ is a constant so that RD has range $[0, 1]$. Here, RD $= 1$ is most fair and RD $= 0$ is least fair. We compare our framework, **NResilient**, against state-of-the-art fair-ranking algorithms: **CSV** ("greedy" in [18]), **SJ** [61], and **GAK** ("DetGreedy" in [27]). We also compare against **MC**, which ranks the items, in the subset output by [48]'s algorithm, to maximize utility. Finally, we compare against the baseline, **Uncons**, which outputs the utility maximizing ranking without fairness considerations. We present additional discussion of baselines and results, additional plots for RD, and comparisons with weighted selection-lift (instead of RD) and different choices of $U$ (than the ones below) in Supplementary Material G.

**Setup.** We consider the DCG model of utilities [33] and a relaxation of equal representation constraints: (1) Given an intrinsic value $w_i \geq 0$, for each item $i$, we set $W_{ij} := w_i \left( \log (j + 1) \right)^{-1} \forall j \in [n]$. (2) Given a parameter $\phi \in [1, p]$, we set upper bounds $U_{k\ell} := \frac{\phi}{p} \cdot k$ for each $k \in [n]$ and $\ell \in [p]$. In simulations, we set $m = 500$, $n = 25$, and vary $\phi$ from $p$ to $1$. To gain some intuition about the relevant values of $\phi$, note that satisfying the 80% rule requires $\phi \leq \frac{5p}{5p-1}$, i.e., $\phi \leq 1.11$ for $p = 2$ and $\phi \leq 1.05$ for $p = 4$. For each $\phi$, we draw $m$ items uniformly without replacement and compute an estimate $\widehat{P}$ of the matrix $P$ (from Definition 3.2) using, e.g., off-the-shelf ML classifiers or public APIs (see the paragraphs "Estimating $\widehat{P}$" in simulation with image data and "Setup" in simulation on name data). We infer socially-salient groups $\widehat{G}_1, \ldots, \widehat{G}_p$ via $\widehat{P}$ by assigning each item to its most-likely group. Finally, we run all algorithms using $\widehat{P}$ or $\widehat{G}_1, \ldots, \widehat{G}_p$ as discussed next.

**Implementation details. NResilient** and **MC** take probabilistic information about socially-salient attributes as input and are given $\widehat{P}$. **CSV**, **SJ**, and **GAK** require access to socially-salient groups and are given $\widehat{G}_1, \ldots, \widehat{G}_p$. **NResilient**, **SJ**, and **CSV** use fairness constraints from Definition 3.1 and are given: for each $k$ and $\ell$, $U_{k\ell} = \frac{\phi}{p} \cdot k$. **MC** requires, for each $\ell \in [p]$, an upper bound on the number of items from $G_\ell$ that can appear in top-$n$ positions. It is given $\frac{\phi}{p} \cdot n$ for each $\ell \in [p]$. **GAK** requires the desired proportion $\alpha_\ell$ for each group $G_\ell$ and, roughly, satisfies the constraint $U_{k\ell} = \alpha_\ell \cdot k$ for each $k \in [n]$ and $\ell \in [p]$. It is given $\alpha_\ell = \frac{1}{p}$ for each $\ell \in [p]$, this corresponds to $\phi = 1$ (hence, figures only plot the **GAK** at $\phi = 1$). As a heuristic, we set $\gamma_k = \frac{1}{20} \max_{\ell \in [p]} \sqrt{\frac{1}{U_{k\ell}}}$ in all simulations. We find that this suffices and expect a more refined approach to improve the performance of **NResilient**.

## 5.1 Simulation on synthetic data

We show that on synthetic data, where error-rates of given socially-salient attributes vary over groups, existing fair-ranking algorithms have worse RD than **Uncons**.

**Data.** We generate $w$ and $P$ for two groups using code by [48] and fix $\widehat{P} = P$. For all items $i$, $w_i$ is i.i.d. from the uniform distribution over $[0, 1]$. $\widehat{P}$ is constructed such that attributes inferred from $\widehat{P}$ have a higher false-discovery rate for the minority group compared to the majority (40% vs 10%).[‡]

---

[‡]This 30% difference in false-discovery rates is comparable to the 34% difference in the false-discovery rates of dark-skinned females and light-skinned men observed by [10] for a commercial classifier.

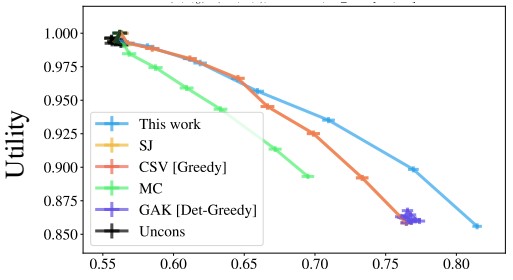

Figure 2: *Real-world image data.* In this simulation, given *non-gender labeled* images and their utilities, our goal is to generate a high-utility gender-balanced ranking. We estimate $P$ using an off-the-shelf ML-classifier and vary $\phi$ from $p = 2$ (less fair) to 1 (more fair). The $y$-axis plots the utility of algorithms and the $x$-axis plots RD. We observe that **NResilient** has the most fair RD and the best fairness-utility trade-off. Error bars show the error of the mean.

**Results.** See Figure 1 for the observed RD averaged over 500 iterations. We observe that **NResilient** achieves best RD ($\approx$0.81), while not losing significant utility ($\geq 98\%$ of max.; see Figure 8). **MC** achieves the next best RD ($\approx$0.79). In contrast, **CSV**, **SJ**, and **GAK**, which do not account for noise in the socially-salient attributes, achieve a worse RD ($\leq$0.68) for $\phi \leq 1.2$ than **Uncons** ($\approx$0.75). This range of $\phi$ can be desired in practice: e.g., a platform must set $\phi \leq 1.1$ to guarantee the 80% rule is satisfied two groups. Thus, we observe that existing fair-ranking algorithms may achieve a worse RD than **Uncons**.

## 5.2 Simulation on real-world image data

In this simulation, given *non-gender labeled* images-search results and their utilities, our goal is to generate a high-utility and gender-balanced ranking.

**Data.** We use the Occupations dataset [15] which contains the top 100 Google Image results for 96 occupation-related queries. For each image, the data has its position in search results, gender (coded as male/female) of the individual depicted in the image, collected via MTurk. We use the (true) gender labels in the data to compute RD and to estimate $\widehat{P}$, but do not provide them to algorithms.

**Setup.** For each image $i$, with rank $r_i$, we define $w_i := \left(\log\left(1 + r_i\right)\right)^{-1}$. We say an occupation is gender-stereotypical if more than 80% of images for this occupation have the same gender label (41/96 occupations). An image is said to be stereotypical if it is in a gender-stereotypical occupation and its gender label is the majority label for its occupation. We define the socially-salient groups as the sets of stereotypical and non-stereotypical images in gender-stereotypical occupations.

**Estimating $\widehat{P}$.** After pre-processing, we use a CNN-based gender-classifier $f$ [59] to predict the (apparent) gender of the person depicted in each image. We calibrate the confidence scores output by $f$ by binning and use these to estimate $\widehat{P}$ (see Supplementary Material G for more details). We perform this calibration once and on all occupations and, then, use it for gender-stereotypical occupations. Because of this $\widehat{P}$ is miscalibrated (and hence, inaccurate). For instance, among samples $i$ for which $0.25 \leq \widehat{P}_{i,\text{male}} \leq 0.5$, more than 75% are labeled as 'man' (instead of some percentage between 25% and 50%). *This violates the assumption that $P$ is accurately known.*

**Results.** See Figure 2 for RD and utilities (NDCG) averaged over 1000 iterations. We observe that **NResilient** achieves the best RD ($\approx$0.81) and has a better RD-utility trade-off than the other baselines. In contrast, **CSV**, **SJ**, and **GAK**, achieve a worse RD ($\leq$0.77). **MC** achieves the worst RD ($\leq$0.70) and a worst RD-utility trade-off. In particular, **NResilient**'s RD-utility trade-off strictly dominates all baselines for RD $\geq$ 0.66. This value of RD can arise in practice: Figure 9 plots the RD vs $\phi$ for this simulation and shows that if $\phi \leq 1.1$ (as required to, e.g., guarantee satisfaction of the 80% rule), then all baselines have RD at least 0.66. We further evaluate the robustness of **NResilient** to varying levels of noise on the Occupations dataset in Supplementary Material G.3.2 and observe **NResilient** has a better or similar RD than each baseline at all noise levels.

## 5.3 Simulation on real-world name data

We consider gender and race (encoded as binary) as socially-salient attributes. Our goal is to ensure equal representation across the four disjoint groups formed by combinations of these: non-White non-men, White non-men, non-White men, and White men.

**Data.** We consider the chess ranking data [29] which has of 3,251 chess players. For each player, among other attributes, the data has their full-name, self-identified gender (coded as male/female),

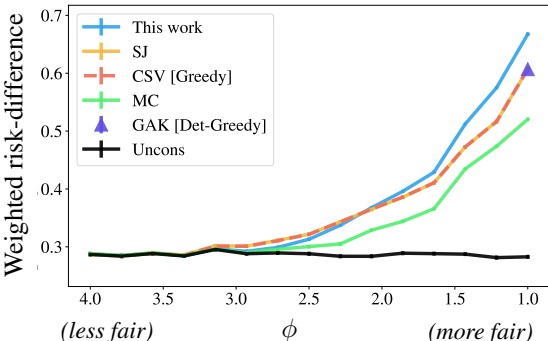

Figure 3: *Real-World Name Data: Multiple Attributes.* In this simulation, the goal is to ensure equal representation across four disjoint groups formed by combinations of two attributes (non-White non-men, White non-men, non-White men, and White men). We estimate $P$ by querying public APIs and libraries with names in the data. The $y$-axis plots RD and $x$-axis plots $\phi$. (*Note that the values decrease toward the right*). We observe that all algorithms have a better RD than **Uncons** and **NResilient** has the best RD compared to all other baselines. Error bars represent the error of the mean.

FIDE rating, and race (Asian, Black, Hispanic, White) collected via MTurk. We use the (true) gender and race labels in the data to evaluate RD, but do not provide them to algorithms.

**Setup.** We partition the races into White (81.66%) and non-White (18.34%). For each player $i$, we query Genderize and EthniColr[§] with $i$'s full-name to obtain the "probabilities" $p_f(i)$ and $p_{nw}(i)$ that player $i$ is labeled as a women and non-white respectively. We assume that these probabilities are correct and that the gender and race of players are drawn independently. Hence, e.g., we set the probability that $i$ is a non-white women as $\widehat{P}_{i,nw+f} = p_{nw}(i)p_f(i)$. Similarly, we set $\widehat{P}_{i,w+f} = (1 - p_{nw}(i))p_f(i)$, $\widehat{P}_{i,nw+m} = p_{nw}(i)(1 - p_f(i))$, and $\widehat{P}_{i,w+m} = (1 - p_{nw}(i))(1 - p_f(i))$.

Notably, we do not calibrate $\widehat{P}$ on this data. We verify that, like the previous simulation, $\widehat{P}$ is miscalibrated in this simulation. E.g., only 31% of the samples $i$ for which $\widehat{P}_{i,nw+m} > 0.75$ are labeled as 'Non-white man' (instead of 75%). *Hence, the assumption that $P$ is accurately known is violated in this simulation.* We expect calibration to improve **NResilient**'s performance.

*Results.* See Figure 3 for RD averaged over 500 iterations. We observe that all algorithms (**NResilient**, **CSV**, **GAK**, **SJ**, and **MC**) have better RD than **Uncons**. Among these, **NResilient** achieves the best RD ($\approx 0.67$), next **CSV**, **GAK**, and **SJ** obtain RD ($\approx 0.61$), and **MC** achieves RD ($\leq 0.53$). More specifically, for all $\phi \leq 1.75$, **NResilient** has a strictly better RD than all baselines (this range of $\phi$ subsumes, e.g., the range $\phi \leq 1.05$–required guarantee satisfaction of the 80% rule with four groups.) Further, in Figure 10, we observe that all algorithms have a similar fairness-utility trade-off.

## 6 Proof of Theorem 4.1

In this section we prove Theorem 4.1. Some of the details are deferred to Supplementary Material E.3 due to space constraints. The proof is divided into two propositions:

**Proposition 6.1.** *For any $\delta \in (0, 1]$, any ranking feasible for Prog. (7) satisfies $(c\gamma, \delta)$-constraint.*

**Proposition 6.2.** *For any $\delta \in (0, \frac{1}{2})$ and $c > 1$, any ranking satisfying the $((c - \sqrt{c})\gamma, \delta)$-constraint is feasible for Program (7).*

*Proof of Theorem 4.1.* Let $R^\star$ be the optimal solution of Program (7). Since $R^\star$ is feasible by definition, Proposition 6.1 implies that $R^\star$ satisfies the $(c\gamma, \delta)$-constraint. Pick any $R'$ that satisfies the $((c - \sqrt{c})\gamma, \delta)$-constraint. Proposition 6.2 implies that $R'$ is feasible for Program (7). Since $R^\star$ is an optimal solution of Program (7), $R^\star$'s utility is at least as large as the utility of $R'$. □

**Notation**. For each item $i$ and group $\ell$, let $Z_{i\ell} \in \{0, 1\}$ be the indicator random variable $Z_i := \mathbb{I}[G_\ell \ni i]$. By Definition 3.2, $\Pr[Z_{i\ell}] = P_{i\ell}$ and $Z_{i\ell}$ and $Z_{j\ell}$ are independent for any $i \neq j$. Given ranking $R \in \mathcal{R}$, group $\ell \in [p]$, and position $k \in [n]$, let $Z_\#(R, \ell, k)$ be the number of items from $G_\ell$ in the top $k$ positions of $R$ and let $P_\#(R, \ell, k) = \mathbb{E}[Z_\#(R, \ell, k)]$. From the above, we get:

$$P_\#(R, \ell, k) = \mathbb{E}[Z_\#(R, \ell, k)] = \sum_{i \in [m]} \sum_{j \in [k]} P_{i\ell} R_{ij}.$$

We use the following concentration result (proved in Supplementary Material E.2) in the proof.

**Lemma 6.3.** *For any position $k \in [n]$, group $\ell \in [p]$, parameters $\varepsilon \geq 0$ and $L, U \in \mathbb{R}$, and ranking $R \in \mathcal{R}$, where $R$ is possibly a random variable independent of $\{Z_{i\ell}\}_{i,\ell}$, if $P_\#(R, \ell, k) \leq U$ or*

---

[§]`gender-api.com` and `github.com/appeler/ethnicolr` respectively

$P_{\#}(R, \ell, k) \geq L$ then the following equations hold respectively $\Pr\left[Z_{\#}(R, \ell, k) < (1+\varepsilon)\,U\right] \geq 1 - e^{-\frac{U\varepsilon^2}{2+\varepsilon}}$ and $\Pr\left[Z_{\#}(R, \ell, k) > (1-\varepsilon)\,L\right] \geq 1 - e^{-\frac{L\varepsilon^2}{2(1-\varepsilon)}}$.

*Proof of Proposition 6.1.* Fix any $k$ and $\ell$. Let

$$\phi := 1 - \frac{1}{2\sqrt{c}}, \quad U' := U_{k\ell}\left(1 + \phi\gamma_k\right), \quad \text{and} \quad \zeta := \frac{(1-\phi)\gamma_k}{1+\phi\gamma_k}. \tag{9}$$

Here, $U'$ and $\zeta$ satisfy $U'(1+\zeta) = U_{k\ell}(1 + c\gamma_k)$. Fix any ranking $R$ that is feasible for Program (7). Since $R$ is feasible, it satisfies that

$$\forall \ell \in [p], \ k \in [n], \quad P_{\#}(R, \ell, k) \leq U_{\ell k}\left(1 + \phi\gamma_k\right). \tag{10}$$

Using $U'(1+\zeta) = U_{k\ell}(1 + c\gamma_k)$, Equation (10), and Lemma 6.3, we get that

$$\Pr\left[Z_{\#}(R, \ell, k) \geq U'(1+\zeta)\right] \quad \leq \quad e^{-\frac{2U'\zeta^2}{2+\zeta}} \overset{(9)}{=} e^{-\frac{(1-\phi)^2 c^2 \gamma_k^2 U_{k\ell}}{2+(1+\phi)c\gamma_k}} \overset{(\phi \leq 1)}{\leq} e^{-\frac{(1-\phi)^2 c^2 \gamma_k^2 U_{k\ell}}{2(1+c\gamma_k)}}. \tag{11}$$

*Fact* 6.4. For all $x, y \geq 0$, if $x \geq y + \sqrt{y}$, then $\frac{x^2}{1+x} \geq y$.

Using Fact 6.4 and Equation (6), we can show that for each $k$, $\frac{c^2\gamma_k^2}{1+c\gamma_k} \geq \frac{2}{(1-\phi)^2 U_{k\ell}} \cdot \log\frac{2np}{\delta}$. (This uses $\delta < \frac{1}{2}$ and $U_{k\ell}, n \geq 1$.) Substituting this in Equation (11) we get:

$$\Pr\left[Z_{\#}(R, \ell, k) \geq U_{\ell k}(1 + c\gamma_k)\right] \leq \frac{\delta}{2np}. \tag{12}$$

Taking the union bound over all positions $k$ and $\ell$, we get (as desired) that with probability at least $1 - \delta$, for all $k \in [n]$ and $\ell \in$, $Z_{\#}(R, \ell, k) \leq U_{\ell k}(1 + c\gamma_k)$. $\square$

*Proof of Proposition 6.2.* Let $\phi := 1 - \frac{1}{2\sqrt{c}}$. Towards a contradiction, suppose that $R'$ satisfies $((c - \sqrt{c})\gamma, \delta)$-constraint but is not feasible for Program (7). Then there exists $\ell$ and $k$ such that $P_{\#}(R', k, \ell) > U_{k\ell} \cdot (1 + \phi\gamma_k)$. Fix any $k$ and $\ell$ satisfying this. Let

$$b := 1 - \frac{1}{\sqrt{c}}, \quad L' := U_{k\ell}\left(1 + \phi\gamma_k\right) \quad \text{and} \quad \zeta := \frac{(1+b)\gamma_k}{1+\phi\gamma_k}. \tag{13}$$

It holds that $L'(1-\zeta) = U_{k\ell}(1 + b\gamma_k)$ and, hence, we get

$$\Pr\left[Z_{\#}(R', k, \ell) \leq L'(1-\zeta)\right] \overset{(13),\ \text{Lem.6.3}}{\leq} e^{-\frac{L'\zeta^2}{2(1-\zeta)}} \overset{(13)}{=} e^{\frac{-(c-b)^2 U_{k\ell}\gamma_k}{2(1+b)}} \leq e^{\frac{-U_{k\ell}c\gamma_k}{4(2\sqrt{c}-1)\sqrt{c}}} \overset{(c>0)}{=} e^{\frac{-U_{k\ell}\gamma_k}{8}}. \tag{14}$$

Since $\gamma_k \geq 8\log\frac{np}{\delta} \cdot \max_\ell \sqrt{\frac{1}{U_{k\ell}}}$, $\delta < \frac{1}{2}$, and $U \geq 1$, we have $\Pr\left[Z_{\#}(R', k, \ell) \leq U_{k\ell}\right] \leq \frac{\delta}{np} < 1 - \delta$. Since $R'$ satisfies $((c - \sqrt{c})\gamma, \delta)$-constraint we have a contradiction, hence $R'$ must be feasible. $\square$

# 7 Limitations and conclusion

Recent studies find that errors in socially-salient attributes can adversely affect fairness and utility of existing fair-ranking algorithms [29]. We consider a model of random and independent errors in socially-salient attributes and present a framework that can output rankings with high fairness and utility in this model. This framework works for a general class of fairness criteria, which involve multiple overlapping groups and upper bounds on the number of items that appear in the first $k$ positions from each group. We also show near-tightness of the framework's fairness guarantee. Empirically, on both synthetic and real-world datasets, we observe that, compared to baselines, our framework can achieve higher fairness-values and a similar or better fairness-utility trade-off for standard metrics.

Compared to existing fair-ranking frameworks, our framework does not need accurate socially-salient attributes, but assumes that errors in attributes are random and independent. When these assumptions do not hold, our framework may not satisfy its guarantees and a careful assessment of this on application-specific data would be important to avoid any (unintended) negative social impact.

Our work only addresses one aspect of how bias may show up in rankings, and more generally, on the web. For instance, while we consider a large class of fairness constraints, it does not capture some important notions such as the qualitative representation of different groups [38, 55]. It is important to take an holistic approach to mitigate bias and incorporate our work as a part of such a broader effort. Finally, our work adds to the line of works that develop fair decision-making algorithms robust to inaccuracies in data [42, 6, 54, 25, 67, 66, 48, 14].

**Acknowledgements.** This research was supported in part by NSF Awards CCF-2112665 and IIS-2045951, and an AWS MLRA Award.

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
