# Contents

## A  Further discussion on applicability of the noise model

The noise in Definition 3.2, arises in real-world settings where local differential privacy is ensured e.g., using the randomized response mechanism.

*Remark* A.1 (**Model's assumptions hold if attributes are perturbed by randomized response**). The randomized response mechanism flips each item's protected attribute to an incorrect value with some (public) probability $0 < \eta < \frac{1}{2}$, independent of all other items. Here, the independence assumption holds (by design) and $P$'s entries can be deduced from $\eta$. To see the latter concretely, consider two protected groups $G_1$ and $G_2$ ($p = 2$), and their noisy versions $N_1$ and $N_2$ corresponding to the "flipped" attributes. For any item $i \in N_1$,

$$P_{i1} = (1 - \eta) \cdot \frac{|G_1|}{|N_1|} \quad \text{and} \quad P_{i2} = 1 - P_{i1}.$$

For items in $N_2$, replace $P_{i1}, P_{i2}, G_1$, and $N_1$ with $P_{i2}, P_{i1}, G_2$, and $N_2$. When there are more than two groups ($p > 2$), then the randomized response mechanism publicly specifies the probability $\eta_{a,b}$ with which it flips protected attribute value $\ell = a$ to another value $\ell = b$ (for any $a, b \in [p]$). As in the binary case above, P's entries can be deduced from parameters $\{\eta_{a,b} : a, b \in [p]\}$.

Further, in other real-world settings such as image search and online recruiting, the entries of $P$ can be estimated using the confidence scores of classifiers or using auxiliary attributes. In more detail:

- If the protected attribute is skin tone, then a classifier $C$ can be used to predict if image $i$ contains a person with a dark skin tone. If $C$ has a calibrated confidence score $0 \le c(i) \le 1$ in this prediction, then $P_{i,\mathrm{darkskin-tone}} = c(i)$. See Figure 2 in Section 5 for results from a simulation that estimates $P$ in this fashion.
- If the protected attribute is race and individuals are uniformly drawn from the population, then for an individual $i$ with surname $S$ and zip-code $Z$, $P_{i,L} = f(Z, S)$, where $f(Z, S)$ is the fraction of individuals with surname $S$ in zip-code $Z$ who have the $L$-th race; which can be estimated using census data [23] (see Figure 3 in Section 5).

**Discussion on the noise model with disjoint groups vs. overlapping groups.**    For each item $i$ and group $G_\ell$ ($\ell \in [p]$), the noise model specifies the marginal probability that $i$ belongs to $G_\ell$:

$$P_{i\ell} := \Pr[G_\ell \ni i].$$

For any $i$, the model allows for any joint probability distribution over the events

$$(G_1 \ni i), (G_2 \ni i), \ldots, (G_p \ni i)$$

that is consistent with the above marginal probabilities. This allows the model to capture the setting where all groups are disjoint – by requiring the events

$$(G_1 \ni i), \ldots, (G_p \ni i)$$

to be mutually exclusive. It also allows the model to capture the cases where all or only some of the groups can overlap. For instance, the case where $G_1$ can overlap with $G_2$ but both $G_1$ and $G_2$ are disjoint from $G_3$ can be captured by requiring the events $(G_3 \ni i)$ to be mutually exclusive of the events $(G_1 \ni i)$ and $(G_2 \ni i)$. Importantly, we do not need additional information to capture these settings–it suffices to know the marginal probabilities specified by $P$.

## B  Other related work on fair decision making with inaccuracies in attributes

Several recent works develop fair algorithms for tasks different from ranking that are robust to inaccuracies in the socially-salient attributes [42, 6, 54, 25, 67, 48, 66, 14, 17, 25]. In particular, several works study classification and clustering [42, 6, 54, 25, 67, 48, 66, 14, 17, 25], and develop fair algorithms robust to inaccuracies in protected attributes. Many of these works consider the same random error model as us (or one of its variants) [42, 6, 67, 48, 14, 25], but some very recent works have also considered adversarial noises in protected attributes [67, 41, 17]. However, because the

underlying algorithmic tasks are fundamentally different from the variant of the ranking problem we study it is not clear how to adapt their approaches to our setting. [48] studies the problem of fair subset selection under the same noise model. In subset selection, given $m$ items the goal is to output an *unordered* subset of $n \leq m$ items with the highest utility. They develop an optimization framework outputs a subset satisfying the fairness constraint up to a small multiplicative error with high probability but leave the problem of ranking open. We compare against an adaptation of their approach, **MC**, to ranking in our empirical results.

## C  Existing fair-ranking algorithms with rounding is insufficient

Since existing fair-ranking algorithms require access to protected attributes, one way to use them under the above model is to imputed groups $\widehat{G}_1, \ldots, \widehat{G}_p$ using the specified probabilities. Then run these algorithms w.r.t. the imputed groups. To see an illustration, consider two groups $G_1$ and $G_2$. A natural imputation strategy is to use the Bayes optimal classifier, which assigns item $i$ to $\widehat{G}_1$ if and only if $P_{i1} > 0.5$ and has the lowest expected imputation error. This may be reasonable when the imputation error is negligible. However, on exploring this strategy with non-negligible imputation error, we find that the output rankings can violate equal representation significantly (see Proposition C.1). To gain some intuition consider an extreme case where all items in some set $S$, of size $n$, have $P_{i1} = 0.51$. The Bayes classifier assigns all items in $S$ to $\widehat{G}_1$, i.e.,

$$|S \cap \widehat{G}_1| = |S| .$$

However, with high probability,

$$|S \cap G_1| \approx 0.51 |S| .$$

Since $|S \cap G_1|$ and $|S \cap \widehat{G}_1|$ are far, a ranking that selects $n$ items from $S$ and satisfies the constraints for $\widehat{G}_1$ and $\widehat{G}_2$ but violate constraints with respect to the true groups. Proposition C.1 gives an example where this occurs.

Another imputation strategy, is independent rounding: it assigns each item $i$ to $\widehat{G}_1$ with probability $P_{i1}$ and otherwise to $\widehat{G}_1$. This addresses the issue with Bayes imputation, because, it has property that for any set $T$ of size $n$, $|T \cap G_1|$ are $|T \cap \widehat{G}_1|$ close with probability $1 - e^{\Theta(n)}$. However, when $m \gg n$, there are

$$\binom{m}{n} \gg e^n$$

sets of size $n$, and hence, with high probability, there exists a set $S$ of size $n$ for which $|S \cap \widehat{G}_1|$ and $|S \cap G_1|$ are arbitrarily far. In this case also, existing fair-ranking algorithms can output rankings which violate equal representation significantly. Proposition C.2 gives an example where this occurs.

**Proposition C.1** (**Imputing protected groups using the bayes optimal classifier is not sufficient**). *Let $R$ be any optimal solution to* (2) *with protected groups imputed using the Bayes optimal classifier for given $p$. There exists a matrix $P \in [0,1]^{m \times 2}$ such that $R$ does not satisfy the $(\varepsilon, \delta)$-equal representation constraint*

$$\text{for any} \quad \delta < \frac{1}{2} \quad \text{and} \quad \varepsilon \quad \text{s.t.} \quad \varepsilon_k < \frac{1}{20} \quad \text{for some} \quad k \geq 2.$$

**Proposition C.2.** *Let $R$ be a random variable denoting the optimal solution to the fair-ranking problem (Program* (2)*) for protected groups imputed using independent rounding with given $P \in [0,1]^{m \times 2}$. For every $\beta > 0$, there exists sufficiently large $n$ and $m$ and a matrix $P \in [0,1]^{m \times 2}$, such that, with probability at least $1 - \beta$ $R$ does not satisfy the $(\varepsilon, \delta)$-equal representation constraint*

$$\text{for any} \quad \delta < 1 - \beta \quad \text{and} \quad \varepsilon \in (0,1)^n.$$

### C.1  Proof of Proposition C.1

*Proof of Proposition C.1.* Pick any even $n \in \mathbb{N}$. Let $m := \frac{3n}{2}$. Let $\beta > 0$ be a small constant that we will fix later. We will divide the items into the following three types:

- Type A: For each $1 \leq i \leq \frac{n}{2}$ and $1 \leq j \leq n$,
$$P_{i1} := 0 = 1 - P_{i2} \text{ and } W_{ij} := 1.$$

- Type B: For each $\frac{n}{2} + 1 \leq i \leq n$ and $1 \leq j \leq n$,
$$P_{i1} := \frac{1}{2} + \beta = 1 - P_{i2} \text{ and } W_{ij} := 1.$$

- Type C: For each $n + 1 \leq i \leq \frac{3n}{2}$ and $1 \leq j \leq n$,
$$P_{i1} := 1 = 1 - P_{i2} \text{ and } W_{ij} := 0.$$

Let $\widehat{G}_1$ and $\widehat{G}_2$ be the groups imputed using maximum likelihood rounding. By construction, $\widehat{G}_1$ contains all items of Types A and B and no items of Type C, whereas $\widehat{G}_2$ contains all items of Type C and no items of Types A and B.

Let $R$ be an optimal solution of Program (2) with parameters $G_1 = \widehat{G}_1$ and $G_2 = \widehat{G}_2$. Since $W_{ij} \leq 1$ for all $i \in [m], j \in [n]$,
$$\langle R, W \rangle \leq n.$$
Because $R$ satisfies the equal representation constraints for two disjoint groups, for any even $k \in [n]$, $R$ places exactly $\frac{k}{2}$ items of Type A and $\frac{k}{2}$ items of Type B in the top $k$ positions. From $\widehat{G}_1$, $R$ only places items of Type A: If $R$ picks no items of Type C, then $\langle R, W \rangle = n$, whereas, if $R$ picks one or more items of Type C, then $\langle R, W \rangle \leq n - 1$, which is a contradiction since there is a ranking with utility $n$ that satisfies equal representation constraints (e.g., a ranking which places items of Type A and B in alternate positions).

Since all items of Type A are (always) in $\widehat{G}_2$, $R$ places at least $\frac{k}{2}$ items from $\widehat{G}_2$ in the first $k$ positions. We will show that with probability larger than $\frac{1}{2}$, at least $\frac{k}{20}$ of the $\frac{k}{2}$ items of Type B are in $\widehat{G}_2$. Thus, with probability larger than $\frac{1}{2}$, $R$ places more than $\frac{k}{2} \cdot \frac{11}{10}$ items from $\widehat{G}_2$ in the top-$k$ positions, and hence, $R$ does not satisfy the $(\varepsilon, \delta)$-equal representation constraint for any $\delta < \frac{1}{2}$ and $\varepsilon \in \left(0, \frac{1}{10}\right)^n$.

It remains to prove our claim. Select any $k \in \{2, 4, \ldots, n\}$. Let $i_1, i_2, \ldots, i_{k/2} \in [m]$ be the $n$ items of Type B that $R$ places in the first $k$ positions. Let $Z_{i_j} \in \{0, 1\}$ be the indicator random variable that $i_j \in \widehat{G}_2$. Thus, $Z_{i_1}, \ldots, Z_{i_{k/2}}$ are independent random variables, such that, for $j \in [k]$, $\Pr[Z_{i_j}] = 1 - P_{i_j} = \frac{1}{2} - \beta$. It follows that $\mathbb{E}[\sum_{j=1}^{k/2} Z_{i_j}] = \frac{k}{2}\left(\frac{1}{2} - \beta\right)$ and $\text{Var}[\sum_{j=1}^{k/2} Z_{i_j}] = \frac{k}{2}\left(\frac{1}{4} - \beta^2\right)$. Thus, using the Chebyshev's inequality on $\sum_{j=1}^{k/2} Z_{i_j}$,

$$\Pr\left[\left|\sum_{j=1}^{k/2} Z_{i_j} - \frac{k}{4}(1 - 2\beta)\right| > \frac{k}{8}\left(1 - 4\beta^2\right) \cdot \sqrt{2 + \beta}\right] \leq \frac{1}{2 + \beta}.$$

Thus,

$$\Pr\left[\sum_{j=1}^{k/2} Z_{i_j} < \frac{k}{4}(1 - 2\beta) - \frac{k}{8}\left(1 - 4\beta^2\right) \cdot \sqrt{2 + \beta}\right] \leq \frac{1}{2 + \beta}.$$

Since $\frac{k}{4}(1 - 2\beta) - \frac{k}{8}\left(1 - 4\beta^2\right) \cdot \sqrt{2 + \beta} = k\left(\frac{1}{4} - \frac{\sqrt{2}}{8}\right) + k \cdot O(\beta)$, for a sufficiently small $\beta > 0$,

$$\frac{k}{4}(1 - 2\beta) - \frac{k}{8}\left(1 - 4\beta^2\right) \cdot \sqrt{2 + \beta} > \frac{k}{20}.$$

Hence,

$$\Pr\left[\sum_{j=1}^{k/2} Z_{i_j} < \frac{k}{20}\right] \leq \frac{1}{2 + \beta} \overset{(\beta > 0)}{<} \frac{1}{2}. \tag{15}$$

$\square$

## C.2 Proof of Proposition C.2

*Proof of Proposition C.2.* Let $\phi > 0$ be a small constant that we will fix later. We will divide the items into the following two types:

- Type A: For each item $i$ of Type A

$$P_{i1} := \phi, \ P_{i2} := 1 - \phi \text{ and } W_{ij} := 1 \text{ for all } j \in [n].$$

- Type B: For each item $i$ of Type B

$$P_{i1} := 1, \ P_{i2} := 0 \text{ and } W_{ij} := 0 \text{ for all } j \in [n].$$

- Type B: For each item $i$ of Type C

$$P_{i1} := 0, \ P_{i2} := 1 \text{ and } W_{ij} := 0 \text{ for all } j \in [n].$$

Let there be $m_A := O\left(\log\left(\frac{n}{\beta}\right) \cdot \frac{n}{\log\left(\frac{1}{1-\phi}\right)}\right)$ items of Type A, $m_B := n$ items of Type B, and $m_C := n$ items of Type C.

Note that a ranking which ranks items of Type B and Type C alternately, satisfies the equal representation constraints with probability 1. So in this instance, there exists a ranking which satisfies $(\delta, \varepsilon)$-equal representation. However, we will show that $R$ does not satisfy $(\delta, \varepsilon)$-equal representation with probability at least $1 - \beta$.

Let $\widehat{G}_1$ and $\widehat{G}_2$ be the groups imputed by independent rounding. Let $\mathscr{E}$ be the event that $\widehat{G}_1$ contains at least $n$ items of Type A and $\mathscr{F}$ be the event that $\widehat{G}_2$ contains at least $n$ items of Type A. Both $\mathscr{E}$ and $\mathscr{F}$ occur with probability at most $O(\beta)$. To see this, divide the items of Type A into $n$ groups of equal size. From each group, at least one item is selected in $\widehat{G}_1$ and $\widehat{G}_2$ with probabilities at least $1 - (1 - \phi)^{\frac{m_A}{n}}$ and $1 - (\phi)^{\frac{m_A}{n}}$ respectively. Taking a union bound over all groups and substituting $m_A$, we get

$$\Pr[\mathscr{E}] \geq 1 - \beta \quad \text{and} \Pr[\mathscr{F}] \geq 1 - \beta.$$

Since only items of Type A have a nonzero contribution to the utility of a ranking and because there are at least $n$ items of Type A in each imputed group, it follows that $R$ only selects items of Type A. Now, the claim follows because, for small $\phi$, most items of Type A belong to $G_1$.

Suppose $\mathscr{E}$ and $\mathscr{F}$ happen and, hence, $R$ only selects items of Type A. Let $Z_j$ be the indicator random variable that the item in the $j$-th position of $R$ is in $G_1$. We have that $\Pr[Z_j] = \phi$. Therefore, $\text{Var}[\sum_{j=1}^{n} Z_j] = n\phi(1 - \phi)$. Thus, using the Chebyshev's inequality we have

$$\Pr\left[\left|\sum_{j=1}^{n} Z_j - n\phi\right| \geq \frac{n\varepsilon_n}{4}\right] \leq \frac{4n\phi(1 - \phi)}{n^2 \varepsilon_n^2}.$$

Hence, for $\phi = \Theta(\varepsilon_n^2 \beta)$, we have that

$$\Pr\left[\sum_{j=1}^{n} Z_j \leq \frac{n\varepsilon_n}{2}\right] \geq 1 - \beta.$$

The result follows since whenever $\sum_{j=1}^{n} Z_j \leq \frac{n\varepsilon}{2}$, $R$ violates the equal representation constraint at the $n$-th position by a multiplicative factor larger than $1 + \varepsilon_n$.

$\square$

# D  Proofs of Theorems 4.2 and 4.3

## D.1  Proof of Theorem 4.2

We consider the family of matrices $U \in \mathbb{R}^{n \times p}$ that satisfy the following condition: For each position $k \in [n]$, there exists an attribute $\ell$ such that

$$U_{k\ell} \leq \frac{k}{4}.$$

Notably, equal representation constraints satisfy this condition for any $p \geq 4$. We will use Fact D.1 to prove Theorem 4.2.

*Fact* D.1 (Theorem 2 in [53]). For all $p \in (0, \frac{1}{4}]$, $0 \leq \varepsilon \leq \frac{1}{p}(1 - p)$, and $s \in \mathbb{N}$ independent 0/1 random variables $Z_1, Z_2, \ldots, Z_s \in \{0, 1\}$, such that for all $i \in [s]$, $\Pr[Z_i = 1] = p$,

$$\Pr\left[\sum_{i \in [s]} Z_i \geq (1 + \varepsilon)ps\right] \geq \frac{1}{4} \exp\left(-2\varepsilon^2 ps\right).$$

*Proof of Theorem 4.2.* Fix the $k$ to the value specified in the theorem. Let $\ell \in [n]$, be any attribute such that $U_{k\ell} \leq \frac{k}{4}$. Such a $\ell$ exists because of the family of constraints we chose. Without loss of generality suppose $\ell \neq 1$. Fix any $n, m \geq k$. For each item $i \in [m]$, set

$$P_{i\ell} := \frac{U_{k\ell}}{k} \quad \text{and} \quad P_{i1} := 1 - \frac{U_{k1}}{k} \tag{16}$$

Further, for all $k \in [p]$, $k \neq p$ and $k \neq 1$, let $P_{ik} := 0$.

Suppose, toward a contradiction, that there is a ranking $R \in \mathcal{R}$ that satisfies the $(\varepsilon, \delta)$-constraint. $R$ must satisfy the following equation:

$$\Pr\left[Z_{\#}(R, k, \ell) \leq U_{k\ell} \cdot (1 + \varepsilon_k)\right] \geq 1 - \delta. \tag{17}$$

For each position $j \in [n]$, let $Z_j \in \{0, 1\}$ be the indicator random variable that the item placed in the $j$-th place in the ranking $R$ is in the protected group $G_\ell$. From Equation (16) and Definition 3.2, it follows that:

$$\forall j \in [n], \quad \Pr[Z_j] = \frac{U_{k\ell}}{k}, \tag{18}$$

$$\forall u, v \in [n], \quad \text{s.t.}, u \neq v, \quad Z_u \text{ and } Z_v \text{ are independent.} \tag{19}$$

Using linearity of expectation and Equation (18), we get that:

$$\Pr\left[Z_{\#}(R, k, \ell) \leq (1 + \varepsilon_k) \cdot U_{k\ell}\right] = \Pr\left[\sum_{j \in [k]} Z_j \geq (1 + \varepsilon_k) \cdot \mathbb{E}\left[\sum_{j=1}^{k} Z_j\right]\right]. \tag{20}$$

Since $0 \leq \varepsilon_k \leq 1$ and $\frac{1}{k} \mathbb{E}\left[\sum_{j=1}^{k} Z_j\right] \leq \frac{1}{4}$, we can use Fact D.1 with $\varepsilon := \varepsilon_k$, $p := \frac{1}{k} \mathbb{E}\left[\sum_{j=1}^{k} Z_j\right] \leq \frac{1}{4}$, $s := k$, and for all $j \in [n]$, $Z_j = Z_j$. Using this, we get that

$$\Pr\left[\sum_{j \in [k]} Z_j \geq (1 + \varepsilon_k) \cdot \mathbb{E}\left[\sum_{j=1}^{k} Z_j\right]\right] \leq 1 - \frac{1}{4} \exp\left(-2\varepsilon_k^2 \cdot \mathbb{E}\left[\sum_{j=1}^{k} Z_j\right]\right)$$

$$\leq 1 - \frac{1}{4} \exp\left(-2\varepsilon_k^2 U_{k\ell}\right).$$

$$\text{(Using Equation (18))} \tag{21}$$

Chaining Equations (17), (20), and (21), we get that

$$1 - \frac{1}{4} \exp\left(-2\varepsilon_k^2 U_{k\ell}\right) \geq 1 - \delta.$$

Hence,

$$\varepsilon_k \geq \sqrt{\frac{1}{2U_{k\ell}} \log \frac{1}{4\delta}}.$$

This is a contradiction since $\varepsilon_k$ is specified to be less than $\sqrt{\frac{1}{2U_{k\ell}} \log \frac{1}{4\delta}}$. Thus, no ranking $R$ satisfies the $(\varepsilon, \delta)$-constraint for any $U$ in the chosen family chosen. $\qquad\square$

## D.2  Proof of Theorem 4.3

In this section, we prove Theorem 4.3. Our algorithm uses the dependent-rounding algorithm of [19] as a subroutine.

***Remark.* Desirable properties and potential approaches for rounding.** *At a high level, the goal of this dependent-rounding algorithm is the following: Given a feasible solution $R_c$ of the standard linear programming relaxation of Program (7) output a ranking $R$ such that for any matrix $A$ with nonnegative entries, $\langle R, A \rangle$ is approximately equal to $\langle R_c, A \rangle$. This property guarantees that, with high probability, $R$ approximately satisfies the fairness constraints and has a similar utility as $R_c$.*

*A naive approach to achieve this property is to do independent rounding: For each $i$ and $j$, set $R_{ij} = 1$ with probability $(R_c)_{ij}$. The desired concentration property then follows from, e.g., the Chernoff bound. However, the resulting $R$ may not be a valid ranking because it could set $R_{ij} = R_{ik} = 1$ for $j \neq k$, hence requiring $i$ to appear at two different positions (which is not possible). Similarly, it could also place more than one items at one position (which also violates the constraints).*

*Another approach is (1) to express $R_c$ as a convex combination of rankings $\sum_i \alpha_i R_i$ ($\alpha_i \geq 0$) (e.g., using the Birkhoff von Neumann decomposition) and (2) set $R := R_i$ with probability $\propto \alpha_i$. Since each $R_i$ is a ranking this guarantees that $R$ is a ranking, but it may violate fairness constraints significantly. For example, consider the fractional assignment in which the $k$-th best female (respectively male) appears in the $k$-th position with weight 0.5 for all positions $k$. This can be decomposed into two rankings: 1) females are ranked in decreasing order of utility, and 2) males are ranked in decreasing order of utility. The fractional solution satisfies equal representation, but both rankings violate equal representation significantly.*

*The dependent-rounding algorithm of [19], which we use, also expresses $R_c$ as a convex combination of rankings $\sum_i \alpha_i R_i$ ($\alpha_i \geq 0$). But it does not output $R_i$ for any $i$. Instead, it initially, sets $R := R_1$. Then it iteratively "merges" $R$ with $R_2$, then $R_3$, and so on.*

[19]'s algorithm satisfies the following guarantees.

**Theorem D.2 (Theorem 1.1 from [19]).** *Let $P \subseteq [0,1]^N$ be either a matroid intersection polytope or a (non-bipartite graph) matching polytope. For any fixed $0 < \alpha \leq \frac{1}{2}$, there is an efficient randomized rounding procedure, such that given a (fractional) point $R_F \in P$, it outputs a random feasible solution $R$ corresponding to a (integer) vertex of $P$ such that $\mathbb{E}[1_R] = (1 - \alpha) \cdot R_F$. In addition, for any linear function $w(R) := \sum_{i \in R} w_i$, where $w_i \in [0,1]$ it holds that*

1. *for any $\delta \in [0,1]$ and $\mu \leq \mathbb{E}[1_R]$, $\Pr[w(R) \leq (1-\delta)\mu] \leq \exp\left(-\frac{1}{20} \cdot \mu\alpha\delta^2\right)$,*

2. *for any $\delta \in [0,1]$ and $\mu \geq \mathbb{E}[1_R]$, $\Pr[w(R) \geq (1-\delta)\mu] \leq \exp\left(-\frac{1}{20} \cdot \mu\alpha\delta^2\right)$,*

3. *for any $\Delta \geq 1$ and $\mu \geq \mathbb{E}[1_R]$, $\Pr[w(R) \geq \mu(1+\Delta)] \leq \exp\left(-\frac{1}{20} \cdot \mu\alpha(2\Delta - 1)\right)$.*

*The algorithm runs in time polynomial in the size of the ground set, $N$, and $\frac{1}{\alpha}$, and makes at most $\mathrm{poly}(N, d)$ calls to the independence oracles for the underlying matroids.*

We claim that the following algorithm satisfies the claim in Theorem 4.3

For each item $i \in [m]$ and protected attribute $\ell \in [p]$, let $Z_{i\ell} \in \{0,1\}$ be the indicator random variable that the $i$-th item is in the $\ell$-th protected group, i.e., if $i \in G_\ell$, then $Z_i = 1$, and other $Z_i = 0$. Using Definition 3.2, it follows that:

$$\forall i \in [m], \ \ell \in [p], \quad \Pr[Z_{i\ell}] = P_{i\ell}, \tag{22}$$
$$\forall i, j \in [m], \ \ell \in [p], \quad \text{s.t.}, i \neq j, \quad Z_{i\ell} \text{ and } Z_{j\ell} \text{ are independent.} \tag{23}$$

**Algorithm 1** Algorithm from Theorem 4.3

---

**Input:** Matrices $P \in [0, 1]^{m \times p}, W \in \mathbb{R}_{\geq 0}^{m \times n}, U \in \mathbb{R}^{n \times p}$

**Parameters:** Constant $d > 2$ and $c > 1$, a failure probability $\delta \in (0, 1]$, and for each $k \in [n]$, a relaxation parameter

$$\gamma_k := 12 \cdot \log\left(\frac{2np}{\delta}\right) \cdot \max_{\ell \in [p]} \sqrt{\frac{1}{U_{k\ell}}}.$$

1. **Initialize** $R_F \leftarrow$ Solve the linear-programming relaxation of Program (7) with the specified inputs
2. **Round** $R \leftarrow$ Run [19]'s rounding algorithm with input $\alpha := \frac{1}{d}$ and $P := \text{conv}(\mathcal{R})$
3. **Return** $R$

---

To simplify the notation, given a ranking $R \in \mathcal{R}$, a protected attribute $\ell \in [p]$, and a position $k \in [n]$, let $Z_\#(R, \ell, k) \in \mathbb{Z}$ be the random variable equal to the number of items from $G_\ell$ in the top $k$ positions of $R$ and let $P_\#(R, \ell, k) \in \mathbb{R}$ be the expectation of $Z_\#(R, \ell, k)$, i.e.,

$$Z_\#(R, \ell, k) := \sum_{i \in [m]} \sum_{j \in [k]} Z_{i\ell} R_{ij} \quad \text{and} \quad P_\#(R, \ell, k) := \mathbb{E}\left[Z_\#(R, \ell, k)\right].$$

Using Equation (22) and linearity of expectation it follows that

$$P_\#(R, \ell, k) = \sum_{i \in [m]} \sum_{j \in [k]} P_{i\ell} R_{ij}.$$

*Proof.*

**Running time.** The Step 1 of Algorithm 1 runs in polynomial time when implemented with any polynomial-time linear programming solver. Observe that $\mathcal{R}$ corresponds to the bipartite matching polytope, whose bi-partitions have size $n$ and $m$ respectively. Since the bipartite matching polytope is a matroid intersection polytope, we can use Theorem D.2. The independence oracle for this polytope can be implemented in $\text{poly}(m)$ time, e.g., using the Birkhoff–von Neumann theorem. Finally, since $\alpha = \frac{1}{d}$ and $N = O(m^2)$, it follows that Step 2 of Algorithm 1 runs in polynomial time in $d$ and the bit complexity of the input (which is at least $m$).

Let

$$\phi := \frac{2\sqrt{c} - 1}{2\sqrt{c}}.$$

Let $R_F$ and $R$ be the rankings from Steps 1 and 2 of Algorithm 1. From Theorem D.2, we have that $\mathbb{E}[1_R] = (1 - \alpha) \cdot R_F$. Hence, for any weights $V \in \mathbb{R}^{n \times m}$, it holds that

$$\mathbb{E}\left[\langle R, V \rangle\right] = (1 - \alpha) \cdot \langle R_F, V \rangle. \tag{24}$$

Fix any position $k \in [n]$ and group $\ell \in [p]$. Since $\ell$, $k$, and $R$ are fixed, we use $Z_\#(R)$ and $Z_\#(R')$ and $P_\#$ to denote $Z_\#(R, \ell, k)$ and $P_\#(R, \ell, k)$ respectively.

**Utility guarantee.** Let $R^\star$ be the solution of Program (7) for $c = d$. Let $V := \langle W, R^\star \rangle$. Let $0 \leq \Delta \leq V$ be a parameter. Since $R_F$ is a solution of the LP-relaxation of Program (7) and $R^\star$ is a solution of Program (7), $R_F$'s utility is at least as large as the utility of $R^\star$. From this it follows that

$$\Pr\left[\langle W, R \rangle \leq \langle W, R^\star \rangle \cdot (1 - \alpha) - \Delta\right] \leq \Pr\left[\langle W, R \rangle \leq \langle W, R_F \rangle \cdot (1 - \alpha) - \Delta\right]. \tag{25}$$

Since $W \in [0, 1]^{m \times n}$, we can use Theorem D.2 with $a = W$. Using this we get can upper bound the RHS of the above equation.

$$\Pr\left[\langle W, R \rangle \leq \langle W, R_F \rangle \cdot (1 - \alpha) - \Delta\right] = \Pr\left[\langle W, R \rangle \leq \mathbb{E}\left[\langle W, R \rangle\right] - \Delta\right] \quad \text{(Using Equation (24))}$$

$$\leq \exp\left(-\frac{\alpha}{20} \cdot \frac{\Delta^2}{\langle W, R_F \rangle \cdot (1 - \alpha)}\right).$$

Let $\Delta := \sqrt{\frac{20}{\alpha} \cdot \langle W, R_F \rangle \cdot (1 - \alpha) \cdot \log\left(\frac{2np}{\delta}\right)}$. Substituting the value of $\Delta$ in the above equation, we have:

$$\Pr\left[\langle W, R \rangle \leq \mathbb{E}\left[\langle W, R \rangle\right] - \Delta\right] \leq \frac{\delta}{2np}. \tag{26}$$

Chaining the inequalities in Equations (25) and (26)

$$\Pr\left[\langle W, R \rangle \leq \langle W, R^\star \rangle \cdot (1 - \alpha) - \Delta\right] \leq \frac{\delta}{2n}.$$

Since each entry of $W$ is at most 1 and $\sum_{i,j} (R_F)_{ij} = n$, it follows that $\langle W, R_F \rangle \leq n$. Using this and that $\alpha = \frac{1}{d}$,

$$\Delta = O\left(\sqrt{dn \cdot \log \frac{2np}{\delta}}\right).$$

Thus, the utility guarantee follows.

**Fairness guarantee.** Since $R_F$ is feasible for the LP-relaxation of Program (7), it holds that

$$P_\#(R_F) \leq U_{k\ell}(1 + \phi\gamma_k). \tag{27}$$

Let $\varepsilon > 0$ be some constant such that

$$\varepsilon \geq \phi\gamma_k. \tag{28}$$

We divide the analysis into two cases depending on the value of $\varepsilon$.

**Case A ($P_\#(R) \geq \frac{1}{2} U_{k\ell}(1 + \varepsilon)$):** Since $P_\#(R) \geq \frac{1}{2} \cdot U_{k\ell}(1 + \varepsilon)$, we have that

$$\frac{U(1 + \varepsilon) - P_\#(R)}{P_\#(R)} \leq 1. \tag{29}$$

We have that

$$\Pr\left[Z_\#(R) > U_{k\ell}(1 + \varepsilon)\right] = \Pr\left[Z_\#(R) > P_\#(R) \cdot \left(1 + \frac{U_{k\ell}(1 + \varepsilon) - P_\#(R)}{P_\#(R)}\right)\right]$$

From Equation (24) it follows that $P_\#(R) = P_\#(R_F)(1 - \alpha)$. Then from Equations (27) and (28) we have that $P_\#(R) \leq U_{k\ell}(1 + \varepsilon)$. Hence, $\frac{U_{k\ell}(1+\varepsilon) - P_\#(R)}{P_\#(R)} \geq 0$. Further, from Equation (29) $\frac{U_{k\ell}(1+\varepsilon) - P_\#(R)}{P_\#(R)} \leq 0$. Hence, we can use the second statement of Theorem D.2. Using this we get

$$\leq \exp\left(-\frac{\alpha}{20} \cdot P_\#(R) \cdot \left(\frac{U_{k\ell}(1 + \varepsilon) - P_\#(R)}{P_\#(R)}\right)^2\right)$$

$$\leq \exp\left(-\frac{\alpha}{20} \cdot P_\#(R_F) \cdot \left(\frac{U_{k\ell}(1 + \varepsilon) - P_\#(R_F)}{P_\#(R_F)}\right)^2\right)$$
$$\text{(Fact E.2 and that } P_\#(R) \leq P_\#(R_F))$$

$$\leq \exp\left(-\frac{\alpha}{20} \cdot U_{k\ell} \cdot \frac{(\varepsilon - \phi\gamma_k)^2}{1 + \phi\gamma_k}\right).$$
$$\text{(Fact E.2 and Equation (27) )} \quad (30)$$

**Case B ($P_\#(R) < \frac{1}{2} U_{k\ell}(1 + \varepsilon)$):** Since $P_\#(R) < \frac{1}{2} \cdot U_{k\ell}(1 + \varepsilon)$, we have that

$$\frac{U_{k\ell}(1 + \varepsilon) - P_\#(R)}{P_\#(R)} \geq 1. \tag{31}$$

We have that

$$\Pr\left[Z_\#(R) > U_{k\ell}(1+\varepsilon)\right] = \Pr\left[Z_\#(R) > P_\#(R) \cdot \left(1 + \frac{U_{k\ell}(1+\varepsilon) - P_\#(R)}{P_\#(R)}\right)\right]$$

$$\leq \exp\left(-\frac{\alpha}{20} \cdot P_\#(R) \cdot \left(2 \cdot \frac{U_{k\ell}(1+\varepsilon) - P_\#(R)}{P_\#(R)} - 1\right)\right)$$

(Using third statement in Theorem D.2 and that Equation (31))

$$= \exp\left(-\frac{\alpha}{20} \cdot (2U_{k\ell}(1+\varepsilon) - 3P_\#(R))\right)$$

$$\leq \exp\left(-\frac{\alpha}{40} \cdot U_{k\ell}(1+\varepsilon)\right).$$

(Using that $P_\#(R) < \frac{1}{2} \cdot U_{k\ell}(1+\varepsilon)$) (32)

Combining Equations (30) and (32) we get that

$$\Pr\left[Z_\#(R) > U(1+\varepsilon)\right] \leq \max\left\{\exp\left(-\frac{\alpha}{20} \cdot U_{k\ell}\frac{(\varepsilon - \phi\gamma_k)^2}{1 + \phi\gamma_k}\right), \exp\left(-\frac{\alpha}{40} \cdot U_{k\ell}(1+\varepsilon)\right)\right\}.$$

(33)

Let

$$\varepsilon := \frac{40}{\alpha} \cdot \gamma_k. \tag{34}$$

We claim that for this value of $\varepsilon$, it holds that

$$\Pr\left[Z_\#(R) > U_{k\ell}(1+\varepsilon)\right] \leq \frac{\delta}{2n}. \tag{35}$$

Now by taking a union bound over bound over all $\ell \in [n]$ and using that $\alpha := \frac{1}{d}$, it follows that $R$ satisfies the fairness guarantee with probability at least $\frac{\delta}{2n}$.

We can upper bound the second term in Equation (33), as follows

$$\exp\left(-\frac{\alpha}{40} \cdot U_{k\ell}(1+\varepsilon)\right) \leq \exp\left(-\frac{\alpha}{40} \cdot U_{k\ell} \cdot \varepsilon\right)$$

$$\leq \exp\left(-U_{k\ell} \cdot \gamma_k\right)$$

$$\leq \frac{\delta}{np}.$$

(Using that $\gamma_k \geq \frac{1}{U_{k\ell}} \cdot \log\frac{2np}{\delta}$; which follows from Equation (6), $U_{k\ell} \geq 1$, and $\log\frac{2np}{\delta} \geq 1$)

To upper bound the first term in Equation (33), we use Fact D.3.

*Fact* D.3. For all $x, y \geq 0$, if $x \geq y + \sqrt{y}$, then $\frac{x^2}{1+x} \geq y$.

*Proof.* Since $1 + x > 0$, $\frac{x^2}{1+x} \geq y$ holds if and only if $x^2 - xy - y \geq 0$. The roots of the quadratic $f(x) := x^2 - xy - y$ are

$$\frac{y}{2} - \sqrt{\frac{y^2}{4} + y} \quad \text{and} \quad \frac{y}{2} + \sqrt{\frac{y^2}{4} + y}.$$

If $x$ is larger than both roots, then $f(x) \geq 0$ and, hence, $\frac{x^2}{1+x} \geq y$. It follows that $x \geq \frac{y}{2} + \sqrt{\frac{y^2}{4} + y}$ suffices. Then using that for all $a, b \geq 0$, $\sqrt{a} + \sqrt{b} \geq \sqrt{a+b}$, we get that

$$y + \sqrt{y} \geq \frac{y}{2} + \sqrt{\frac{y^2}{4} + y}.$$

Thus, it suffices $x \geq y + \sqrt{y}$ implies that $\frac{x^2}{1+x} \geq y$. $\square$

We have

$$\frac{(\varepsilon - \phi\gamma_k)^2}{1 + \phi\gamma_k} \geq \left(\frac{39}{\alpha}\right)^2 \cdot \frac{\gamma_k^2}{1 + \phi\gamma_k} \qquad \text{(Using that } 0 \leq \phi \leq 1, \, \alpha \leq \tfrac{1}{2}, \text{ and Equation (34))}$$

$$\geq \left(\frac{39}{\alpha}\right)^2 \cdot \frac{\gamma_k^2}{1 + \gamma_k}. \qquad \text{(Using that } 0 < \phi \leq 1)$$

To proof Equation (35), it suffices to prove that

$$\frac{\gamma_k^2}{1 + \gamma_k} \geq \frac{1}{U_{k\ell}} \cdot \log\left(\frac{n+2}{\delta}\right). \tag{36}$$

Further, Fact D.3 implies that to prove Equation (36) it suffices to prove that

$$\gamma_k \geq y + \sqrt{y},$$

where $y := \frac{1}{U_{k\ell}} \cdot \log \frac{n+2}{\delta}$. To prove this, observe that

$$\log \frac{np}{\delta} \cdot \frac{1}{U_{k\ell}} \leq \log \frac{np}{\delta} \cdot \sqrt{\frac{1}{U_{k\ell}}}, \qquad \text{(Using that } U_{k\ell} \geq 1)$$

$$\sqrt{\log \frac{np}{\delta} \cdot \frac{1}{U_{k\ell}}} \leq \log \frac{np}{\delta} \cdot \sqrt{\frac{1}{U_{k\ell}}}. \qquad \text{(Using that } \log \frac{np}{\delta} \geq \tfrac{1}{2} \text{ as } n \geq 1 \text{ and } \delta \leq \tfrac{1}{2})$$

Hence, Equation (36) follows from Equation (6). □

# E  Proofs of additional theoretical results

## E.1  Proof of Proposition 3.3

*Proof of Proposition 3.3.* Suppose $R$ is deterministic. Suppose it places items $i, j \in [m]$ on the first and second position respectively. With probability $p_i \cdot p_j = \frac{1}{4}$, both $i$ and $j$ belong to $G_1$, and with probability $p_i \cdot p_j = \frac{1}{4}$ both $i$ and $j$ belong to $G_2$. Thus, at least one of these events occurs with probability $\frac{1}{2}$. If either of these events hold, then $R$ violates the equal representation constraint on the top-2 positions by a multiplicative factor of 2. The last two statements imply that $R$ violates $(\rho, \delta)$-equal representation for any $\rho < 1$ and $\delta < \frac{1}{2}$.

If $R$ is a random variable, then any draw $R'$ of $R$ is a deterministic ranking, and hence, by the above argument $R'$ violates the equal representation constraint on the top-2 positions by a multiplicative factor of 2 with a probability $\frac{1}{2}$ (over the randomness in $G_1$ and $G_2$). Since this holds for all draws of $R$ and $R$ is independent of $G_1$ and $G_2$, it follows that $R$ violates the equal representation constraint on the top-2 positions by a multiplicative factor of 2 with a probability $\frac{1}{2}$ (over the randomness in $G_1$ and $G_2$, and $R$). Thus, $R$ does not satisfy $(\rho, \delta)$-equal representation for any $\rho < 1$ and $\delta < \frac{1}{2}$. □

## E.2  Proof of Lemma 6.3

In this section, we prove certain concentration inequalities which are used in the proof of Theorem 4.1. We divide the proof of Lemma 6.3 into two parts: Lemmas E.1 and E.6

For each item $i \in [m]$ and protected attribute $\ell \in [p]$, let $Z_{i\ell} \in \{0, 1\}$ be the indicator random variable that the $i$-th item is in the $\ell$-th protected group, i.e., if $i \in G_\ell$, then $Z_i = 1$, and other $Z_i = 0$. Using Definition 3.2, it follows that:

$$\forall i \in [m], \, \ell \in [p], \quad \Pr[Z_{i\ell}] = P_{i\ell}, \tag{37}$$

$$\forall i, j \in [m], \, \ell \in [p], \quad \text{s.t.,} \, i \neq j, \quad Z_{i\ell} \text{ and } Z_{j\ell} \text{ are independent.} \tag{38}$$

To simplify the notation, given a ranking $R \in \mathcal{R}$, a protected attribute $\ell \in [p]$, and a position $k \in [n]$, let $Z_\#(R, \ell, k) \in \mathbb{Z}$ be the random variable equal to the number of items from $G_\ell$ in the top $k$ positions of $R$ and let $P_\#(R, \ell, k) \in \mathbb{R}$ be the expectation of $Z_\#(R, \ell, k)$, i.e.,

$$Z_\#(R, \ell, k) := \sum_{i \in [m]} \sum_{j \in [k]} Z_{i\ell} R_{ij} \quad \text{and} \quad P_\#(R, \ell, k) := \mathbb{E}\left[Z_\#(R, \ell, k)\right].$$

Using Equation (37) and linearity of expectation it follows that

$$P_\#(R, \ell, k) = \sum_{i \in [m]} \sum_{j \in [k]} P_{i\ell} R_{ij}.$$

**Lemma E.1.** *For any position $k \in [n]$, attribute $\ell \in [p]$, parameters $\varepsilon \geq 0$ and $L \in \mathbb{R}$, and ranking $R \in \mathcal{R}$, where $R$ is possibly a random variable and is independent of $\{Z_{i\ell}\}_{i,\ell}$, if $P_\#(R, \ell, k) \geq L$ then with probability at least $1 - \exp\left(-\frac{L\varepsilon^2}{2(1-\varepsilon)}\right)$, it holds that $Z_\#(R, \ell, k) > L(1 - \varepsilon)$.*

*Proof.* Since $\ell$, $k$, and $R$ are fixed, we use $Z_\#$ and $P_\#$ to denote $Z_\#(R, \ell, k)$ and $P_\#(R, \ell, k)$ respectively.

Since $R$ and $\{Z_{i\ell}\}_{i,\ell}$ are independent, we can bound the required probability as follows

$$
\begin{aligned}
\Pr\left[Z_\# \leq L(1-\varepsilon)\right] &= \Pr\left[Z_\# \leq P_\# \cdot \left(1 - \frac{P_\# - L(1-\varepsilon)}{P_\#}\right)\right] \\
&\leq \exp\left(-\frac{P_\#}{2} \cdot \left(\frac{P_\# - L(1-\varepsilon)}{P_\#}\right)^2\right) \quad \text{(Chernoff bound, see [52])} \\
&= \exp\left(-\frac{1}{2} \cdot \frac{(P_\# - L(1-\varepsilon))^2}{P_\#}\right). \tag{39}
\end{aligned}
$$

To bound the right-hand side of Equation (39), we will use the following fact.

*Fact* E.2. For all $L, \varepsilon > 0$, $\frac{(x - L(1-\varepsilon))^2}{x}$ attains its minima at $L$ over the domain $[L, \infty)$.

Since $P_\# \geq L$, from Fact E.2 it follows that the right-hand side of Equation (39) attains its maxima at $P_\# = L$. Substituting $P_\# = L$ in Equation (39), we get:

$$\Pr\left[Z_\# \leq L(1-\varepsilon)\right] \leq \exp\left(-\frac{1}{2} \cdot \frac{(L\varepsilon)^2}{L(1-\varepsilon)}\right) = \exp\left(\frac{-L\varepsilon^2}{2(1-\varepsilon)}\right).$$

$\square$

**Lemma E.3.** *For any position $k \in [n]$, attribute $\ell \in [p]$, parameters $\varepsilon \geq 0$ and $U \in \mathbb{R}$, and ranking $R \in \mathcal{R}$, where $R$ is possibly a random variable and is independent of $\{Z_{i\ell}\}_{i,\ell}$, if $R$ satisfies that $P_\#(R, \ell, k) \leq U$ then with probability at least $1 - \exp\left(-\frac{U\varepsilon^2}{2+\varepsilon}\right)$, it holds that $Z_\#(R, \ell, k) < (1 + \varepsilon) \cdot U$.*

*Proof.* Since $\ell$, $k$, and $R$ are fixed, we use $Z_\#$ and $P_\#$ to denote $Z_\#(R, \ell, k)$ and $P_\#(R, \ell, k)$ respectively. Since $R$ and $\{Z_{i\ell}\}_{i,\ell}$ are independent, we can bound the required probability as follows

$$
\begin{aligned}
\Pr\left[Z_\# \geq U(1+\varepsilon)\right] &= \Pr\left[Z_\# \leq P_\# \cdot \left(1 + \frac{U(1+\varepsilon) - P_\#}{P_\#}\right)\right] \\
&\leq \exp\left(P_\# \cdot \left(\frac{U(1+\varepsilon) - P_\#}{P_\#}\right)^2 \cdot \frac{1}{2 + \frac{U(1+\varepsilon) - P_\#}{P_\#}}\right).
\end{aligned}
$$

Where we used the fact that: For any $\delta > 0$ and independent 0/1 random variables $Y_1, Y_2, \ldots, Y_n$, $\Pr\left[\sum_i Y_i > (1+\delta)\mu\right] < \exp\left(\frac{\mu\delta^2}{2+\delta}\right)$, where $\mu := \mathbb{E}[\sum_i Y_i]$ (see[52]). Simplifying the right-hand side of the above equation, we get:

$$\Pr\left[Z_\# \geq U(1+\varepsilon)\right] = \exp\left(-\frac{(U(1+\varepsilon) - P_\#)^2}{U(1+\varepsilon) + P_\#}\right). \tag{40}$$

To bound the right-hand side of Equation (40), we will use the following fact.

*Fact* E.4. For all $U, \varepsilon > 0$, $\frac{(U(1+\varepsilon)-x)^2}{U(1+\varepsilon)+x}$ attains its minima at $U$ over the domain $[0, U]$.

Since $P_\# \leq U$, from Fact E.4 it follows that the right-hand side of Equation (40) attains its maxima at $P_\# = U$. Substituting $P_\# = U$ in Equation (40), we get:

$$\Pr\left[Z_\# \geq U(1+\varepsilon)\right] \leq \exp\left(\frac{-U\varepsilon^2}{2+\varepsilon}\right). \tag{41}$$

$\square$

### E.3 Improved dependence of Theorem 4.1 on $\gamma$ on $\delta$

In this section, we show that given a constant $\psi > 0$, if $U$ satisfies that

$$\forall \ell \in [p], \forall k \in [n], \quad U_{k\ell} \geq \psi k,$$

then we can improve the dependence of $\gamma$ (from Equation (6)) on $\log\frac{2np}{\delta}$ and $\alpha$. Concretely, Theorem 4.1 holds for the following $\gamma$:

$$\forall k \in [n], \quad \gamma_k := \max_{\ell \in [p]} \sqrt{\frac{1}{2\psi} \cdot \log\left(\frac{2np}{\delta}\right) \cdot \frac{1}{U_{k\ell}}}. \tag{42}$$

The proof of this relies on analogous of Lemmas E.1 and E.3: Lemmas E.5 and E.6.

**Lemma E.5.** *For any position $k \in [n]$, attribute $\ell \in [p]$, parameter $\varepsilon \geq 0$, and lower bound constraint $L \in \mathbb{Z}_{\geq 0}^{n \times p}$, and ranking $x \in \mathcal{R}$, if $x$ satisfies that $P_\#(R, \ell, k) \geq L$ then with probability at least $1 - \exp\left(-2L^2\varepsilon^2 k^{-1}\right)$, it holds that $Z_\#(R, \ell, k) > L(1-\varepsilon)$.*

**Lemma E.6.** *For any position $k \in [n]$, attribute $\ell \in [p]$, parameters $\varepsilon \geq 0$ and $U \in \mathbb{R}$, and ranking $R \in \mathcal{R}$, where $R$ is possibly a random variable and is independent of $\{Z_{i\ell}\}_{i,\ell}$, if $R$ satisfies that $P_\#(R, \ell, k) \leq U$ then with probability at least $1 - \exp\left(-\frac{2U^2\varepsilon^2}{k}\right)$, it holds that $Z_\#(R, \ell, k) < U(1+\varepsilon)$.*

To prove the improved dependence of $\gamma$, it suffices to prove Propositions 6.1 and 6.2. For the new value of $\gamma$, their proofs change as follows:

**Proof of Proposition 6.1.** The parameters in Equation (9) remain the same. Hence, following the same argument, Equation (10) holds. Now, we can prove Equation (12) as follows:

$$
\begin{aligned}
\Pr\left[Z_\#(R, \ell, k) \geq U_{\ell k}(1 + \phi\gamma_k)\right] &= \Pr\left[Z_\#(R, \ell, k) \geq U'(1 + \zeta)\right] \\
&\qquad\qquad \text{(Using that } U'(1+\zeta) = U_{k\ell}(1 + \phi\gamma_k)) \\
&\leq \exp\left(-\frac{2(U')^2\zeta^2}{k}\right) \qquad \text{(Using Lemma E.6)} \\
&= \exp\left(-\frac{2(1-\phi)^2 U_{\ell k}^2 \gamma_k^2}{k}\right) \qquad \text{(Using Equation (9))} \\
&\leq \exp\left(-2\psi(1-\phi)^2 U_{\ell k}\gamma_k^2\right) \qquad \text{(Using that } U_{k\ell} \geq \psi k) \\
&\leq \frac{\delta}{2np}. \qquad\qquad \text{(Using Equation (42)) (43)}
\end{aligned}
$$

Proposition 6.1 follows by replacing Equation (12) by Equation (43) in the rest of its proof.

**Proof of Proposition 6.2.** The parameters in Equation (13) remain the same. Now, we can prove $\Pr\left[Z_{\#}(R', k, \ell) \leq U_{k\ell}\right] < 1 - \delta$ as follows:

$$\Pr\left[Z_{\#}(R', k, \ell) \leq U_{k\ell}\right] = \Pr\left[Z_{\#}(R', k, \ell) \leq L' \cdot (1 - \zeta)\right]$$

$$\text{(Using that } L'(1-\zeta) = U_{k\ell}(1 + b\gamma_k))$$

$$\leq \exp\left(-\frac{2\left(L'\right)^2 \zeta^2}{k}\right) \qquad \text{(Using Lemma E.5)}$$

$$= \exp\left(-\frac{2(\phi - b)^2 \gamma_k^2 U_{k\ell}^2}{k}\right) \qquad \text{(Using Equation (13))}$$

$$\leq \exp\left(-2\psi(\phi - b)^2 \gamma_k^2 U_{k\ell}\right) \qquad \text{(Using that } U_{k\ell} \geq \psi k)$$

$$< \frac{\delta}{2np} \qquad \text{(Using Equation (42) and Equation (13))} \quad (44)$$

$$< 1 - \delta. \qquad \text{(Using that } \delta < \tfrac{1}{2} \text{ and } n \geq 1) \quad (45)$$

The rest of the proof is identical.

*Proof of Lemma E.5.* First, note that since $x$ is not a function of the outcomes of the random variables $Z_{i\ell}$, $x$ is independent of the random variables $\{Z_{i\ell}\}_{i,\ell}$. Since $\ell$, $k$, and $x$ are fixed, we use $Z_{\#}$ and $P_{\#}$ to denote $Z_{\#}(R, \ell, k)$ and $P_{\#}(R, \ell, k)$ respectively. Now, we can bound the required probability as follows

$$\Pr\left[Z_{\#} \leq L(1 - \varepsilon)\right] = \Pr\left[Z_{\#} \leq P_{\#} \cdot \left(1 - \frac{P_{\#} - L(1 - \varepsilon)}{P_{\#}}\right)\right]$$

$$\leq \exp\left(-\frac{2}{k} \cdot P_{\#}^2 \cdot \left(\frac{P_{\#} - L(1 - \varepsilon)}{P_{\#}}\right)^2\right)$$

(Where we used the fact that: For any $\delta > 0$ and bounded random variables $Y_1, Y_2, \ldots, Y_n \in [0, 1]$, $\Pr\left[\sum_i Y_i < (1 - \delta)\mu\right] < \exp\left(-2\mu^2\delta^2 n^{-1}\right)$, where $\mu := \mathbb{E}[\sum_i Y_i]$)

$$= \exp\left(-\frac{2}{k} \cdot (P_{\#} - L(1 - \varepsilon))^2\right)$$

$$\leq \exp\left(-2L^2\varepsilon^2 k^{-1}\right).$$

$\square$

*Proof of Lemma E.6.* Since $\ell$, $k$, and $R$ are fixed, we use $Z_{\#}$ and $P_{\#}$ to denote $Z_{\#}(R, \ell, k)$ and $P_{\#}(R, \ell, k)$ respectively. Since $R$ and $\{Z_{i\ell}\}_{i,\ell}$ are independent, we can bound the required probability as follows

$$\Pr\left[Z_{\#} \geq U(1 + \varepsilon)\right] = \Pr\left[Z_{\#} \leq P_{\#} \cdot \left(1 + \frac{U(1 + \varepsilon) - P_{\#}}{P_{\#}}\right)\right]$$

$$\leq \exp\left(-\frac{2}{k} \cdot P_{\#}^2 \cdot \left(\frac{U(1 + \varepsilon) - P_{\#}}{P_{\#}}\right)^2\right).$$

Where we used the fact that: For any $\delta > 0$ and bounded random variables $Y_1, Y_2, \ldots, Y_n \in [0, 1]$, $\Pr\left[\sum_i Y_i > (1 + \delta)\mu\right] < \exp\left(-2\mu^2\delta^2 n^{-1}\right)$, where $\mu := \mathbb{E}[\sum_i Y_i]$ ([52]). Simplifying the right-hand side of the above equation, we get

$$\Pr\left[Z_{\#} \geq U(1 + \varepsilon)\right] \leq \exp\left(-\frac{2}{k}(U(1 + \varepsilon) - P_{\#})^2\right)$$

$$\leq \exp\left(-\frac{2U^2\varepsilon^2}{k}\right). \qquad \text{(Using that } P_{\#} \leq U)$$

$\square$

## E.4 NP-hardness result

**Theorem E.7.** *Given constants $c > 1$ and vector $\gamma \in \mathbb{R}^n_{\geq 0}$, , and matrices $P \in [0,1]^{m \times p}$, $W \in \mathbb{R}^{m \times n}_{\geq 0}$, $U \in \mathbb{R}^{n \times p}$, it is* **NP***-hard to decide if Program* (7) *is feasible.*

Theorem E.7 follows from Theorem 5.2 of [48], which proves that checking the feasibility of the following program is **NP**-hard.[¶]

$$\max_{x \in \{0,1\}^m} \quad \sum_{i=1}^m w_i^\circ x_i \tag{46}$$

$$\text{s.t.,} \quad \forall \ell \in [p^\circ], \quad \sum_{i=1}^{m^\circ} q_{i\ell}^\circ x_i \leq U_\ell^\circ, \tag{47}$$

$$\sum_{i=1}^{m^\circ} x_i = n^\circ. \tag{48}$$

Where we used a superscript "$\circ$" on the variables of [48], to differentiate between ours and [48]'s variables. Theorem E.7 follows from Theorem 5.2 of [48] by observing that Program (46) is a special case of Program (7), when:

$$n := n^\circ, m := m^\circ, p := p^\circ, \gamma := 1_n, P = q^\circ,$$
$$\forall k \in [n], \quad \gamma_k = 1,$$
$$U_{n\ell} = U_\ell^\circ,$$
$$\forall k \in [n] \setminus \{1\}, \quad U_{k\ell} = n,$$
$$\forall i \in [m], j \in [n], \quad W_{ij} = w_i^\circ.$$

Finally, we can choose any $c > 1$.

## E.5 Proof of Proposition E.8

Given a non-empty subset $\mathcal{C} \subseteq \mathcal{R}$ denoting a constraint, let $R_\mathcal{C}$ be the ranking with the highest utility in $\mathcal{C}$, i.e.,

$$R_\mathcal{C} := \text{argmax}_{R \in \mathcal{C}} \langle R, W \rangle.$$

In other words, $R_\mathcal{C}$ is the utility maximizing ranking subject to satisfying the "constraint" $\mathcal{C}$.

**Proposition E.8.** *Let $\mathcal{C}^\star$ be the set of all rankings that satisfy $(\varepsilon, \delta)$-constraint. For any subset $\mathcal{C} \subseteq \mathcal{R}$, such that $\mathcal{C} \neq \mathcal{C}^\star$, at least one of the following holds:*

- *there exists a matrix $W \in \mathbb{R}^{m \times n}_{\geq 0}$ such that, $R_\mathcal{C}$ does not satisfy $(\varepsilon, \delta)$-equal representation,*

- *there exists a matrix $W \in \mathbb{R}^{m \times n}_{\geq 0}$ such that, $\langle R_\mathcal{C}, W \rangle \leq \langle R_{\mathcal{C}^\star}, W \rangle \cdot \left(1 - \frac{1}{n}\right)$.*

We will use the following lemma in the proof of Proposition E.8.

**Lemma E.9.** *For all rankings $R \in \mathcal{R}$, there exists a matrix $W \in \mathbb{R}^{m \times n}_{\geq 0}$ such that for all other rankings $R' \in \mathcal{R}$, $R \neq R'$, it holds that $\langle R', W \rangle \leq \langle R, W \rangle \cdot \left(1 - \frac{1}{n}\right)$.*

*Proof.* Suppose $R$ ranks items $i_1, i_2, \ldots, i_n$, in that order, in the first $n$ positions. Pick $W \in [0,1]^{n \times m}$ such that $W_{ij} = 1$ if $i = i_j$ and 0 otherwise. $R$ has utility $\langle W, R \rangle = \sum_{j=1}^n (W)_{i_j j} = n$. We claim that $\langle W, R' \rangle \leq n - 1$. If this is true, then the lemma follows.

---

[¶]Theorem 5.2 of [48] states an **NP**-hardness result holds for a generalization of Program (46). However, in their proof they only consider the special case of Program (46). Thus, their proof also implies **NP**-hardness of Program (46).

Since $R \neq R'$, there exists a position $k \in [n]$ such that $(x_{\mathcal{C}})_{i_k k} = 0$. We can upper bound $\langle W, R' \rangle$ as follows:

$$
\begin{aligned}
\langle W, R' \rangle &= \sum_{j=1}^{n} \sum_{i=1}^{m} \mathbb{I}[i = i_j] \, (R')_{ij} && \text{(By the choice of } W) \\
&= \sum_{j=1}^{n} (R')_{i_j j} \\
&= \sum_{j=1}^{k-1} (R')_{i_j j} + 0 + \sum_{j=k+1}^{n} (R')_{i_j j} && \text{(Using that } (R')_{i_k k} = 0) \\
&\leq n - 1. && \text{(Using that for all } i \in [m] \text{ and } j \in [n], (W)_{ij} \leq 1)
\end{aligned}
$$

$\square$

*Proof of Proposition E.8.* Since $\mathcal{C} \neq \mathcal{C}^\star$, at least one of the sets $\mathcal{C} \setminus \mathcal{C}^\star$ or $\mathcal{C}^\star \setminus \mathcal{C}$ is nonempty. We divide the proof into two cases.

**Case A ($|\mathcal{C} \setminus \mathcal{C}^\star| \neq 0$):** In this case, there exists a ranking $R \in \mathcal{C}$ such that $R \notin \mathcal{C}^\star$. Since $\mathcal{C}^\star$ is the set of all rankings that satisfy $(\varepsilon, \delta)$-constraint, it follows that $R$ does not satisfy $(\varepsilon, \delta)$-constraint. Further, from Lemma E.9 it follows that there exists a matrix $W$ such that $R := \operatorname{argmax}_{R' \in \mathcal{R}} \langle R', W \rangle$. Since $\mathcal{C} \subseteq \mathcal{R}$, it follows that $R_{\mathcal{C}} = R$. Therefore, for this $W$, $R_{\mathcal{C}}$ does not satisfy $(\varepsilon, \delta)$-constraint.

**Case B ($|\mathcal{C}^\star \setminus \mathcal{C}| \neq 0$):** In this case, there exists a ranking $R \in \mathcal{C}^\star$ such that $R \notin \mathcal{C}$. From Lemma E.9 it follows that there exists a matrix $W$ such that, for rankings $R'$ different from $R$ (i.e., $R \neq R'$),

$$
\langle R', W \rangle \leq \langle R, W \rangle \cdot \left( 1 - \frac{1}{n} \right).
$$

Thus, for this $W$, it follows that

$$
\langle R_{\mathcal{C}^\star}, W \rangle \cdot \left( 1 - \frac{1}{n} \right) \geq \langle R, W \rangle \cdot \left( 1 - \frac{1}{n} \right) \geq \langle R', W \rangle.
$$

In particular, for $R' = R_{\mathcal{C}}$, we get $\langle R_{\mathcal{C}^\star}, W \rangle \cdot \left( 1 - \frac{1}{n} \right) \geq \langle R', W \rangle$.

$\square$

### E.6 Proof of Lemma E.10

Suppose there are two groups $G_1$ and $G_2$. Let $R_E$ be the optimal solution to Equation (5) and let $R^\star$ be the ranking with the highest utility subject to satisfying $(\gamma, \delta)$-equal representation constraints for the following $\gamma$:

$$
\forall k \in [n], \quad \gamma_k := \frac{1}{k} + 2 \sqrt{\frac{6}{k} \cdot \log\left( \frac{2n}{\delta} \right)}. \tag{49}
$$

**Lemma E.10.** *There exists a matrices $P \in [0,1]^{m \times 2}$ and $W \in [0,1]^{m \times 2}$ such that*

- *$R_E$ satisfies $(\gamma, \delta)$-equal representation and has utility 0,*
- *$R^\star$ has utility 1.*

*Proof.* Let $P$ be the matrix with $P_{i1} = P_{i2} = \frac{1}{2}$ for all $i \in \{1, 2, \ldots, m-1\}$ and $P_{m1} = 1$ and $P_{m1} = 0$. Let $W$ be the matrix whose first $m-1$ rows are 0, and the last row has is all 1s. Hence, only the last item, say $i_m$, has a nonzero contribution to the utility: If a ranking $R$ ranks $i_m$ in the first $n$ positions, then the utility of $R$ is 1, otherwise the utility of $R$ is 0.

Our first claim will follow because the choice of $P$ ensures that any ranking which ranks $i_m$ in the first $n$ positions cannot satisfy Equation (5). To see this, suppose $R$ ranks $i_m$ at the $k$-th position, then

$$
\begin{aligned}
\mathbb{E}\left[\sum_{i \in G_1} \sum_{j=1}^{k} R_{ij}\right] &= \sum_{i \in [m]} \sum_{j=1}^{k} P_{i1} R_{ij} \\
&= 1 + \sum_{i \in [m] \setminus \{i_m\}} \sum_{j=1}^{k-1} P_{i1} R_{ij} && \text{(Using that } P_{i_m,1} = 1\text{)} \\
&= \frac{k+1}{2} && \text{(Using that } P_{i,1} = \tfrac{1}{2} \text{ for all } i \neq i_m\text{)} \\
&> \frac{k+1}{2}.
\end{aligned}
$$

Hence, $R$ cannot satisfy Equation (5).

To prove our second claim, we will construct a ranking which has utility 1 and satisfies $(\gamma, \delta)$-equal representation . It suffices to choose any ranking $R$ which places $i_m$ in the first $n$ position satisfies constraint. By our earlier argument this ranking has a utility 1. Let $Z_j$ be the indicator random variable that the item in the $j$-th position in $R$ belongs to $G_1$. This implies that $\sum_{i \in G_1} \sum_{j=1}^{k} R_{ij} = \sum_{j=1}^{k} Z_j$ for all $k$. Further, by the choice of $P$, we have

$$
\frac{k}{2} \leq \mathbb{E}\left[\sum_{j=1}^{k} Z_j\right] \leq \frac{k+1}{2}. \tag{50}
$$

Further, by Definition 3.2, we have that $Z_j$ is independent of $Z_k$ for any $j \neq k$. Let $\varepsilon_k := \sqrt{\frac{6}{k} \cdot \log\left(\frac{2n}{\delta}\right)}$. Using the above, we have

$$
\begin{aligned}
\Pr\left[\sum_{i \in G_1} \sum_{j=1}^{k} R_{ij} \geq \frac{k+1}{2} \cdot (1 + \varepsilon_k)\right] &= \Pr\left[\sum_{j=1}^{k} Z_j \geq \frac{k+1}{2} \cdot (1 + \varepsilon_k)\right] \\
&\leq \Pr\left[\sum_{j=1}^{k} Z_j \geq \mathbb{E}\left[\sum_{j=1}^{k} Z_j\right] \cdot (1 + \varepsilon_k)\right] \\
&\qquad\qquad \text{(Using Equation (50))} \\
&\leq \exp\left(-\frac{\varepsilon_k^2}{3} \cdot \mathbb{E}\left[\sum_{j=1}^{k} Z_j\right]\right) \\
&\qquad\qquad \text{(Using the Chernoff's bound, see [52])} \\
&\leq \exp\left(-\frac{\varepsilon_k^2 k}{6}\right) && \text{(Using Equation (50))} \\
&\leq \frac{\delta}{2n}. && \text{(Using that } \varepsilon_k := \sqrt{\frac{6}{k} \cdot \log\left(\frac{2n}{\delta}\right)}\text{)}
\end{aligned}
$$

Further, as $\gamma_k \geq \frac{k+1}{k} \cdot (1 + \varepsilon_k)$, we get

$$
\begin{aligned}
\Pr\left[\sum_{i \in G_1} \sum_{j=1}^{k} R_{ij} \geq \frac{k}{2} \cdot (1 + \gamma_k)\right] &\leq \Pr\left[\sum_{i \in G_1} \sum_{j=1}^{k} R_{ij} \geq \frac{k+1}{2} \cdot (1 + \varepsilon_k)\right] \\
&\leq \frac{\delta}{2n}.
\end{aligned}
$$

Further, considering $1 - Z_j$ and repeating a similar argument for $G_2$, we get

$$\Pr\left[\sum_{i \in G_2}\sum_{j=1}^{k} R_{ij} \geq \frac{k}{2} \cdot (1 + \varepsilon_k)\right] = \Pr\left[\sum_{j=1}^{k}(1 - Z_j) \geq \frac{k}{2} \cdot (1 + \gamma_k)\right]$$

$$\leq \Pr\left[\sum_{j=1}^{k}(1 - Z_j) \geq \mathbb{E}\left[\sum_{j=1}^{k}(1 - Z_j)\right] \cdot (1 + \gamma_k)\right]$$
(Using Equation (50))

$$\leq \exp\left(-\frac{\gamma_k^2}{3} \cdot \mathbb{E}\left[\sum_{j=1}^{k}(1 - Z_j)\right]\right)$$
(Using the Chernoff's bound, see [52])

$$\leq \exp\left(-\frac{\gamma_k^2(k-1)}{6}\right)$$
(Using Equation (50))

$$\leq \frac{\delta}{2n}.$$
(Using Equation (49))

By taking the union bound over all $k$, one can show that $R$ satisfies $(\gamma, \delta)$-equal representation. $\quad\square$

### E.7 Proof of Proposition E.11

**Proposition E.11.** *There exist $p \in [0,1]^m$ such that (4) is non-convex in $R$.*

*Proof.* It suffices to specify $n$, $m$, $p$, $\varepsilon$, $\delta$, and two rankings $R_1$ and $R_2$ such that both $R_1$ and $R_2$ satisfy $(\varepsilon, \delta)$-equal representation, but $\frac{R_1 + R_2}{2}$ does not satisfy $(\varepsilon, \delta)$-equal representation.

Define $n := 2$, $m := 4$, and $\varepsilon := \begin{bmatrix} \frac{1}{3} & \frac{1}{3} \end{bmatrix}^\top$. Fix any $0 < \delta < \frac{1}{2}$. Define

$$p := \begin{bmatrix} 1 & 0 & \delta & 1 - \delta \end{bmatrix}^\top.$$

Let $R_1$ be the ranking that places items 1 and 3 in the first and second position, and $R_2$ be the ranking that places items 2 and 4 in the first and second position, i.e.,

$$R_1 := \begin{bmatrix} 1 & 0 & 0 & 0 \\ 0 & 0 & 1 & 0 \end{bmatrix} \quad \text{and} \quad R_2 := \begin{bmatrix} 0 & 1 & 0 & 0 \\ 0 & 0 & 0 & 1 \end{bmatrix}.$$

If $1 \in G_1$ and $3 \in G_2$, then $R_1$ places an equal number of items from $G_1$ and $G_2$ in the first two positions, and hence, satisfies equal representation. This event, happens with probability $p_1(1 - p_3) = 1 - \delta$. Thus, $R_1$ satisfies $(0, \delta)$-equal representation, and hence, $(\varepsilon, \delta)$-equal representation. Replace item 1 and 3 with 2 and 4 and swap $G_1$ and $G_2$ in the above argument, to get that $R_2$ also satisfies $(\varepsilon, \delta)$-equal representation.

However, we claim that $\frac{R_1 + R_2}{2}$ does not satisfy $(\varepsilon, \delta)$-equal representation. Note that with probability 1, $1 \in G_1$ and $2 \in G_2$. If $3, 4 \in G_1$ or $3, 4 \in G_2$, then $\frac{R_1 + R_2}{2}$ violates the equal representation constraint on the top-2 positions by a multiplicative factor of $\frac{3}{2}$. At least one of these events happens with probability $p_3 p_4 + (1 - p_3)(1 - p_4) = 2\delta(1 - \delta) > \delta$, as $\delta < \frac{1}{2}$. Thus, $\frac{R_1 + R_2}{2}$ does not satisfy $(\varepsilon, \delta)$-equal representation for the specified $\varepsilon := \begin{bmatrix} \frac{1}{3} & \frac{1}{3} \end{bmatrix}^\top$ and $\delta < \frac{1}{2}$. $\quad\square$

### E.8 Proof of Theorem E.12

In this section, we prove the following theorem.

**Theorem E.12.** *Given $p \in [0,1]^m$, $\delta \in (0,1]$, $W \in \mathbb{R}_{\geq 0}^{m \times n}$, $\varepsilon \in [0,1]^n$, and $V \geq 0$ it is **NP**-hard to decide if the value of Program (4) is at least $V$.*

Recall that constraint (52) is necessary and sufficient to satisfy $(\varepsilon, \delta)$-equal representation, and hence, the value of (51) is the maximum utility of a ranking subject to satisfying $(\varepsilon, \delta)$-equal representation.

$$\max_{R \in \mathcal{R}} \quad \langle R, W \rangle \tag{51}$$

$$\text{s.t. w.p. at least } 1 - \delta \text{ over draw of } G_1, G_2, \tag{52}$$

$$\forall k \in [n], \ \forall \ell \in [2], \quad \sum_{i \in G_\ell} \sum_{j=1}^{k} R_{ij} \leq \frac{k}{2} \cdot (1 + \varepsilon_k).$$

We will show that the decision version of (51) is **NP**-hard:

**Theorem E.13.** *Given $L \geq 0$, $\delta \in [0, 1]$, $\varepsilon \in [0, 1]^n$, $P \in [0, 1]^{m \times p}$, and $W \in \mathbb{R}_{\geq 0}^{m \times n}$ it is **NP**-hard to decide if the value of (51) is at least $L$.*

The proof of Theorem E.13 proceeds in two steps. In the first step, we reduce (53) to (51). In the second step, we prove that (53) is **NP**-hard because the **NP**-complete product partition problem reduces to (53). Together, the two steps imply the hardness of (51). The proof of the second step is inspired by the construction of [57] for the product knapsack problem, which is similar to (53).

***Step 1: Reduction from* (53) *to* (51)*.*    In this step, we will reduce the following problem to (51).

---

*Input:* $L \geq 0$, $n \in [m]$, $\delta \in [0, 1]$, $U \in \left[0, \frac{n}{2}\right]$ $v \in \mathbb{R}_{\geq 0}^m$, and $P \in [0, 1]^{m \times p}$
*Decision problem:* Is the value of (53) at least $L$?

$$\max_{S \subseteq [m]: \ |S| = n} \quad \sum_{i \in S} v_i \tag{53}$$

$$\text{s.t.} \quad \text{w.p. at least } 1 - \delta \text{ over draw of } G_1, G_2,$$

$$|S \cap G_1| \leq U + \frac{n}{2} \quad \text{and} \quad |S \cap G_2| \leq U + \frac{n}{2}.$$

---

**Reduction.** Given an instance of (53) we construct the following instance of (51):

$$W := v 1_n^\top, \tag{54}$$

$$\varepsilon_1 = \varepsilon_2 = \cdots = \varepsilon_{n-1} := \frac{2n}{k} - 1, \tag{55}$$

$$\varepsilon_n := \frac{2U}{n} - 1, \tag{56}$$

where $1_n := (1, \ldots, 1) \in \mathbb{R}^n$.[‖] The parameters $L$, $\delta$, and $P$ are the same as the instance of (53).

The reduction from (53) to (51) is as follows: First solve (51) to obtain a ranking $R$. Let $S$ be the set of items $R$ places in the top-$n$ positions. Output $S$. Clearly, this is a polynomial-time reduction. It remains to prove that it is sound and complete.

In our construction, Condition (54) implies that the utility of a ranking only depends on the set of $n$ items it places in the top-$n$ positions, and hence, any two rankings that place the same set of items in the top-$n$ positions have the same utility. Condition (55) ensures that any ranking satisfies the constraints in the first $n - 1$ positions with probability 1. This is because, for all $k \in [n-1]$, $\frac{k}{2}(1 + \varepsilon_k) = n > k$. Thus, a ranking $R$ is feasible for (51) if and only if it satisfies: With probability at least $1 - \delta$ over draw of $G_1, G_2$,

$$\forall \ell \in [2], \quad \sum_{i \in G_\ell} \sum_{j=1}^{k} R_{ij} \leq \frac{n}{2} \cdot (1 + \varepsilon_n) = U + \frac{n}{2}.$$

---

[‖]To be precise, we consider $\varepsilon_1 = \varepsilon_2 = \cdots = \varepsilon_{n-1} := \min\left\{1, \frac{2n}{k} - 1\right\}$ and $\varepsilon_n := \min\left\{1, \frac{2U}{n} - 1\right\}$.

**Soundness and completeness.** Fix any $R \in \mathcal{R}$. Let $S$ be the set of items $R$ places in the top-$n$ positions. It holds that

$$\langle R, W \rangle \overset{(54)}{=} \sum_{i \in S} v_i.$$

It remains to show that $R$ is feasible for (51) if and only if $S$ is feasible for (53). Due to conditions (55) and (56), $R$ is feasible for (51) iff: With probability at least $1 - \delta$ over draw of $G_1, G_2$,

$$\forall \ell \in [2], \quad \sum_{i \in G_\ell} \sum_{j=1}^{k} R_{ij} \leq U + \frac{n}{2}.$$

Since by the definition of $S$, for all $T \subseteq [m]$, $\sum_{i \in T} \sum_{j=1}^{n} R_{ij} = |S \cap T|$, it follows that with probability 1 $\sum_{i \in G_\ell} \sum_{j=1}^{n} R_{ij} = |S \cap G_\ell|$. Thus, $S$ is feasible for (53) if and only if $R$ is feasible for (51). Thus, the reduction is sound and complete.

***Step 2: Reduction from product partition problem to** (53).* We consider the following version of the product partition problem:

---

*Cardinality constrained product partition problem (CPPP)*

*Input:* $a_1, a_2, \ldots, a_q \in \mathbb{Z}_{\geq 0}$ and $\ell \in \{0, 1, \ldots, q\}$.
*Decision problem:* Is there a set $S \subseteq [q]$ of size $\ell$ such that

$$\prod_{i \in S} a_i = \prod_{i \in [q] \setminus S} a_i?$$

---

The usual product partition problem (PPP) does not require $S$ to have size $\ell$ and is known to be **NP**-complete. CPPP is clearly in **NP**. To see that CPPP is **NP**-complete, one can reduce PPP to CPPP: To see this, given an instance of PPP, construct $q + 1$ instances of CPPP, one for each value of $\ell \in \{0, 1, \ldots, q\}$. Then, PPP is a 'Yes' instance if and only if at least one of the $q + 1$ CPPP instances in a 'Yes' instance. Thus, it follows that CPPP is also **NP**-complete.

**Assumptions on CPPP instances without loss of generality.** The decision problem for CPPP is simple for instances with $\ell = 0$, or with one or more of $a_1, \ldots, a_q$ as 0. As all inputs are integral, without loss of generality, we assume that $\ell \geq 1$ and $a_1, \ldots, a_q \geq 1$. Note that if in an CPPP $\sqrt{\prod_{i=1}^{q} a_i}$ is non-integral, then it is a 'No' instance. This can be verified in polynomial time, and hence, without loss of generality, we assume that $\sqrt{\prod_{i=1}^{q} a_i}$ is integral.

**Reduction from CPPP to** (53). Given an instance of CPPP, we construct an instance of (53) with

$$n := 2\ell, \quad m := q + \ell, \quad U := \ell - 1, \quad \text{and} \quad \delta := \left(\frac{1}{a_{\max}}\right)^{\ell^2}, \tag{57}$$

where $a_{\max} := \max_{i \in [q]} a_i$. Further, define constants

$$M := (\ell + 2) \cdot \sqrt{\prod_{i=1}^{q} a_i} \quad \text{and} \quad B := q \lceil M \log(a_{\max}) \rceil + 1. \tag{58}$$

We choose $v$ so that the first $q$ items correspond to the $q$ numbers in the CPPP instance, and the next $\ell$ items have a "high" value:

$$\forall i \in [q], \quad v_i := \lceil M \log(a_i) \rceil, \tag{59}$$
$$\forall i \in [\ell], \quad v_{i+q} := L. \tag{60}$$

Note that each of the last $\ell$ items has a value larger than the total value of the first $q$ items, i.e.,

$$\forall\, i \in [\ell], \quad v_{i+q} = B > \sum_{j \in [q]} v_j. \tag{61}$$

We choose $P$ so that for the first $q$ items $P_{i,1} \propto a_i^\ell$ and the next $\ell$ are in $G_1$ with probability 1:

$$\forall i \in [q], \qquad P_{i,1} := \left(\frac{a_i}{a_{\max}}\right)^\ell \cdot \frac{1}{\sqrt{\prod_{i=1}^q a_i}} \quad \text{and} \quad P_{i,2} = 1 - P_{i,1}, \tag{62}$$

$$\forall i \in [\ell], \quad P_{i+q,1} := 1 \qquad\qquad\qquad\qquad \text{and} \;\; P_{i+q,2} = 1 - P_{i+q,1}. \tag{63}$$

Finally, let

$$L := \ell B + \left\lfloor \frac{M}{2} \sum_{i=1}^q \log(a_i) \right\rfloor. \tag{64}$$

The reduction from CPPP to (53) is as follows: First solve the constructed instance of (53) to get $S$. Then output $S \backslash Q$, where
$$Q := [\ell + q] \setminus [q]$$
is the set of the last $\ell$ items.

Let $C \in \mathbb{Z}$ be the bit complexity of the input for the given instance of (53). To show that the reduction is polynomial time, it suffices to show that $L$ and $\lceil M \log(a_1) \rceil, \ldots, \lceil M \log(a_q) \rceil$ can be computed in $\mathrm{poly}(C)$ time. Note that, $M \leq 2^{O(C)}$, and hence, to compute $\lceil M \log(a_i) \rceil$ it suffices to compute $\log(a_i)$ up to $O(C)$ bits, which can be done in $\mathrm{poly}(C)$ time. Similarly, to compute $L$ it suffices to compute $\sum_{i=1}^q \log(a_i)$ up to $O(C)$ bits, which can be done in $\mathrm{poly}(C)$ time. Thus, the reduction is polynomial time.

The choice of $L$ and $v$ ensures that the following fact holds.

*Fact* E.14. If a set $S \subseteq [q]$ satisfies $\sum_{i \in S} v_i \geq L$ and $|S| = n$, then $S \supseteq Q$.

*Proof.* Suppose toward a contradiction that satisfies $\sum_{i \in S} v_i \geq L$ and $|S| = n$ but $S$ does not contain $Q$. Since $S = n = 2\ell$ Then,

$$\begin{aligned}
\sum_{i \in S} v_i \;&=\; \sum_{i \in S \cap Q} v_i + \sum_{i \in S \setminus Q} v_i \\
&\leq\; |S \cap Q| \cdot \max_{i \in Q} v_i + \sum_{i \in [q] \setminus Q} v_i && \text{(Using } S \subseteq [q] \text{ and } v_i \geq 0) \\
&\overset{(60),\,(61)}{<}\; |S \cap Q| \cdot B + B \\
&<\; |Q| \cdot B && \text{(Using that } |S \cap Q| \leq |Q| - 1 \text{ and } B > 0) \\
&\leq\; L. && \text{(Using (64), } |Q| = \ell, \text{ and } L \geq \ell B)
\end{aligned}$$

$\square$

**Soundness.** Suppose $S$ is feasible for (53) and satisfies $\sum_{i \in S} v_i \geq L$. Due to (63), with probability 1, $G_1 \supseteq Q$. Hence, $G_2 \cap Q = \emptyset$. Thus,

$$\text{with probability 1,} \quad |S \cap G_2| = |(S \setminus Q) \cap G_2| \leq |S \setminus Q|.$$

Since $\sum_{i \in S} v_i \geq L$ and $|S| = n$ (as $S$ is feasible for (53)), Fact E.14 implies that $S \supseteq Q$, hence $|S \setminus Q| = |S| - \ell$. Combining this with the above equation, we get that

$$\text{with probability 1,} \quad |S \cap G_2| \leq |S| - \ell = \ell. \qquad\qquad \text{(Using that } |S| = n = 2\ell)$$

Since $U \geq 0$,

$$\text{with probability } 1, \quad |S \cap G_2| \leq U + \ell. \tag{65}$$

$S$ is feasible for (53) iff:

$$\Pr_{G_1, G_2}[|S \cap G_1| \leq U + \ell \text{ and } |S \cap G_2| \leq U + \ell] \geq 1 - \delta$$

$$\stackrel{(65)}{\iff} \Pr_{G_1, G_2}[|S \cap G_1| \leq U + \ell] \geq 1 - \delta$$

$$\iff \Pr_{G_1, G_2}[|(S \setminus Q) \cap G_1| \leq U + \ell] \geq 1 - \delta$$

$$\text{(Using that with probability 1, } S, G_1 \supseteq Q)$$

$$\iff \Pr_{G_1, G_2}[|S' \cap G_1| \leq U] \geq 1 - \delta$$

$$\iff \Pr_{G_1, G_2}[|S' \cap G_1| > U] \leq \delta$$

$$\iff \Pr_{G_1, G_2}[|S' \cap G_1| = n] \leq \delta \qquad \text{(Using that } U = n - 1 \text{ and } |S'| = \ell)$$

$$\iff \prod_{i \in S'} P_{i1} \leq \delta$$

$$\stackrel{(63),(62),(57)}{\iff} a_{\max}^{-\ell \cdot |S'|} \cdot \left(\prod_{i \in [q]} a_i\right)^{-|S'|/2} \cdot \prod_{i \in S'} a_i^{\ell} \leq \left(\frac{1}{a_{\max}^{\ell}}\right)^{\ell}$$

$$\iff \prod_{i \in S'} a_i \leq \sqrt{\prod_{i \in [q]} a_i}. \qquad \text{(Using that } \ell > 0, a_1, \ldots, a_q > 0, \text{ and } |S'| = \ell) \tag{66}$$

Since $S$ is feasible for (53), it holds that

$$\prod_{i \in S'} a_i \leq \sqrt{\prod_{i \in [q]} a_i}.$$

To show that $S'$ is feasible for CPPP, it remains to show that the above equation holds with equality. Suppose toward a contradiction that $\prod_{i \in S'} a_i < \sqrt{\prod_{i \in [q]} a_i}$. Then, because $\sqrt{\prod_{i \in [q]} a_i}$ and $a_1, \ldots, a_q$ are integral

$$\prod_{i \in S'} a_i \leq \sqrt{\prod_{i \in [q]} a_i} - 1.$$

Because $M \geq 0$, taking the logarithm we get

$$M \sum_{i \in S'} \log a_i \leq M \log \left(\sqrt{\prod_{i \in [q]} a_i} - 1\right). \tag{67}$$

To upper bound the RHS, we will use the following fact:

*Fact* E.15. For all $x \geq 1$, $\log x - \log (x - 1) \geq \frac{1}{x}$.

Using Fact E.15 with $x = \sqrt{\prod_{i \in [q]} a_i}$ (as $a_1, \ldots, a_q \geq 1$), we get

$$\log \left(\sqrt{\prod_{i \in [q]} a_i}\right) - \log \left(\sqrt{\prod_{i \in [q]} a_i} - 1\right) \geq \frac{1}{\sqrt{\prod_{i \in [q]} a_i}}.$$

Hence, by (58)

$$M = (\ell + 2) \cdot \sqrt{\prod_{i \in [q]} a_i} \geq \frac{\ell + 2}{\log \left( \sqrt{\prod_{i \in [q]} a_i} \right) - \log \left( \sqrt{\prod_{i \in [q]} a_i} - 1 \right)}.$$

On rearranging, we get

$$M \log \left( \sqrt{\prod_{i \in [q]} a_i} - 1 \right) \leq M \log \left( \sqrt{\prod_{i \in [q]} a_i} \right) - \ell - 2.$$

Substituting this in (67), we get

$$M \sum_{i \in S'} \log a_i \leq M \log \left( \sqrt{\prod_{i \in [q]} a_i} \right) - \ell - 2.$$

Since for all $i \in S'$, $v_i \leq M \log (a_i) + 1$, it follows that

$$\sum_{i \in S'} v_i \leq \frac{M}{2} \log \left( \prod_{i \in [q]} a_i \right) - 2 < \left\lfloor \frac{M}{2} \log \left( \prod_{i \in [q]} a_i \right) \right\rfloor. \tag{68}$$

Thus,

$$\sum_{i \in S} v_i = \sum_{i \in S \cap Q} v_i + \sum_{i \in S \setminus Q} v_i$$

$$= \ell B + \sum_{i \in S'} v_i \qquad \text{(Using that } S \supseteq Q \text{ and } S' := S \setminus Q)$$

$$\overset{(68)}{<} \ell B + \left\lfloor M \log \left( \sqrt{\prod_{i \in [q]} a_i} \right) \right\rfloor$$

$$= L.$$

This is a contradiction to $\sum_{i \in S} v_i \geq L$.

**Completeness.** It suffices to show that if $S'$ is feasible for the given instance of CPPP, then $S := S' \cup Q$ is feasible for (53) and satisfies $\sum_{i \in S} v_i \geq A$.

Due to (63), with probability 1, $G_1 \supseteq Q$. Hence, $G_2 \cap Q = \emptyset$. Thus,

$$\text{with probability 1,} \quad |S \cap G_2| = |(S \setminus Q) \cap G_2| \leq |S \setminus Q| = |S'| = \ell,$$

where the last equality holds as $S'$ is feasible for the given instance of CPPP. This implies that (65) holds. Hence, by following the same arguments, (66) also holds. Thus, $S := S' \cup Q$ is feasible for (53)

It remains to show that $\sum_{i \in S} v_i \geq L$.

$$\sum_{i \in S} v_i = \sum_{i \in Q} v_i + \sum_{i \in S'} v_i \qquad \text{(Using that } S := S' \cup Q)$$

$$\overset{(60)}{=} \ell B + \sum_{i \in S'} v_i$$

$$\overset{(59)}{\geq} \ell B + \sum_{i \in S'} M \log a_i$$

$$= \ell B + \frac{M}{2} \log \left( \prod_{i \in [q]} a_i \right) \qquad \text{(Using that } \prod_{i \in S'} a_i = \prod_{i \in [q]} a_i)$$

$$\overset{(64)}{\geq} A.$$

# F Extension of theoretical results to position-weighted constraints

In this section, we extend Theorem 4.1 to position-weighted version of fairness constraints. In particular, given position-discounts

$$v_1 \geq v_2 \geq \cdots \geq v_n$$

and a matrix $U \in \mathbb{Z}_+^{n \times p}$ the position-weighted fairness constraint requires a ranking $R$ to satisfy:

$$\forall k \in [n], \ell \in [p], \quad \sum_{i \in G_\ell} \sum_{j \in [k]} v_j R_{ij} \leq U_{k\ell}$$

for all $k$ and $\ell$. For these constraints, we consider the following analogue of $(\varepsilon, \delta)$-constraints: A ranking $R$ is said to satisfy $(\varepsilon, \delta, v)$-constraint if with probability at least $1 - \delta$ over the draw of $G_1, \ldots, G_p$

$$\forall k \in [n] \, \forall \ell \in [p], \quad \sum_{i \in G_\ell} \sum_{j=1}^{k} v_j R_{ij} \leq U_{k\ell}(1 + \varepsilon_k). \tag{69}$$

For these position-dependent constraints, our framework largely remains the same and is stated in Program (72). Compared to Program (7), the main difference is in the left-hand side of Program (71). We can prove the guarantees on the fairness and accuracy of the optimal solution of Program (72), under the additional assumption that, for a constant $\psi > 0$, $U$ satisfies that

$$\forall \ell \in [p], \forall k \in [n], \quad U_{k\ell} \geq \psi k. \tag{70}$$

The parameter $\psi$ shows up in Equation (71).

---

**Our Fair-Ranking Framework for Position-Dependent Constraints**

*Input:* Matrices $P \in [0,1]^{m \times p}$, $W \in \mathbb{R}_{\geq 0}^{m \times n}$, $U \in \mathbb{R}^{n \times p}$
*Parameters:* A constant $c > 1$, a failure probability $\delta \in (0, 1]$, and for each $k \in [n]$, a relaxation parameter

$$\gamma_k := \frac{1}{\psi} \cdot \log\left(\frac{2np}{\delta}\right) \cdot \max_{\ell \in [p]} \sqrt{\frac{1}{U_{k\ell}}}. \tag{71}$$

---

*Program:*

$$\max_{R \in \mathcal{R}} \langle R, W \rangle, \tag{72}$$
$$\text{s.t. } \forall \ell \in [p] \;\; \forall k \in [n]$$

$$\sum_{i \in [m], j \in [k]} v_j P_{i\ell} R_{ij} \leq U_{k\ell}\left(1 + \frac{2\sqrt{c} - 1}{2\sqrt{c}} \cdot \gamma_k\right). \tag{73}$$

---

We prove the following guarantees on the fairness and accuracy of the optimal solution of Program (72).

**Theorem F.1.** *Let $\gamma \in \mathbb{R}^n$ be as defined in Equation (71). If the matrix $U \in \mathbb{Z}_+^{n \times p}$ satisfies that for all $\ell \in [p]$ and $k \in [n]$, $U_{k\ell} \geq \psi k$, then is an optimization program Program (72), parameterized by a constant $c$ and failure probability $\delta$, such that for any $c > 1$ and $\delta \in (0, \frac{1}{2}]$ its optimal solution satisfies $(c\gamma, \delta, v)$-constraint and has a utility at least as large as the utility of any ranking satisfying $((c - \sqrt{c})\gamma, \delta, v)$-constraint.*

The proof of Theorem F.1 is analogous to the proof of Theorem 4.1. Here, we highlight the differences.

**Notation**. Recall that for each item $i \in [m]$ and group $\ell \in [p]$, let $Z_{i\ell} \in \{0, 1\}$ is indicator random variable that $Z_i := \mathbb{I}[G_\ell \ni i]$.

The first change is in the definition of $Z_\#(R, \ell, k)$. In particular, we need to define

$$Z_\#(R, \ell, k) = \sum_{i \in G_\ell} \sum_{j=1}^{k} v_j R_{ij}.$$

For the new definition of $Z_\#$, we have following concentration result.

**Lemma F.2.** *For any position $k \in [n]$, group $\ell \in [p]$, parameters $\varepsilon \geq 0$ and $L, U \in \mathbb{R}$, and ranking $R \in \mathcal{R}$, where $R$ is possibly a random variable independent of $\{Z_{i\ell}\}_{i,\ell}$, if $P_{\#}(R, \ell, k) \leq U$ or $P_{\#}(R, \ell, k) \geq L$ then the following equations hold respectively*

$$\Pr\left[Z_{\#}(R, \ell, k) < (1 + \varepsilon)\, U\right] \geq 1 - e^{-\frac{2U^2 \varepsilon^2}{k}},$$

$$\Pr\left[Z_{\#}(R, \ell, k) > (1 - \varepsilon)\, L\right] \geq 1 - e^{-\frac{2L^2 \varepsilon^2}{k}}.$$

The proof of Lemma F.2 is identical to the proofs of Lemmas E.5 and E.6; the only change is the new definition of $Z_{\#}$.

To prove Theorem F.1, it suffices to prove analogues of Propositions 6.1 and 6.2 for the new definition of $Z_{\#}$. Their proofs change as follows:

**Proof of Proposition 6.1** The parameters in Equation (9) remain the same. Hence, following the same argument, Equation (10) holds. Now, we can prove Equation (12) as follows:

$$
\begin{aligned}
\Pr\left[Z_{\#}(R, \ell, k) \geq U_{\ell k}(1 + \phi \gamma_k)\right] \quad &= \quad \Pr\left[Z_{\#}(R, \ell, k) \geq U'(1 + \zeta)\right] \\
&\qquad\qquad \text{(Using that } U'(1 + \zeta) = U_{k\ell}(1 + \phi\gamma_k)) \\
&\leq \quad \exp\left(-\frac{2\left(U'\right)^2 \zeta^2}{k}\right) \qquad\qquad \text{(Using Lemma F.2)} \\
&= \quad \exp\left(-\frac{2(1 - \phi)^2 U_{\ell k}^2 \gamma_k^2}{k}\right) \qquad \text{(Using Equation (9))} \\
&\leq \quad \exp\left(-2\psi(1 - \phi)^2 U_{\ell k} \gamma_k^2\right) \quad \text{(Using that } U_{k\ell} \geq \psi k) \\
&\leq \quad \frac{\delta}{2np}. \qquad\qquad\qquad \text{(Using Equation (71))} \quad (74)
\end{aligned}
$$

Proposition 6.1 follows by replacing Equation (12) by Equation (74) in the rest of its proof.

**Proof of Proposition 6.2** The parameters in Equation (13) remain the same. Now, we can prove $\Pr\left[Z_{\#}(R', k, \ell) \leq U_{k\ell}\right] < 1 - \delta$ as follows:

$$
\begin{aligned}
\Pr\left[Z_{\#}(R', k, \ell) \leq U_{k\ell}\right] \quad &= \quad \Pr\left[Z_{\#}(R', k, \ell) \leq L' \cdot (1 - \zeta)\right] \\
&\qquad\qquad \text{(Using that } L'(1 - \zeta) = U_{k\ell}(1 + b\gamma_k)) \\
&\leq \quad \exp\left(-\frac{2\left(L'\right)^2 \zeta^2}{k}\right) \qquad\qquad \text{(Using Lemma F.2)} \\
&= \quad \exp\left(-\frac{2(\phi - b)^2 \gamma_k^2 U_{k\ell}^2}{k}\right) \qquad \text{(Using Equation (13))} \\
&\leq \quad \exp\left(-2\psi(\phi - b)^2 \gamma_k^2 U_{k\ell}\right) \qquad \text{(Using that } U_{k\ell} \geq \psi k) \\
&< \quad \frac{\delta}{2np} \qquad\qquad \text{(Using Equation (71) and Equation (13))} \quad (75) \\
&< \quad 1 - \delta. \qquad\qquad \text{(Using that } \delta < \tfrac{1}{2} \text{ and } n \geq 1) \quad (76)
\end{aligned}
$$

The rest of the proof is identical.

## G   Implementation details and additional empirical results

In this section, we present the implementation details of our simulations (Supplementary Materials G.1 and G.1.1), give additional plots for the simulation in Section 5 (Supplementary Material G.2), and additional simulations that use weighted-selection risk as the fairness metric or vary the amount of noise in the data (Supplementary Materials G.3 and G.3.2)

**Code.** The code for all simulations is available at `https://github.com/AnayMehrotra/FairRankingWithNoisyAttributes`.

## G.1 Implementation details

In this section, we give implementation details of our algorithm and baselines.

- **NResilient**: We implement **NResilient** in Python 3 and use the Gurobi optimization library to solve the linear program in Step 1 of Algorithm 1. We state complete pesudocode of **NResilient**'s implementation as Algorithm 2.
- **SJ**: This is [61]'s algorithm. **SJ** (1) solves a linear program whose objective encodes the utility of the ranking and whose constraints capture the fairness constraints, and (2) decomposes the solution as a convex combination of the rankings, and uses this convex combination to generate rankings (see [61, Section 3.4]).
    - More precisely, [61]'s approach works for any linear constraint on the ranking (see the last equation in [61, Section 3.3]). For instance, as noted in [61, Section 3.3], their approach can satisfy multiple constraints of the form: Given any vectors $f \in \mathbb{R}^m$, $g \in \mathbb{R}^n$, and $h \in \mathbb{R}$, require the ranking $R \in \{0,1\}^{m \times n}$ to satisfy $f^\top R g = h$. By introducing a class variable $s$ with constraint $s \geq 0$, their approach extends to constraints of the form

    $$f^\top R g \leq h.$$

    These are sufficient to encode the constraint in Definition 2.2: For any $k$ and $\ell$, define

    $$\begin{aligned} \forall i \in [m], \quad & f_i = \mathbb{I}[i \in G_\ell], \\ \forall j \in [n], \quad & g_j = \mathbb{I}[j \leq k], \\ & h = U_{k\ell}. \end{aligned}$$

    The constraint $f^\top R g \leq h$ with the above values is equivalent to the upper bound specified by $U_{k\ell}$ in Definition 3.2. Repeating this construction for each $k$ and $\ell$, we get a set of constraints that capture the fairness constraints specified by $U$.

    [61] do not provide an implementation of **SJ** and we implement **SJ** in Python3: We (1) construct an optimization program as defined above, (2) use the Gurobi optimization library to solve the linear program constructed by [61], and (3) use the code available at `https://github.com/jfinkels/birkhoff` to compute the Birkhoff-von Neumann decomposition of the solution ([61] also use the same code to compute the decomposition, see [61, Section 3.4]).
- **CSV**: This is the greedy algorithm from [18, Theorem 3.3]. [18] do not provide an implementation of **CSV**, we implement their algorithm in Python3 with NumPy.
- **GAK**: This is the Det-Greedy algorithm of [27]. [27] do not provide an implementation of **GAK**, we implement **GAK** in Python3 with NumPy.
- **MC** : This first uses the algorithm of [48] to compute a subset $S$ and then selects a ranking of these items that maximize the utility (in the simulations this amounts to sorting items by $w_i$). We used the implementation of [48]'s algorithm available at `https://github.com/AnayMehrotra/Noisy-Fair-Subset-Selection` and use Python3's in-built sorting function to generate the ranking. [48]'s algorithm takes $P$ and parameters $U$ specifying upper bound constraints as input.
- **Uncons**: This is the baseline that outputs the ranking with the maximum utility. In the simulation, this amounts to sorting all items in decreasing order of $w_i$ and outputting the ranking with the first $n$ items (in that order). We implement **Uncons** in Python3 with NumPy.

**Computational resources used.** All simulations were run on a `t3.xlarge` instance with 4 vCPUs and 16Gb RAM, on Amazon's Elastic Compute Cloud (EC2).

### G.1.1 Pre-processing details of the simulation with image data

In this section, we present additional preprocessing details to estimate $\widehat{P}$ in the simulation with the Occupations dataset presented in Section 5.

---

**Algorithm 2** Pseudo-code of the implementation of **NResilient**

---

**Require:** Matrices $P \in [0,1]^{m \times p}$, $W \in \mathbb{R}^{m \times n}_{\geq 0}$, $U \in \mathbb{R}^{n \times p}$
**Ensure:** A ranking $R \in \mathcal{R}$
**Parameters:** Constant $c > 1$, failure probability $\delta \in (0,1]$, and for each $k \in [n]$, relaxation parameter $\gamma_k > 0$

1: Compute a solution $R_F$ to the standard linear programming relaxation of Program (72)
  ▷ In the implementation, we use the Gurobi optimization library in Python 3 to compute $R_F$

2: Compute rankings $R_1, R_2, \ldots, R_\Delta \in \mathcal{R}$ and coefficients $\alpha_1 \geq \alpha_2 \geq \cdots \geq \alpha_\Delta \in [0,1]$ such that

$$R_F = \sum_{i=1}^{\Delta} \alpha_i R_i.$$

  ▷ In the implementation, we use the code available at `https://github.com/jfinkels/birkhoff` to compute this decomposition. This code implements an algorithm to compute the Birkhoff-von Neumann decomposition. The value of $\Delta$ does not need to be specified: It is the number of rankings output by the algorithm to compute the Birkhoff-von Neumann decomposition.

3: Construct matchings $M_1, \ldots, M_\Delta$ corresponding to each ranking $R_1, \ldots, R_\Delta$ such that, for each $t \in [\Delta]$, $M_t$ has an edge between item $i$ and position $j$ if $i$ appears in position $j$ in $R_t$

4: Initialize $N_1 = M_i$
5: **for** $t = 1, 2, ; \Delta - 1$ **do**
6:   $N_{t+1} = \mathbf{Merge}(\alpha_{t+1}, M_{t+1}, \sum_{\ell=1}^{t} \alpha_\ell, N_t)$
7: **end for**

8: Construct a ranking $R$ corresponding to $N_\Delta$: Item $i$ appears at position $j$ in $R$, if and only if, $i$ and $j$ are matched in $N_\Delta$
9: **return** $R$

---

---

**Algorithm 3** Merge procedure used by Algorithm 2

---

**Require:** Numbers $0 < \alpha, \beta \leq 1$ and matchings $M$ and $N$
**Ensure:** A matching $K$
**Parameters:** A constant $t$ (Set to $t := 100$ in the implementation)

1: **while** $M \neq N$ **do**
2:   $P = \mathbf{getPaths}(M, N, t)$
3:   $P' = \mathbf{getPaths}(M, N, t)$
4:   Set $\rho := \frac{t-1}{|P|}$ and $\sigma := \frac{t}{|P'|}$, and $p = \frac{\beta\sigma}{a\rho + \beta\sigma}$
5:   Draw variables $v, u$ u.a.r. from $[0,1]$
6:   **if** $u \leq \frac{\beta\sigma}{\alpha\rho + \beta\sigma}$ **then**
7:     Draw $i$ u.a.r. from $[|P|]$ and set $M = M \Delta P_i$
8:   **else**
9:     Draw $i$ u.a.r. from $[|P'|]$ and set $N = N \Delta P'_i$
10:   **end if**
11: **end while**
12: **return** $K := M$

---

**Estimating** $\widehat{P}$. We begin by removing all images with gender label NA; this leaves 5,825 images (out of 9600). On the remaining images, we use an off-the-shelf face-detector [1] to extract the faces of the people from the images and remove all images where the face-detector did not detect a face; this leaves 4,494 the images. We use a CNN-based gender classifier [59] on the detected faces to predict the apparent gender of the depicted individuals. For each image $i$, the classifier outputs a gender (coded as male and female) and an uncalibrated confidence score $c_i \in [0,1]$. We take the set of uncalibrated confidence scores $\{c_i \in [0,1]\}_i$ and calibrate them by first binning them, then computing the distribution of gender labels (provided in the dataset) for each bin. For each image $i$, we set $\widehat{P}_{i1}$ (respectively $\widehat{P}_{i2}$) equal to the fraction of images in the same bin as $i$ whose gender label is female (respectively male). We perform this calibration once and on all occupations and, then, use it for a subset of occupations.

**Algorithm 4** getPaths procedure used by Algorithm 3

---

**Require:** Matchings $M$ and $N$ and a parameter $t \geq 1$
**Ensure:** A set of paths $P$
 1: Set $P = \emptyset$
 2: **if** $|M \triangle N| \leq 2t$ **then**
 3:     Construct $t$ paths $p_1, \ldots, p_t$, where $p_i = M \triangle N$ for each $i$
 4:     Let $N \setminus M := \{v_1, \ldots, v_n\}$
 5:     For each $i \in [t]$, remove $v_i$ from $p_i$
 6:     Set $P := \{p_1, \ldots, p_t\}$
 7: **else if** $M \triangle N$ is a path **then**
 8:     Let the path formed by $M \triangle N$ be $(v_1, \ldots, v_n)$
 9:     **for** $j = 1, 2, \ldots, t+1$ **do**
10:         If $v_1 \in N$ set $\ell = 1$ else set $\ell = 0$
11:         Set $D := \{v_{\ell + 2tk} : k \in \mathbb{N}, \ \ell + 2tk \leq |M \triangle N|\}$
12:         Set $P = P \cup \{(M \triangle N) \setminus D\}$
13:     **end for**
14: **else**                                                      $\triangleright$ Here, $M \triangle N$ is a cycle
15:     Let the cycle formed by $M \triangle N$ be $(v_1, \ldots, v_n)$
16:     **for** $i = 1, 2, \ldots, |M \triangle N|$ **do**
17:         If $v_i \in M$: **continue**
18:         $S := \{v_{(i+j)\%|M \triangle N|} : j = 0, 1, \ldots, 2t - 1\}$
19:         Set $P = P \cup S$
20:     **end for**
21: **end if**
22: **return** $K := M$

---

### G.2 Further discussion and plots for simulations from Section 5

**Illustrating the fairness vs. utility trade-off.** In our empirical results, we use fairness metrics such as weighted risk-difference (Section 5) and weighted selection-lift (Supplementary Material G.3) to measure the algorithms' *achieved* fairness. We do not use the parameter $\phi$ to measure fairness because the output of algorithms may have lower fairness than specified by $\phi$. Figures 2, 8 and 10 plot utility vs. weighted risk-difference and Figures 14(b), 15(b) and 16(b) plot utility vs. weighted selection-lift (SL) for the simulations in Section 5. They show that **NResilient** better or similar (up to standard errors) achieved fairness vs utility trade-off compared to baselines. For example, in Figure 15(b), to achieve SL= 0.55 use Figure 15(a) to choose $\phi = 1.19$ for **NResilient** and $\phi = 1.15$ for **CSV** or **SJ**. For these values of $\phi$, **NResilient** has 2% higher utility than **CSV** and **SJ**.

**Comparison to baseline which has access to *accurate* protected attributes.** Let **Clean-Fair** be the algorithm that, given utilities and accurate protected attributes, outputs the ranking with the maximum utility subject to satisfying equal representation constraint. Note that **Clean-Fair** can only be run in the ideal scenario where one has access to accurate protected attributes. We repeated the simulations in Section 5 and, for each of them, also measured the utility and fairness of **Clean-Fair**. We observe that the rankings output by **Clean-Fair** have a RDclose to 1 ($>0.99$), this is expected because **Clean-Fair** has access to the clean protected attributes. We observe that the ranking output by **NResilient** (for any parameter $0 \leq \phi \leq 1$, specifying the fairness constraints for **NResilient**) has a utility that is at most 2%, 10%, and 4% smaller than that the ranking output by **Clean-Fair**.

RD **of Uncons.** **Uncons**'s RDand utility does not vary with $\phi$ because it does not take $\phi$ as input. Note that, **Uncons** also does not take the protected groups or $P$ as input.

**Plots with a small number of iterations.** Figures 11 to 13 present results from simulations in Section 5 with 25, 50, and 100 iterations; compared to 500 or 1000 iterations in Figures 1 to 3. We observe that:

- the error bars for both utility and fairness (w.r.t. RD) are a larger (up to 0.025 compared to at most 0.0125 with 500/1000 iterations).

- the mean utilities and fairness (w.r.t. RD) of all algorithms at all values of $\phi$ are additively within 0.05 of their corresponding values in Figures 1 to 3.

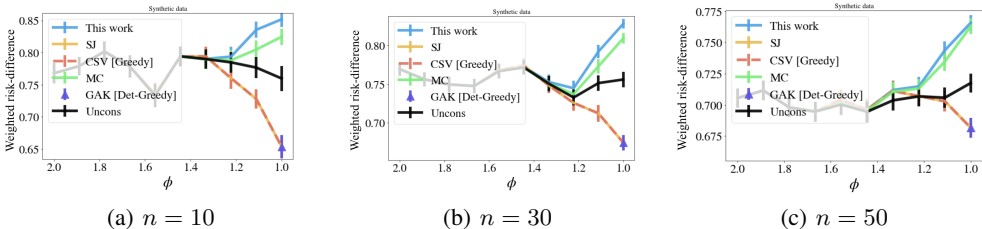

(a) $n = 10$      (b) $n = 30$      (c) $n = 50$

Figure 4: Simulation on synthetic data with different values of $n$. The details appear in Supplementary Material G.2.

Moreover, the relative order of the algorithms with respect to both their fairness (w.r.t. RD) and utility is the same as in Figures 1 to 3 for all $\phi$.

**Plots with different values of $n$.** Figures 4 to 6 plot the RD and utilities (NDCG) with $n \in \{10, 30, 50\}$ in the simulations from Section 5; compared to $n = 25$ in Figures 1 to 3. We observe that the best RD attained by **NResilient** increases with $n$: Increasing $n$ from 10 to 30, increases RD from $0.76$ to $0.85$ with the synthetic data, from $0.75$ to $0.84$ with the real-world image data, and from $0.61$ to $0.71$ with the real-world name data (see Figures 4 to 6). Further, in all simulations, **NResilient**'s maximum RD is 2% to 8% higher than that of the baselines (see Figures 4 to 6). One exception is the simulation with real-world image data and $n = 10$. In this simulation, **NResilient**'s best RD is equal to **GAK**'s best RD. Both of them have $> 6\%$ higher best RD than any other algorithm. (See Figure 5.)

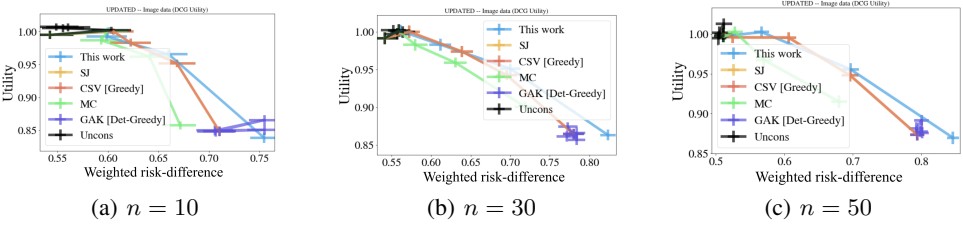

(a) $n = 10$      (b) $n = 30$      (c) $n = 50$

Figure 5: Simulation on image data with different values of $n$. The details appear in Supplementary Material G.2.

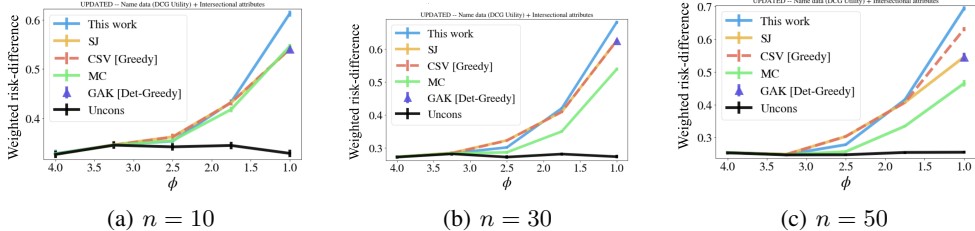

(a) $n = 10$      (b) $n = 30$      (c) $n = 50$

Figure 6: Simulation on real-world name data with different values of $n$. The details appear in Supplementary Material G.2.

**Empirical results with real-world name dataset and overlapping groups.** We present a variant of the simulation in Figure 3 that considers four overlapping groups: The sets of all women players, all male players, all non-White players, and all White players. In contrast, the simulation in Figure 3 uses four disjoint groups: The sets of non-White non-men players, White non-men players, non-White men players, and White men players.

*Setup.* The same setup as the simulation in Figure 3. The only difference is in estimating $\widehat{P}$: For each $i$, we estimate $\widehat{P}$ as:

$$\widehat{P}_{i,\text{women}} = p_{\text{women}}(i), \qquad\qquad \widehat{P}_{i,\text{men}} = 1 - p_{\text{women}}(i),$$
$$\widehat{P}_{i,\text{non-white}} = p_{\text{non-white}}(i), \qquad\qquad \widehat{P}_{i,\text{white}} = 1 - p_{\text{non-white}}(i).$$

Where $p_{\text{women}}(i)$ and $p_{\text{non-white}}(i)$ are values output by Genderize API and EthniColr Library that estimate the probability that player $i$ is labeled as a women and non-white respectively. (**CSV** and **GAK** require protected groups to be disjoint and, hence, are not applicable to this simulation.)

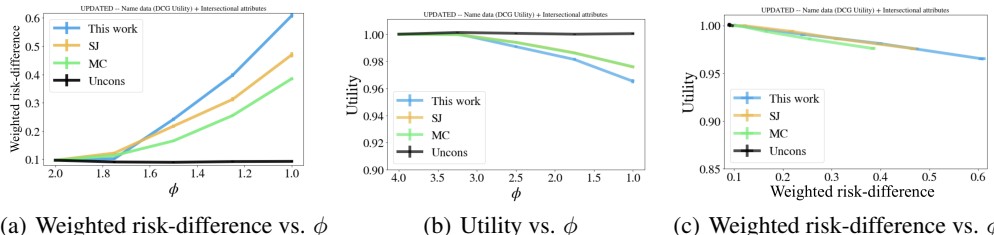

(a) Weighted risk-difference vs. $\phi$      (b) Utility vs. $\phi$      (c) Weighted risk-difference vs. $\phi$

Figure 7: Simulation with the real-world name data and overlapping groups. The details appear in Supplementary Material G.2.

*Observations.* Figure 7 plots RD and utilities (NDCG) averaged over 200 iterations. The results are similar to the corresponding simulation on the same data with disjoint groups. In particular, compared to other baselines, **NResilient** achieves the highest RD. The maximum RD of **NResilient** in this simulation is 0.64 compared to 0.67 in Figure 3. **SJ** achieves the next highest RD followed by **MC** as in Figure 3.

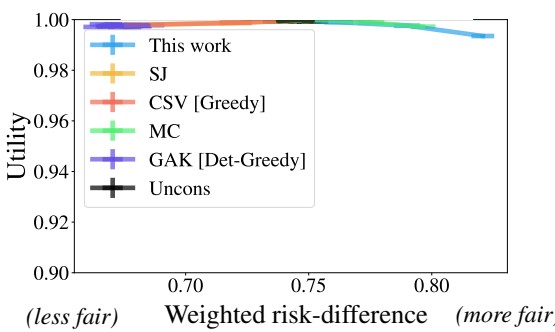

Figure 8: *Synthetic Data: Nonuniform Error Rate.* This simulation considers synthetic data where imputed socially-salient attributes have a higher false-discovery rate for one group compared to the other. We vary the fairness constraint from $\phi$ from 2 (less fair) to 1 (more fair) and observe the weighted risk-difference (weighted risk-difference) of different algorithms. The $y$-axis plots utility and $x$-axis shows weighted risk-difference (*Note that the values decrease toward the right*). Error-bars denote the error of the mean.

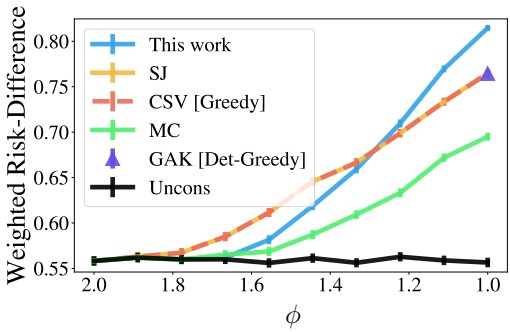

Figure 9: *Real-world image data.* This simulation considers images-search results which are known to overrepresent the stereotypical gender [38]. Given relevant *non-gender labeled* images and their utilities, our goal is to generate a high-utility gender-balanced ranking. We estimate $P$ using an off-the-shelf ML-classifier and vary $\phi$ from $p = 2$ (less fair) to 1 (more fair). In the first subfigure, the $y$-axis plots weighted risk-difference and $x$-axis shows $\phi$ (*Note that the values decrease toward the right*). Error bars show the error of the mean.

## G.3    Additional empirical results

### G.3.1    Empirical results with weighted selection-lift

In this section, we present empirical results with the weighted selection-lift fairness metric (Figures 14 to 16). Weighted selection-lift is a position-weighted version of the standard selection-difference metric. Like weighted risk-difference, it also measures the extent to which a ranking violates equal representation. The weighted selection-lift of a ranking $R$ is:

$$\frac{1}{Z} \sum_{k=5,10,\dots} \frac{1}{\log k} \min_{\ell,q\in[p]} \left| \frac{\sum_{i\in G_\ell,\, j\in[k]} R_{ij}}{\sum_{i\in G_q,\, j\in[k]} R_{ij}} \right|,$$

Where $G$ denotes the ground-truth protected groups and $Z$ is a constant so that RD has range $[0,1]$. Here, a value of 1 is most fair and 0 is least fair.

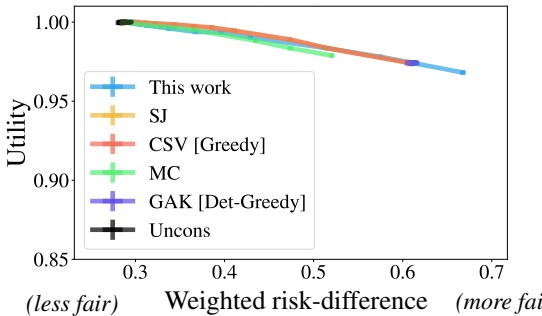

Figure 10: *Real-World Name Data: Intersectional Attributes.* This simulation considers two socially-salient attributes, gender and race, and our goal is to ensure equal representation across the four *intersectional* socially-salient groups (non-White non-men, White non-men, non-White men, and White men). We estimate $P$ from the full names using public APIs and libraries. We vary $\phi$ from $p = 4$ (less fair) to 1 (more fair) and observe weighted risk-difference of all algorithms. The $y$-axis plots utility and $x$-axis shows weighted risk-difference (*Note that the values decrease toward the right*). Error bars represent the error of the mean.

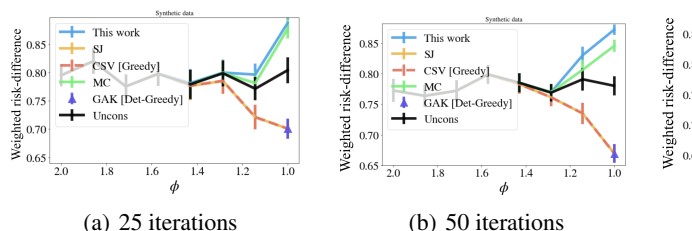

|     (a) 25 iterations     |     (b) 50 iterations     |     (c) 100 iterations     |
| --- | --- | --- |

Figure 11: Simulations on synthetic data from Section 5 with 25, 50, and 100 iterations. The details appear in Supplementary Material G.2.

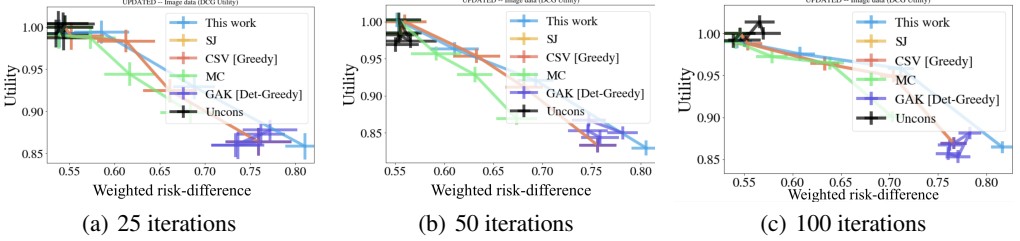

|     (a) 25 iterations     |     (b) 50 iterations     |     (c) 100 iterations     |
| --- | --- | --- |

Figure 12: Simulations on image data from Section 5 with 25, 50, and 100 iterations. The details appear in Supplementary Material G.2.

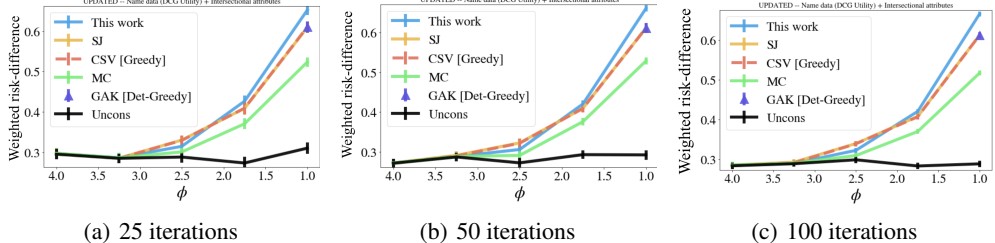

|     (a) 25 iterations     |     (b) 50 iterations     |     (c) 100 iterations     |
| --- | --- | --- |

Figure 13: Simulations on real-world name data from Section 5 with 25, 50, and 100 iterations. The details appear in Supplementary Material G.2.

### G.3.2   Empirical results with varying amount of noise

In this section, we present a simulation which uses the randomized response mechanism to generate noisy protected attributes and compares the performance of algorithms at varying noise levels.

**Data.** We use the Occupation images data [15]. We refer the reader to Section 5 for a discussion of the data.

**Setup.** We fix equal representation constraints ($\phi = 1$) and consider the same protected groups as the simulation with the same data in Section 5. We vary the noise level $0 \leq \eta \leq \frac{1}{2}$. For each $\eta$, we construct noisy attributes by mislabeling true protected attribute with probability $\eta$. Here, $P$ is specified by $\eta$ as explained in Remark A.1. Specifically, if $N_1$ and $N_2$ be the noisy versions of true

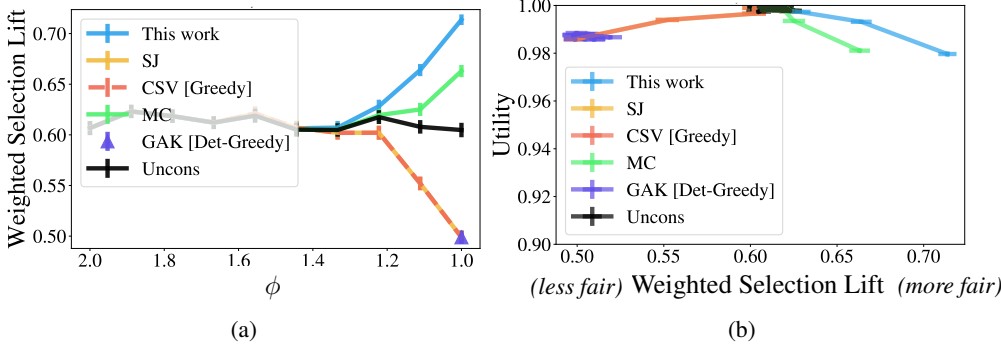

(a)                                       (b)

Figure 14: *Synthetic Data (Weighted Selection Lift): Nonuniform Error Rate.* This simulation considers synthetic data where imputed socially-salient attributes have a higher false-discovery rate for one group compared to the other. We vary the fairness constraint from $\phi$ from 2 (less fair) to 1 (more fair) and observe the weighted risk-difference (weighted risk-difference) of different algorithms. In the first sub-figure, the $y$-axis plots weighted selection-lift and $x$-axis shows $\phi$. In the second sub-figure, the $y$-axis plots utility and $x$-axis shows weighted selection-lift. Error bars represent the error of the mean.

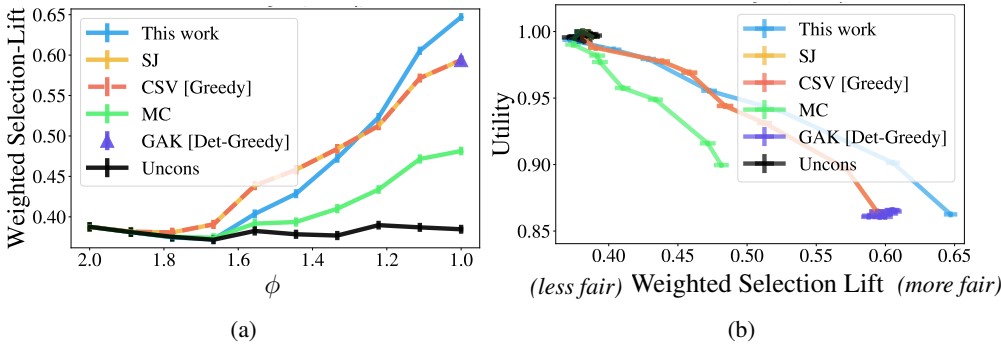

(a)                                       (b)

Figure 15: *Real-world image data.* This simulation considers images-search results which are known to overrepresent the stereotypical gender [38]. Given relevant *non-gender labeled* images and their utilities, our goal is to generate a high-utility gender-balanced ranking. We estimate $P$ using an off-the-shelf ML-classifier and vary $\phi$ from $p = 2$ (less fair) to 1 (more fair). In the first sub-figure, the $y$-axis plots weighted selection-lift and $x$-axis shows $\phi$. In the second sub-figure, the $y$-axis plots utility and $x$-axis shows weighted selection-lift. Error bars represent the error of the mean.

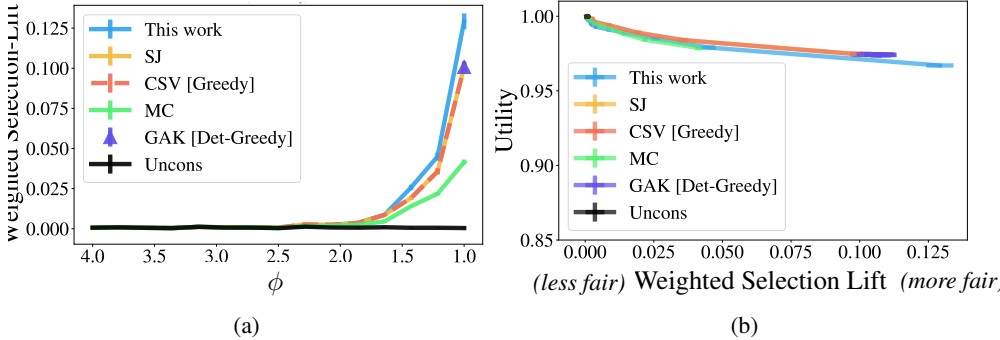

(a)                                       (b)

Figure 16: *Real-World Name Data: Intersectional Attributes.* This simulation considers two socially-salient attributes, gender and race, and our goal is to ensure equal representation across the four *intersectional* socially-salient groups (non-White non-men, White non-men, non-White men, and White men). We estimate $P$ from the full names using public APIs and libraries. We vary $\phi$ from $p = 4$ (less fair) to 1 (more fair) and observe RD of all algorithms. In the first sub-figure, the $y$-axis plots weighted selection-lift and $x$-axis shows $\phi$. In the second sub-figure, the $y$-axis plots utility and $x$-axis shows weighted selection-lift. Error bars represent the error of the mean.

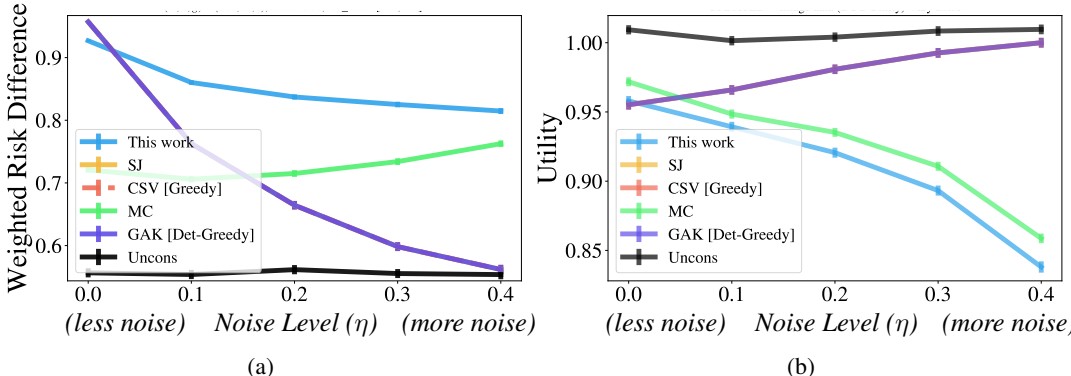

Figure 17: *Simulation varying the amount of noise.* In this simulation, we use the Occupation's images data [15] and generate noisy protected attributes using the randomized response mechanism, with parameter $\eta$. We vary the amount of noise added from $\eta = 0$ (no noise) to $\eta = 0.4$ (large noise) and compare the performance of different algorithms. The $y$-axis plots RD and $x$-axis plots $\eta$. We present the key observations in the paragraph above the figure. Error-bars denote the error of the mean.

protected groups $G_1$ and $G_2$ (corresponding to the "flipped" protected attributes), then we set: For each item $i \in N_1$,

$$\widehat{P}_{i1} = (1 - \eta) \cdot \frac{|G_1|}{|N_1|} \quad \text{and} \quad \widehat{P}_{i2} = 1 - \widehat{P}_{i1}.$$

For items in $N_2$, replace $\widehat{P}_{i1}$, $\widehat{P}_{i2}$, $G_1$, and $N_1$ with $\widehat{P}_{i2}$, $\widehat{P}_{i1}$, $G_2$, and $N_2$. We do not have access to $G_1$ (and, hence, $|G_1|$), and in the above expression we estimate $|G_1|$ by $\alpha_1 := \frac{(1-\eta) \cdot}{1-2\eta} \cdot ((1 - \eta) |N_1| - \eta |N_2|)$. This is because $\alpha_1$ can be shown to be concentrated around $|G_1|$.

Like the simulations in Section 5, **CSV**, **GAK**, and **SJ** are given the noisy attributes (as they require) and **NResilient** and **MC** are given $\widehat{P}$ (computed above).

**Observations.** See Figure 17 for RD and utilities (NDCG) averaged over 100 iterations. We observe that for each $\eta \geq 0.1$, **NResilient**'s RD is $>6.8\%$ better than any baseline (Figure 17(a)) and its utility is $<3\%$ smaller than the baseline (**CSV**) with best RD (Figure 17(b)). At $\eta = 0$, **NResilient** 3.3% lower RD than **CSV**, **GAK**, and **SJ** and the same utility as them.

Note that in Figures 17(a) and 17(b) the plots of **CSV**, **GAK**, and **SJ** overlap. This is consistent with the other simulations where **CSV**, **GAK**, and **SJ** have the same RD and utility at $\phi = 1$.

### G.3.3 Empirical results with proportional representation constraints

In this section, we present variants of the simulations in Figures 1 to 3 that use proportional representation fairness constraints. To measure the deviation of a ranking from proportional representation, we consider an adaptation of weighted risk-difference metric, Prop-RD. Prop-RD of a ranking $R$ is

$$1 - \frac{1}{Z} \sum_{k=5,10,\dots} \frac{1}{\log k} \max_{\ell,q \in [p]} \left| \frac{n}{|G_\ell|} \cdot \sum_{i \in G_\ell, j \in [k]} R_{ij} - \frac{n}{|G_q|} \cdot \sum_{i \in G_q, j \in [k]} R_{ij} \right|. \quad (77)$$

Where $G$ denotes the ground-truth protected groups and $Z$ is a constant so that RD has range $[0, 1]$. Here, Prop-RD$= 1$ is most fair and Prop-RD$= 0$ is least fair.

**Setup.** The setup of the simulations is identical to the simulations in Figures 1 to 3 except that, given $\phi \geq 1$, the upper bounds are set to $U_{k\ell} := \phi \cdot \frac{|G_\ell|}{n} \cdot k$ for each $k \in [n]$ and $\ell \in [p]$.

**Observations.** Figure 18 presents the values of Prop-RD averaged over 50 iterations. We observe that, relative to the baselines, **NResilient**'s performance is similar to Figures 1 to 3. In particular, in all simulations, **NResilient** achieves a higher value of the fairness metric than any baselines, as in Figures 1 to 3. Further, in the simulation with the real-world image data, **NResilient** has a better

fairness-utility trade-off than all baselines, as in Figure 2. One difference is that, with the synthetic data, the value of the fairness metric achieved by **NResilient** can be non-monotonous in $\phi$, whereas it is increasing in $\phi$ in Figure 1 (see Figure 18).

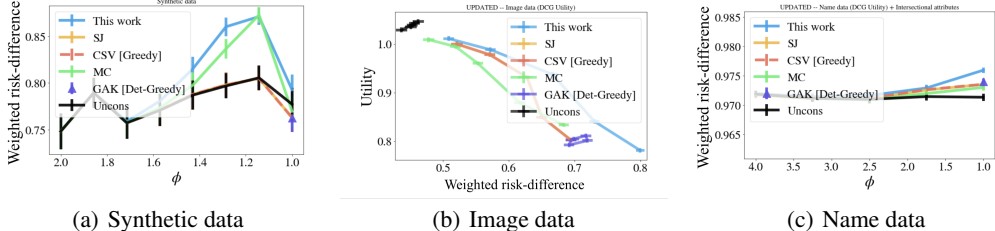

| (a) Synthetic data | (b) Image data | (c) Name data |

Figure 18: Simulations with proportional representation constraints and variant of RD for proportional representation constraints. The details appear in Supplementary Material G.3.3.

### G.3.4 Empirical results with varying false-discovery rates

In this section, we present a variant of the simulation in Figure 1. This simulation varies the difference in false-discovery rates (FDRs) of the attributes inferred from $\widehat{P}$ for the groups.

**Synthetic data.** We generate utilities $w_1, w_2, \ldots, w_m$ by drawing $w_i$ is independently from the uniform distribution over $[0, 1]$ for each $1 \leq i \leq m$. Fix $\mu_1 := 1 - \frac{1}{20}$, $\mu_2 := \frac{1}{2} - \frac{1}{20}$, $\sigma_1 := \frac{1}{50}$, and $\sigma_2 := \frac{1}{10}$. Given a parameter $0 \leq \tau \leq 1$, controlling the FDRs of the two groups, we construct $P$ as follows: For each $i$, with probability 0.6, $P_{i1}$ is drawn from

$$\mathcal{N}\left((1 - \tau) \cdot \mu_1 + \tau, (1 - \tau) \cdot \sigma_1\right)$$

and otherwise $P_{i1}$ is drawn

$$\mathcal{N}\left((1 - \tau) \cdot \mu_2 + \tau \cdot 0, (1 - \tau) \cdot \sigma_2\right).$$

We set $P_{i2} := 1 - P_{i1}$ for each $i$. This ensures that, with high probability,

$$|G_1| = 0.6n \pm o_n(1) \quad \text{and} \quad |G_2| = 0.4n \pm o_n(1)$$

Let $\mathrm{FDR}_1(\tau)$ and $\mathrm{FDR}_2(\tau)$ be the false-positive rates of attributes inferred from $P$ on groups $G_1$ and $G_2$ for a given $\tau$. Let $\Delta(\tau) := \mathrm{FDR}_2(\tau) - \mathrm{FDR}_2(\tau)$. We have $\Delta(0) = 0.4$, $\Delta$ decreases with $\tau$, and $\Delta(1) = 0$.

**Setup.** The setup is identical the simulation in Figure 1 except that we use the above synthetic data. We consider three values $\tau_1, \tau_2$, and $\tau_3$ of $\tau$ such that $\Delta(\tau_1) = 20\%$, $\Delta(\tau_2) = 20\%$, and $\Delta(\tau_3) = 20\%$. (For comparison, the FDRs of the two groups differ by 30% for the simulation in Figure 1.)

**Observations.** See Figure 19 for RD averaged over 50 iterations. We observe that the difference between the best RD of **NResilient** those of **SJ** and **CSV** decreases with $\Delta$: At $\Delta = 20\%, 10\%, 5\%$, **NResilient**'s RD is 12%, 4%, and 0% higher than **SJ**'s and **CSV**'s RD respectively.

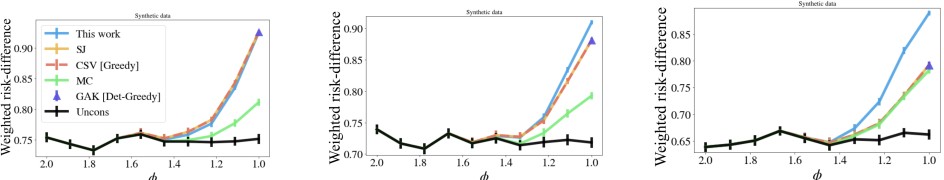

(a) Minority group's FDR is 5% (b) Minority group's FDR is 10% (c) Minority group's FDR is 20% smaller than the majority's FDR    smaller than the majority's FDR    smaller than the majority's FDR

Figure 19: Simulation on synthetic data where the minority group's FDR is $\Delta = 5\%, 10\%, 20\%$ smaller than the majority's FDR. The details appear in Supplementary Material G.3.4.

### G.3.5   Empirical results with a large number of groups

In this section, we present simulations on synthetic datasets with 4, 6, 8, and 10 protected groups.

For simplicity, all groups have equal sizes. In particular, we construct variants of the synthetic dataset in Section 5 so that the false-discovery rates of the attributes inferred from the matrix $\widehat{P}$ on the groups are spread at equal intervals in the interval $[10\%, 40\%]$. For instance, for $p = 4$, the FDRs of the four groups are $10\%$, $20\%$, $30\%$, and $40\%$ respectively.

**Synthetic data.**   We generate utilities $w_1, w_2, \ldots, w_m$ by drawing $w_i$ is independently from the uniform distribution over $[0, 1]$ for each $1 \le i \le m$. Fix $\mu_1 := 1 - \frac{1}{20}$, $\mu_2 := \frac{1}{2} + \frac{1}{20}$, $\sigma_1 := \frac{1}{50}$, and $\sigma_2 := \frac{1}{10}$. For each group $G_\ell$, there is a parameter $0 \le \tau_\ell \le 1$, that controls the corresponding FDR. We construct $P$ as follows: For each $\ell$ and $i \in G_\ell$,

- $P_{i1}$ is iid from $\mathcal{N}\left((1 - \tau) \cdot \mu_1 + \tau \cdot \mu_2, (1 - \tau) \cdot \sigma_1 + \tau \cdot \sigma_2\right).$
- $P_{iz} := 1 - P_{i1}$ where $z$ is drawn uniformly at random from $[p] \backslash \{\ell\}$
- $P_{ij} = 0$ for each $j \in [p] \backslash \{\ell, z\}.$

Let $\Delta(\tau)$ be the FDR of $G_\ell$ when $\tau_\ell = \tau$. (By construction, this function is independent of $\ell$.)

**Setup.**   The setup is identical the simulation in Figure 1 except that we use the above synthetic data to generate $w$ and $P$. We vary $p \in \{4, 6, 8, 10\}$. For each $p$, we fix $\tau_\ell$ such that $\Delta(\tau_\ell) := 10\% + \frac{\ell-1}{p-1} \cdot 30\%$ (for each $\ell \in [p]$).

**Observation.**   Figure 20 plots RD averaged over 50 iterations. We observe that **NResilient** has a better or similar (within $1\%$) utility and RD compared to the best performing baseline at all values of $\phi$.

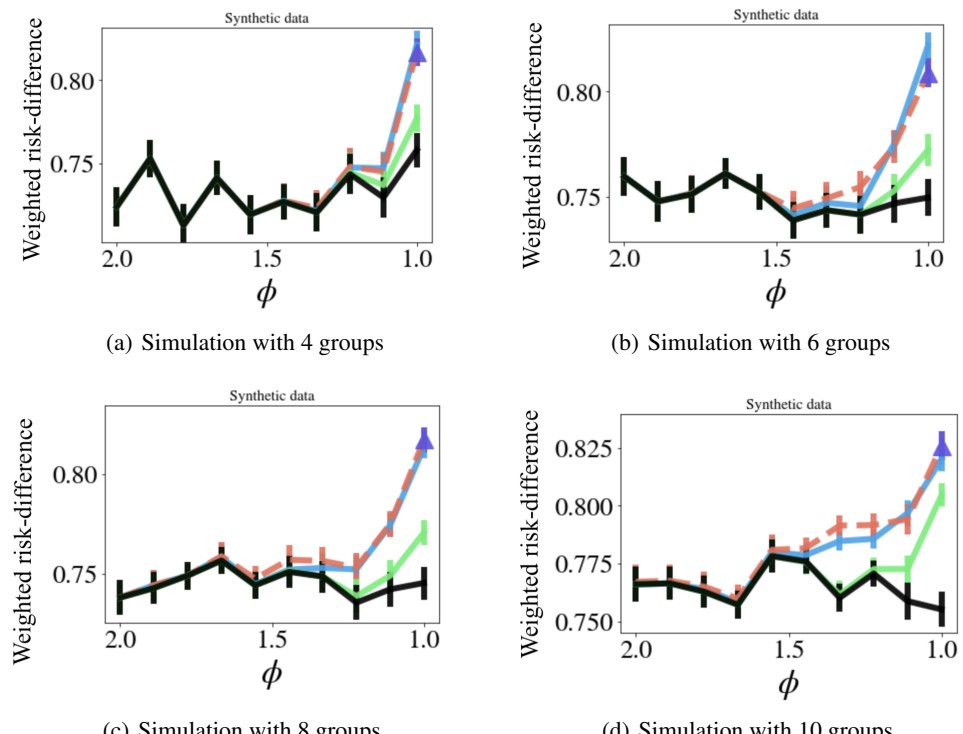

(a) Simulation with 4 groups

(b) Simulation with 6 groups

(c) Simulation with 8 groups

(d) Simulation with 10 groups

Figure 20: Simulation on synthetic data with four, six, eight, and ten groups. The details appear in Supplementary Material G.3.5.