# OpenReview forum: "Fair Ranking with Noisy Protected Attributes"
_NeurIPS.cc/2022/Conference — NeurIPS 2022 Accept_

### Official Review · Reviewer_TTow · 2022-06-29

**Rating:** 8
**Confidence:** 4
**Soundness:** 4 excellent
**Presentation:** 4 excellent
**Contribution:** 4 excellent

**Summary:**

Given:
1. $m$ items,
2. $G_i$ - Possible overlapping sensitive groups to which items belong,
3. $W_{ij}$ - The utility of ranking item $i$ at position $j$, and
4. $U_{ij}$ - The maximum number of items from sensitive group $G_i$ that can appear in the first $j$ positions in the ranking.

The goal of fair ranking is to find a ranking of top-$n$ items such that the utility with respect to $W_{ij}$s is maximised under the constraint that the ranking obeys the limits imposed by $U_{ij}$s.

This paper studies the fair ranking problem under a setting where groups $G_i$ are not deterministically known. Instead, the algorithm only observes $P_{ij}$, the probability of item $i$ belonging to group $G_j$.

The authors begin by defining an $(\epsilon, \delta)$-fairness constraint for ranking. For $G_i$s sampled from $P_{ij}$ values, this constraint requires the ranking to satisfy the upper bound constraint defined by $U_{ij}$ values (relaxed by a factor of $1 + \epsilon$) with high probability.

They show that directly incorporating this constraint leads to a NP-hard problem. On the other hand, naively replacing the high probability constraint with an expectation constraint leads to feasibility issues.

The main contribution in the paper is a variant of the expectation constraint. The authors show that a ranking satisfying the $(\epsilon, \delta)$-constraint also satisfies the new expectation constraint and vice versa.

Because the expectation constraint is linear, this allows them to write the problem as an integer program with linear constraints. This is then relaxed to a linear program to get the final algorithm.

Besides providing a fairness guarantee (both for the solution of the integer program and the corresponding linear program), the authors also show that their relaxed expectation constraint is tight in the sense that any further relaxation makes it impossible to find a clustering that satisfies the original $(\epsilon, \delta)$-constraint.

Experiments on synthetic and real data highlight the utility and limitations of the proposed work.


**Questions:**

This question has no impact on my assessment. I just want to understand some aspects about the fairness notion used in the paper. Why would one like to have an upper bound on the number of occurrences from a particular sensitive group instead of having a lower bound? I understand that in some cases (like equal representation), having an upper bound indirectly also places a lower bound, but why is upper bound a good idea in general? Are there any unintended negative consequences of such a choice?

**Limitations:**

The authors have done a fantastic job of discussing potential limitations of their work.

**Strengths And Weaknesses:**

**Originality:** The proposed expectation constraint is novel. The authors have clearly distinguished their contributions from the existing literature.

**Quality:** I have not read the proofs in detail but the outline looks sound to me. The authors have answered a number of interesting questions related to their framework (upper bounds and lower bound, problems with naive solutions) and have supported their claims with detailed experiments.

**Clarity:** Overall, the paper is very clearly written and easy to follow. A few minor issues:
1. L185-186 - "The issue is that ... for each position" - My understanding is that for smaller values of k (position in the ranking), it is harder to satisfy the upper bound constraint and a larger $\rho$ is needed. Is this correct? Adding a line on the intuition here will be helpful.
2. Adding a high level idea behind the dependent-rounding algorithm from reference [16] in the paper will improve readability.
3. The organization of Section 4 can be improved. For example, the description about estimating $\hat{P}$ in L292 becomes clear only near L326, even though $\hat{P}$ is referenced several time in between as well.

**Significance:** Because ranking is an important practical problem and the authors consider a fairly general setup, the results are indeed significant in my opinion. The authors also point out avenues for improvement (mis-calibrated P estimation models, non-independent errors etc.) and the ideas presented in this paper will be helpful in exploring these directions.

---

> ### Author Response · Authors · 2022-08-02
> **Response to Reviewer TTow**
>
> We are glad that you appreciate the practical importance of the problem and results. We have updated the paper to incorporate your suggestions (and uploaded the revision to OpenReview). Below we address your specific questions and outline changes in the paper to incorporate your suggestions.
>
>
> **"[For Proposition 2.3] my understanding is that ... [and, hence, a ] larger $\rho$ is needed. Is this correct?"** Yes, that is correct.  In the revised submission, we include this intuition after Proposition 2.3, in Lines 195-196 of Section 2.
>
> **"Adding a high level idea behind the dependent-rounding algorithm from reference [16] in the paper will improve readability."** Thanks for the suggestion, we added a discussion on the high level idea of the dependent-rounding algorithm in Lines 1014-1034 in Supplementary Material D.4 of the revised submission and referenced it from Lines 264-265 in Section 3 of the revised submission.
>
> **"The organization of Section 4 can be improved. For example, the description about estimating $\hat{P}$ in L292 becomes clear only near L326"** Thanks for this suggestion. In the revised submission, we reference the places where details about estimating $\hat{P}$ are presented from "description of estimating $\hat{P}$" (Line 292 of the initial submission and Lines 299-301 of the revised submission). In the final version, we will use the extra space to further improve the organization of Section 4.
>
> **"Why would one like to have an upper bound on the number of occurrences from a particular sensitive group instead of having a lower bound? I understand that in some cases (like equal representation), having an upper bound indirectly also places a lower bound, but why is upper bound a good idea in general? Are there any unintended negative consequences of such a choice?"**
> While both upper bound and lower bound constraints can encode the fairness requirements of interest, the reason we use upper bound constraints instead of lower bound constraints is that the rounding algorithm we use (in Theorem 3.3) requires the constraints to be upper bounds. This, in turn, is because the set of feasible solutions obtained after placing upper bound constraints forms a "matroid." (The set of feasible solutions obtained by placing lower bound constraints does not form a "matroid" and, as a consequence, does not have the right structure for the above algorithm to work as desired.)
>
> A (potentially negative) consequence of this choice is that, while all feasible solutions (in either case) will satisfy the desired fairness constraints, these two sets may be different for the lower and upper bound cases, causing the utilities of the respective optimal sets chosen by the algorithm to have different utilities.

---

> > ### Comment · Reviewer_TTow · 2022-08-07
> > **Thanks for your response**
> >
> > Thank you for your response. I especially appreciate your explanation for why you use upper bounds instead of lower bounds.

---

### Official Review · Reviewer_9Qwf · 2022-07-12

**Rating:** 7
**Confidence:** 3
**Soundness:** 4 excellent
**Presentation:** 3 good
**Contribution:** 3 good

**Summary:**

This paper studies fair-ranking where, for each item, its group memberships are only known probabilistically.  The authors introduce an approximation algorithm to solve the fair-ranking problem and prove that it approximately satisfies the considered fairness criteria and (subject to fairness) maximizes utility. In experiments, the proposed approach is the most fair of all baselines and achieves the best tradeoff between utility and fairness.


**Questions:**

- In your simulated experiment, you set $\hat{P}$ such that false-discovery is much more likely for minority groups. Is this a reasonable setting? How do the results differ if false-discovery rates are more equal?
- In your experiments, you measure fairness with weighted risk-difference. For what other metrics do you expect your proposed framework to perform similarly well? Are there metrics whether other methods could be more competitive?
- Theoretically and empirically, how do your results change as $n$ grows or shrinks? For lower values of $n$, would your algorithm be as competitive in terms of fairness?
- How would your empirical results compare to other algorithms if there were a larger number of groups? The current experiments only use a small number of groups (e.g., religious groups or ethnicities).


**Limitations:**

- The proposed framework makes a number of assumptions (clearly): the probabilities of item-group membership are known, membership probabilities are independent, and item relevances are known. These assumptions may not be realistic in many cases, and as the authors mention, may adversely affect their proposed framework. However, the authors do perform 2 experiments that directly address these cases, and the experiments show that their algorithm is performant.


**Strengths And Weaknesses:**

This is a clearly written paper with strong theoretical components. The empirical work is relatively good, but it’s hard to know whether the results reflect significant improvements over state of the art. The paper is dense with many details appearing in the supplementary material.

**Originality:** while this work draws on many aspects of other work in fair ranking, the particular setting studied is original.
- This work is the first to study ranking when socially salient attributes are only known probabilistically. The fairness constraints employed in this paper appear in many other pieces on fair ranking. The algorithm proposed is a rounding algorithm that was introduced in reference [16]. However, the algorithm is applied in a new context and the problem setup is novel.
- There has been some work on classification with noisy protected attributes, which may deserve a mention (despite this work being focused on ranking).
- As far as I can tell, the related work is adequately cited.

**Clarity:** overall, this is a very clear paper.
- While the writing is good, I found the introduction to be too abstract, despite references to specific work. Consider including some specific examples in the introduction to ground the discussion.
- A few typos:
    - Line 224: “Constraint (5) _applies_ upper bounds on…”
    - Line 276: “we evaluate our framework’s performance _on_ synthetic…”
    - Line 293 & 294: you have G_1, …, G_2, but I think you mean G_p
    - Line 323: “its” -> “it’s”
    - Line 399: “this framework works _for_ a…”
- Line 312: “MC achieves the best RD” this is right after a sentence that says that NResilient achieves the best RD.
- While the paper is clear, most details are omitted in the main text (there are nearly 30 pages of supplementary material).  I found myself often desiring more detail and examples in various places in the main text, e.g., proof of Proposition 2.3, the last paragraph after Constraint (5), a description of the rounding scheme, and a description of the baselines–to name a few.

**Quality:** from a theoretical standpoint, the paper seems technically rigorous.
- The paper contains a number of theoretical statements related to the fairness and utility of the proposed approach. The supplementary material includes most of the proofs.
- Experiments only measure fairness via weighted risk-difference. I wonder if the results would differ if using a different metric.
- In experiments, it is hard to interpret the significance of the improvement achieved by the proposed algorithm. It would be useful to describe/ground the results in some application, for example: resume ranking.
- Similarly, it seems as though an advantage of the proposed algorithm is that it can achieve higher utility when more fairness is required. For example, in Figure 2, CSV and NResilient are about equal until weighted risk-difference is greater than 0.66. However, it’s unclear whether these levels of fairness would be required or desirable in some application.

**Significance:** this work is a useful piece in the fair-ranking landscape
- Theoretically, the work is interesting, although I am unsure of how the theoretical results are likely to be further improved.  I think that this work may inspire future work to study more fair ranking settings in which socially salient attributes are only known probabilistically
- The problem setup and theoretical proofs are unique, but the algorithm is borrowed from existing work.
- Empirically, it was interesting to see that when groups are sampled, fair algorithms can be worse than unconstrained maximization in terms of fairness. This observation is likely to be further scrutinized.
- In experiments, the proposed framework appears to achieve about the same utility-fairness tradeoff as all competitors in 2 or 3 experiments (Fig4 & 6). However, the proposed algorithm is able to produce rankings that are fairer. It’s difficult to know whether this is significant and likely to be used by practitioners.

---

> ### Author Response · Authors · 2022-08-02
> **Response to Reviewer 9Qwf (1/2)**
>
> *(This is part one of a two part response.)*
>
> It is great that you appreciate the writing and the theoretical rigor of the paper. We have updated the paper to incorporate your suggestions (and uploaded the revision to OpenReview). We answer specific questions and outline the changes in the paper to incorporate your suggestions below.
>
>
> **"work on classification with noisy protected attributes ... may deserve a mention"** Thanks for your suggestion. In the revised submission, we discuss works on classification with noisy protected attributes in Supplementary Material M.
>
> **"Consider including some specific examples in the introduction"** Thanks for the suggestion. We will use the extra space in the final version to include specific examples in the introduction.
>
> **"most details are omitted in the main text" and "I found myself often desiring more detail and examples"** Thanks for the suggestion, we will use the extra page available for the final submission to add additional details and examples. Meanwhile, we added  (1) discussion on why straightforward rounding approaches are insufficient in Lines 1011-1031 in Supplementary Material D.4 of the revised submission and referenced it from Lines 265-266 in Section 3 of the revised submission, and (2) discussion for Proposition 2.3 in Lines 195-196 in Section 2 of the revised submission.
>
>
> **"Experiments only measure fairness via weighted risk-difference. I wonder if the results ... differ .. using a different metric."** In the initial submission, we also have simulations that use a different fairness metric (weighted selection-lift) than weighted risk-difference (Figures 7-9 in Supplementary Material B.4 of both the initial and revised submission). In these simulations, we observed that compared to the baselines, NResilient has 2\% to 5\% higher maximum weighted selection-lift and a similar or better fairness-utility trade-off (Figures 7-9 in Supplementary Material B.4 of both the initial and revised submission).
>
> In the revised submission, repeated the simulations from Figures 1-3 (in Section 4 of both the original and the revised submission) with proportional representation fairness constraints To measure the deviation of a ranking from proportional representation, we consider a variant of weighted risk-difference, Prop-RD (see Equation (77) in Supplementary Material G of the revised version). In the new simulations, we observe that,  relative to the baselines, NResilient's performance is similar to Figures 1-3 (in Section 4 of both the initial and the revised submission):
> - In all simulations, NResilient achieves a higher value of the fairness metric than any baselines, as in Figures 1-3 (in Section 4 of both the initial and the revised submission). (See Figures 11-13 in Supplementary Material G of the revised submission.)
> - With the real-world image data, NResilient has a better fairness-utility trade-off than all baselines, as in Figure 2 (in Section 4 of both the initial and the revised submission). (See Figure 12 in Supplementary Material G of the revised submission.)
>
> One difference is that, with the synthetic data, the value of the fairness metric achieved by NResilient can be non-monotonous in $\phi$, whereas it is increasing in $\phi$ in Figure 1 (in Section 4 of both the initial and the revised submission). (See Figure 11 in Supplementary Material G of the revised submission.) In the revised submission, we include the details from these simulations in Supplementary Material G.
>
> **"in Figure 2, CSV and NResilient are about equal until weighted risk-difference is greater than 0.66" and "it’s unclear whether these levels of fairness would be required or desirable in some application."** High levels of weighted risk-difference may be required in certain applications: For instance, if an online recruiting platform wants to guarantee that the ranking of resumes or candidate profiles they display on their platform satisfies the 80\% rule, then they must set $\phi\leq 1.1$. For any $\phi\leq 1.1$, all baselines have RD at least 0.66 (Figure 5 in Supplementary Material B.3 of both the initial and the revised submission).

---

> > ### Author Response · Authors · 2022-08-02
> > **Response to Reviewer 9Qwf (2/2)**
> >
> > *(This is part two of a two part response.)*
> >
> > **"Is [the choice of false-discovery rates] a reasonable setting? How do the results differ if false-discovery rates are more equal?"** The choice of the false-discovery rate is consistent with the findings of [Buolamwini and Gebru, FAT, 2018] for a commercial image-based gender classifier: [Buolamwini and Gebru, FAT, 2018] observe that the classifier has a 34\% higher false-discovery rate for dark-skinned females than for light-skinned men. In contexts where dark-skinned females form a minority, this is comparable to the choice of 30\% higher false-discovery rate for the minority compared to the majority in the simulation. In the revised submission, we include this comparison to [Buolamwini and Gebru, FAT, 2018]'s observation in Footnote 3 in Section 3.
> >
> > We added variations of the simulation in Figure 1 (in Section 4 of both the initial and the revised submission) which vary the difference in FDRs across groups. In particular, the new simulations vary the difference in the FDRs between the groups, $\Delta$, over $5%, 10%, 20%$. ($\Delta$ is 30\% for the simulation in Figure 1 in Section 4 of both the initial and the revised submission.) We observe that the difference between the best RD of NResilient those of SJ and CSV decreases with $\Delta$: At $\Delta=30\%, 20\%, 10\%,5\%$, NResilient's RD is 18\%, 12\%, 4\%, and 0\% higher than SJ's and CSV's RD respectively.
> >
> > **"Theoretically and empirically, how do your results change as $n$ grows or shrinks? For lower values of $n$, would your algorithm be as competitive in terms of fairness?"** Theorem 3.3 guarantees that with high probability, NResilient outputs a ranking that violates the constraint at the $k$-th position by at most $O(\gamma_k)$. Holding other parameters fixed, $\gamma_k=O(\log(n))$. Thus, as $n$ grows the fairness guarantee loosens at a rate $O(\log(n))$.
> >
> > We repeated the simulations in Figures 1-3 (in Section 4 of the initial and the revised submission) with $n\in \{10, 30, 50\}$. (The simulations in Figures 1-3 themselves fix $n=25$.) We observe the following:
> > - The best RD attained by NResilient increases with $n$: Increasing $n$ from 10 to 30, increases RD from  $0.76$ to 0.85 with the synthetic data, from 0.75 to 0.84 with the real-world image data, and from 0.61 to 0.71 with the real-world name data. (Figures 17-19 in Section K of the  revised submission respectively.)
> > - In all simulations (except the one with real-world image data and $n=10$), NResilient's maximum RD is 2\% to 8\% higher than that of the baselines (Figures 17-19 in Section K of the revised submission).  This is similar to the 2\% to 10\% higher maximum RD achieved with $n=25$ (Figures 1-3 in Section 4 of both the initial and the revised submission).
> > - In the simulation with real-world image data and $n=10$, NResilient's best RD is equal to GAK's best RD. Both of them have $>6\%$ higher best RD than any other algorithm. Figure 18 in Supplementary Material K of the revised submission.)
> >
> > In the revised submission, we include the these results in Supplementary Material K.
> >
> > (Note that the first observation does not contradict the theoretical guarantee because RD aggregates the fairness constraint violation across all positions and NResilient is violates the constraints by a smaller multiplicative amount at later positions. Hence, while increasing  $n$ increases the constraint violation at each position it also adds additional positions where NResilient has a smaller violation. Thus, aggregate violation can reduce with $n$.)
> >
> > **"How would your empirical results compare to other algorithms if there were a larger number of groups?"** We added new simulations with the synthetic data that consider $p=4,6,8,$ and $10$ groups that, for simplicity, have an equal size. These simulations consider the same setup as the simulation in Figure 1 (in Section 4 of both the initial and the revised simulation). For each $p$, the corresponding simulation uses a variant of the synthetic data described in Section 4 (of both the initial and the revised simulation), where the false-discovery rates (FDRs) of the groups are spread at equal intervals in the interval $[10\%, 40\%]$. For instance, for $p=4$, the FDRs of the four groups are 10\%, 20\%, 30\%, and 40\% respectively. We observe that NResilient has a better or similar (within 1\%) utility and fairness (measured by RD) compared to the best performing baseline at all values of $\phi.$ (Figure 16 in Supplementary Material J of the revised submission.) In the revised submission, we include these simulations in Supplementary Material J.

---

> > > ### Comment · Reviewer_9Qwf · 2022-08-05
> > > **Thorough response**
> > >
> > > Thank you for the detailed response, providing further clarifications and additional experimental results.

---

### Official Review · Reviewer_6pGr · 2022-07-12

**Rating:** 6
**Confidence:** 3
**Soundness:** 3 good
**Presentation:** 3 good
**Contribution:** 3 good

**Summary:**


Given a model of limited knowledge about sensitive demographic attributes where each item has a corresponding distribution of probabilities over groups, and where noise is independent for each item, the work introduces a class of fairness constraints which guarantee (with high probability) both utility and (approximate) fairness. Theorem 3.1, their main theoretical result, upper bounds the fairness violation and lower bounds the utility of the ranking solution found using these constraints. Then, they provide a feasible relaxation of these constraints such that existing methods for optimization can be applied, and demonstrate empirical results on two “real” and one synthetic dataset. These results illustrate the relationships between RD (their chosen fairness metric), phi (a parameter to the optimization problem), and total utility.

The problem setting makes the key assumption that the true utilities W are known and true (i.e., the measurement of utility does not worsen for different groups), and that total utility of a final ranking can simply be computed additively (i.e. aside from the fairness constraint, there are no other reasons that certain combinations may be disproportionately better or worse than others). Though strong, these are reasonable assumptions for the specific problem which this paper addresses.

**Questions:**

By iterations, do you mean you ran the simulation procedure 500/1000 times? This seems high? Are individual iterations highly noisy / what do results look like averaged over fewer iterations?

By the end of the paper I’m not sure how to interpret phi; I understand that it’s varied between 1 and p, the total number of groups, but what does it represent? Why is it interesting or notable that the method presented in this work can achieve higher RD for a given value of phi?

**Limitations:**

Yes - the authors are very clear about the specific scope of their work and the assumptions that their models rely on. As discussed above, I would have appreciated some discussion of these assumptions and choices (e.g. the existence of the utility matrix, choice of fairness constraint and associated values), but understand the need for a focused work.

**Strengths And Weaknesses:**

/// Strengths ///

I appreciate the storytelling for why the solution unfolded the way it did, particularly why some initial directions are limited. For example, scaffolding for why existing fair ranking work work fails (70-80), why standard approaches to integer programs fail (199-204), why an initial attempt at a constraint (Constraint 5) fails (211-219), and intuition behind the necessity of pseudo-optimality (243-239).

Figure 2 illustrates pretty good results!

The appendix is thoughtful. I especially enjoyed the discussion of the model for group membership in Appendix A, and the series of baselines (additional metrics, varying noise, perfect access to group membership).

/// Weaknesses ///

Most of my comments here are related to wanting more results/substantiation relative to what was claimed in the paper, not necessarily fatal flaws.

Sensitivity to noise levels: the paper claims their method (NResilient) “has better or similar RD than each baseline at all noise levels” (336-338) — unless I’m drastically misinterpreting figure 10b in the appendix, this seems to be untrue? -> leading me to also wonder at what noise levels do the fairness-utility curves (as shown in the main paper) start looking less nice.

Group membership: The model for group membership is not clear from the main body of the paper, specifically what it means for the model to handle “multiple” sensitive features, or the case where each individual item can belong to “one or more” groups. The experiments also only illustrate results where group membership is disjoint. This is addressed in the appendix (use marginals to model more complex relationships between different groups), but given that [the uncertainty of] group membership is the core motivation of the paper, it feels important for this model, and how to achieve specific instantiations of the model, to be clearer! I’d further be curious to see empirical results, though I could live without them.

Fairness metrics: Though the work as a whole claims to present a framework for a class of metrics/constraints (e.g. 158-160), empirical results are only shown (even in the appendix) for constraints/metrics that implement versions of equal representation, and thm. 3.2 is specific to equal representation (though 3.1 is more general to this class of constraints). Equal representation feels like an "easier" metric to achieve than proportional representation (similar to demographic parity being "easier" than equalized odds in classification), so I'm just curious as to what performance looks like for other constraints. Furthermore, I do wish there had been at least a nominal discussion, even in the appendix, of choice of constraint and what the consequences of different constraints would be, more than just the handwavy mention of appropriate choice being “context dependent.”

/// Typos ///
- 191 problem 2.5 format?
- 224 constraint 5 appl[ies]
- 249 “solve[d]”
- 311 “l[o]sing”
- 399 “works [for?] a general”

---

> ### Author Response · Authors · 2022-08-02
> **Response to Reviewer 6pGr (1/2)**
>
> *(This is part one of a two part response.)*
>
> Thanks for your insightful questions and suggestions. We are glad that you appreciate the storytelling and empirical results. We have updated the paper to incorporate your suggestions (and uploaded the revision to OpenReview). Below, we answer your specific questions and outline the changes in the paper to incorporate your suggestions.
>
> **"unless I'm ... misinterpreting figure 10b ... this seems to be untrue?"** Sorry for the confusion. When we say "has better or similar RD than each baseline at all noise levels" we referred to Figure 10(a) in Supplementary Material B.5. Figure 10(a) shows that for each $\eta\geq 0.1$, NResilient's RD is at least 6.8\% better than any baseline's RD. Figure 10(b), on the other hand, plots the utility (and not the RD) at different noise levels $\eta$. In the revised version, we have clarified this (Line 349 of the revised submission).
>
> **"given that [the uncertainty of] group membership is the core motivation ... it feels important ... [for the] specific instantiations of the model to be clearer!"** Thanks for this suggestion. As you note, we discuss this in Lines 675-684 in Supplementary Material A of the initial submission (Lines 720-729 in Supplementary Material A of the revised submission). In the final version, we will use the additional page provided to move these details from Supplementary Material A to the main body.
>
> **"I'd further be curious to see empirical results [with overlapping groups]"**
>         We added a variant of the simulation in Figure 3 (in Section 3 of both the initial and revised submission) that considers four overlapping groups: The sets of all women players, all male players, all non-White players, and all White players. (The simulation in Figure 3 itself uses considers four disjoint groups: The sets of non-White non-men players, White non-men players, non-White men players, and White men players.)
>
> In the new simulation, we estimate the matrix $\hat{P}$ as $\hat{P}_i {}_f =p_f(i)$, $\hat{P}_i {}_m =1-p_f(i)$, $\hat{P}_i {}_n {}_w=p_n{}_w(i)$, and $\hat{P}_i{}_w=1-p_n{}_w(i)$, where $p_f(i)$ and $p_n{}_w(i)$ are values output by Genderize API and EthniColr Library that estimate the probability that player $i$ is labeled as a women and non-white respectively. The results of this simulation are similar to the corresponding simulation on the same data with disjoint groups (in Figure 3 in Section 4 of both the revised submission and the initial submission):
>
> - Compared to other baselines, NResilient achieves the highest RD. The maximum RD of NResilient in this simulation is 0.64 compared to 0.67 in Figure 3. (Figure 14 in Supplementary Material H of the revised submission.)
> - SJ achieves the next highest RD followed by MC as in Figure 3. (Figure 14 in Supplementary Material H of the revised submission.)
>
> (CSV and GAK require protected groups to be disjoint and, hence, are not applicable in this simulation.) In the revised submission, we include the details of this simulation in Supplementary Material H.
>
>
> **"Equal representation feels like an `easier' metric ... than proportional representation ... I'm just curious as to what performance looks like for other constraints"** We added analogs of the simulations in Figures 1, 2, and 3 (in Section 4 of both the initial and the revised submission) that use proportional representation fairness constraints. To measure the deviation of a ranking from proportional representation, we consider an adaptation of weighted risk-difference metric, Prop-RD (Equation (77) in Supplementary Material G of the revised submission).
>
> In the new simulations, we observe that, relative to the baselines, NResilient's performance is similar to Figures 1, 2, and 3 (in Section 4 of both the initial and the revised submission):
> - In all simulations, NResilient achieves a higher value of the fairness metric than any baselines, as in Figures 1, 2, and 3 (in Section 4 of both the initial and the revised submission). (See Figures 11, 12, and 13 in Supplementary Material G of the revised submission.)
> - With the real-world image data, NResilient has a better fairness-utility trade-off than all baselines, as in Figure 2 (in Section 4 of both the initial and the revised submission).  (See Figure 12 in Supplementary Material G of the revised submission.)
>
> One difference is that, with the synthetic data, the value of the fairness metric achieved by NResilient can be non-monotonous in $\phi$, whereas it is increasing in $\phi$ in Figure 1 (in Section 4 of both the initial and the revised submission). (See Figure 11 in Supplementary Material G of the revised submission.) In the revised submission, we include the details from these simulations in Supplementary Material G.

---

> > ### Author Response · Authors · 2022-08-02
> > **Response to Reviewer 6pGr (2/2)**
> >
> > *(This is part two of a two part response.)*
> >
> >
> > **"thm. 3.2 is specific to equal representation"** Theorem 3.2 works for some upper bounds beyond those encoding the equal representation constraints: The proof of Theorem 3.2 holds for any upper bounds $U$ such that, for each position $k$, there is a value $\ell$, such that $U_{k\ell}$ is at most $\frac{k}{4}$. Apart from upper bounds for equal representation, upper bounds for proportional representation constraints also satisfy this condition for any $p\geq 4$.
> >
> > In the initial submission, we mention this in Lines 936-938 of Supplementary Material D.3 (Lines 986-989 in the revised submission). In the revised submission, we clarify this in the statement of Theorem 3.2 (Lines 257-258 in Section 3) and retain the explanation in Supplementary Material D.3.
> >
> > **"do you mean you ran the simulation procedure 500/1000 times?" and "what do results look like averaged over fewer iterations?"** Yes, we mean that we ran the simulations 500 or 1000 times, and reported the means and  standard-errors of the mean across the repetitions. With 25, 50, and 100 iterations, we observe that:
> > - the error bars for both utility and fairness (w.r.t. RD) are a larger (up to 0.025 compared to at most 0.0125 with 500/1000 iterations) (Figures 20-22 in Supplementary Material L of the revised submission and Figures 1-2 in Section 4 of both the initial and the revised submission)
> > - the mean utilities and fairness (w.r.t. RD) of all algorithms at all values of $\phi$ are additively within 0.05 of their corresponding values in Figures 1-3 (in Section 4 of both the initial and the revised submission.) (Figures 20-22 in Supplementary Material L of the revised submission.)
> >
> > Moreover, the relative order of the algorithms with respect to both their fairness (w.r.t. RD) and utility is the same as in Figures 1-3 (in Section 4 of both the initial and the revised submission) for all $\phi$  (Figures 20-22 in Supplementary Material L of the revised submission). In the revised submission, we include the details of these results in Supplementary Material L.
> >
> >
> > **"What does [$\phi$] represent? Why is it interesting or notable that the method presented in this work can achieve higher RD for a given value of phi?"** $\phi$ encodes the desired "level"** of equal representation. For instance, $\phi=1$ requires the output ranking to exactly satisfy the equal representation constraints, larger values of $\phi$ allow larger relaxations to equal representation constraints, and $\phi=p$ places no constraint on the output ranking. Since the level of fairness can vary depending on the context, it is desirable to have an algorithm that performs well for all values of $\phi$. For instance, to satisfy the 80\% rule one has to choose $\phi\leq 1.11$ with two groups, the presented method (NResilient) achieves a higher RD than baselines for all $\phi\leq 1.2$ (Figures 1-3 in Section 4 of the initial and revised submission). In the revised submission, we add discussions on the relevant ranges of $\phi$ in Section 3 (Lines 296-298, 321-322, 347-348, and 370-372).
> >
> > **"I would have appreciated some discussion of these assumptions and choices (e.g. the existence of the utility matrix, choice of fairness constraint and associated values), but understand the need for a focused work."** In the revised submission of the paper, we discuss the following points:
> >
> > - The entries of $W$ may be skewed by an unknown amount [Kleinberg and Raghavan,  ITCS, 2018] [Celis, Mehrotra, and Vishnoi, 2020, FAT*] or not known accurately [Singh, Kempe, and Joachims, NeurIPS, 2021] (Lines 149-151 in Section 2 of the revised submission).
> > - The utility of the ranking may not be linear in the entries of $W$ [Agrawal, Gollapudi, Halverson, and Ieong, WSDM, 2009] (Line 151 in Section 2 of the revised submission).
> > - The class of fairness constraints we study do not capture qualitative differences among groups (such as, misrepresentation of demographics in image results [Kay, Matuszek, and Munson, CHI, 2015] [Noble, NYU Press, 2018]), which could arise even when the ranking has a sufficient number of items from each group (Lines 165-167 in Section 2 and Lines 419-420 in Section 6 of the revised submission).

---

> > > ### Comment · Reviewer_6pGr · 2022-08-08
> > > **thank you!**
> > >
> > > Thanks for the thorough responses and corresponding revisions - esp. the new experiments :)

---

### Official Review · Reviewer_3sFW · 2022-07-13

**Rating:** 6
**Confidence:** 3
**Soundness:** 3 good
**Presentation:** 3 good
**Contribution:** 3 good

**Summary:**

The paper studies the fair ranking problem under a model where socially-salient (protected) attributes of items are randomly and independently perturbed. The authors provide a fair ranking framework that incorporates group fairness requirements along with probabilistic information about perturbations in socially-salient attributes. This novel framework works for multiple non-disjoint attributes and a general class of fairness constraints that include proportional and equal representation. In the experiments, it has been observed that, this new algorithm outputs rankings with higher fairness compared to baselines, and has a similar or better fairness-utility trade-off compared to them.

**Questions:**

I have some questions that I mention below:

1) line 4: "socially-salient (including protected) attributes" Aren't they the same?

2) line 159: I don't think that [54] uses the similar constraints. The exposure based set of fairness constraints are different form constraints used in this paper. You also have used [54] as a baseline. What fairness constraints you used for this method. Did you implement RD or the demographic parity used in the paper?

3) line 221: how did you find constraint (6)?

4) line 267: The algorithm runs in polynomial time in d and the bit complexity of input --> what does it mean bit complexity?

5) Why not using [1] as a baseline in your experiments? This method seems to perform the same task as your approach.

[1] Serena Wang, Wenshuo Guo, Harikrishna Narasimhan, Andrew Cotter, Maya R. Gupta, and Michael I. Jordan. Robust Optimization for Fairness with Noisy Protected Groups. In NeurIPS, 2020.




**Limitations:**

The limitations are discussed briefly in the conclusion:

"Compared to existing fair-ranking frameworks, our framework does not need accurate socially-salient attributes, but assumes that errors in attributes are random and independent."

**Strengths And Weaknesses:**

Strengths:

The paper is well-written and easy to read. The paper proposes a solution for this problem that can be applied to multiple non-disjoint attributes and both proportional and equal representation classes of fairness constraints. Basically, the paper discusses a problem and the corresponding optimization to solve, and then discusses the impossibility of solving such a problem (due to it being NP-hard), then simplifies the problem in order to solve the optimization with guaranteed properties.
The paper looks theoretically sound with clearly described assumptions and has provided proofs when needed. In the experiments section, the method has been evaluated on synthetic and real-world data. The authors have compared their approach against 5 different baselines and shown that their approach outperforms others.

Weaknesses/Concerns:

I think the paper is technically sound and well-written, yet it operates under strong (though clearly stated) assumptions, which makes its contributions of interest only to a relatively narrow audience.

I have a few concerns/suggestions to improve the presentation of the paper:

1) line 22: recruiting problems [38, 11, 7] --> add dot to the end of sentence.

2) line 32: equal representation requires that .. --> add citation for equal representation

3) line 34: "Proportional representation requires ..." --> add citation

4) line 98: remove extra "that"

5) line 224: Constraint (5) apply --> applies

6) line 225: Further, Constraint (5) --> shouldn't be it Constraint (8)??

7) line 276: we evaluate our framework's performance synthetic --> on synthetic

8) line 311: "We observe that NResilient achieves best RD (≈0.81), while not loosing significant utility (≥ 0.98% of maximum; see Figure 4). MC achieves the best RD (≈0.79)." --> both methods achieve best RD?!

9) Missing related work: This work studies fair ranking with noisy protected attributes. Related to this work, there is a recent work that studies fair ranking in the presence of label noise [1]. It is an in-processing method which presents a preferable trade-off between fairness and utility.

[1] Memarrast, Omid, Ashkan Rezaei, Rizal Fathony, and Brian Ziebart. "Fairness for Robust Learning to Rank." arXiv preprint arXiv:2112.06288 (2021).

---

> ### Author Response · Authors · 2022-08-02
> **Response to Reviewer 3sFW**
>
> Thank you for your feedback – we are happy that you found the paper to be well-written. We have updated the paper on OpenReview to address your questions and incorporate your suggestions. We answer your specific questions below.
>
> **"shouldn't be it Constraint (8)"** Yes, we have updated this, see Line 234 of the revised submission.
>
> **"Missing related work"** Thanks, we have included this work in the related work section (Lines 122-125 of the revised submission).
>
> **"Aren't [socially-salient attributes and protected attributes] the same?"** Legally protected attributes (such as race, gender, and age) are also socially salient. However, some socially-salient attributes such as education-level or dialect may not be legally protected.
>
> **"I don't think that [54] uses the similar constraints. The exposure based set of fairness constraints are different"** Sorry for the confusion – we meant that [54] give an algorithm that can take the constraints in Definition 2.1 as input and output a ranking that maximizes the utility subject to satisfying these constraints. We did not mean that the exposure based set of fairness constraints in [54] are the same as the fairness constraints in Definition 2.1.  We have made this distinction clear in the revision (Lines 742-751 of Supplementary Material B).
>
> **"You also have used [54] as a baseline. What fairness constraints you used for this method. Did you implement RD or the demographic parity used in the paper?"** In our submission, we implemented [54]'s algorithm for the following  constraints: For each $k$ and $\ell$, $U_{k\ell}=\frac{k}{p}\cdot \phi$ (See Lines 305-306 in Section 4 of the revised response). When $\phi=1$, the above constraints specialize to demographic parity. We also applied RD to measure the fairness of rankings output by [54]'s algorithm.
>
> **"how did you find constraint (6)?"** We chose $\gamma_k$ (defined in Constraint (6)) to be the smallest value, up to logarithmic factors, such that any ranking that satisfies the $(\gamma,\delta)$-constraint must also satisfy the fairness constraint specified by $\gamma_k$ in Equation (8) (of both the initial and the revised submission) for each position $k$. In the revised submission, we specify this in Lines 238-239 of Section 3.
>
> **"what does it mean bit complexity?"** The bit complexity of the input is the number of bits required to encode the input using the standard binary encoding (which encodes integers in their binary representation, rational numbers as pair of integers, and vectors/matrices as a tuple of their entries). In the revised submission, we include this detail and specific references for bit complexity in Footnote 1 on Page 6 in Section 3.
>
> **"Why not using [1] as a baseline in your experiments?"** [1] considers errors in protected attributes in the data for the binary classification problem. Since binary classification and (the variant of) the ranking problem we study are  different, we are not sure how to adapt their approach for our simulations.

---

### Meta-Review · Area_Chair_tp4B · 2022-08-30

**Recommendation:** Accept
**Confidence:** Certain

**Metareview:**

This paper looks at the fair ranking problem, a known variant of ranking where group fairness constraints (typically hard, sometimes soft) are imposed on the traditional ranking objective, but where membership of each item to be ranked in a group (aka the sensitive attribute's value associated with that item) is unknown.  The paper provides strong theoretical results and, especially post-rebuttal, strong experimental backing of the setting at hand.  Some assumptions are relatively strong, as surfaced by reviewers (e.g., 3sFW), but by and large reviewers believed the work to be well motivated and complete, and I agree with that.

**Award:**

No

---

### Decision · Program_Chairs · 2022-09-14

Accept